# KL-Regularization Is Sufficient in Contextual Bandits and RLHF

## Abstract

Recently, reinforcement learning from human feedback (RLHF) has demonstrated remarkable efficiency in fine-tuning large language models (LLMs), fueling a surge of interest in KL-regularization. Yet, the theoretical foundations of KL-regularization remain underexplored. Many prior works employ either explicit online exploration strategies—such as UCB, Thompson sampling, and forced sampling—or optimism-embedded optimization techniques (e.g., Xie et al. 2024) *in addition to KL-regularization* to achieve sublinear regret in online RLHF. In this paper, we show, for the first time to our best knowledge, that such additional exploration strategies are unnecessary if KL-regularization is already included. That is, KL-regularization alone suffices to guarantee sublinear regret. To handle general function classes, we assume access to an online regression oracle and propose `KL-EXP` (and its RLHF variant, `OEPO`), which achieves logarithmic KL-regularized regret—the standard objective in KL-regularized contextual bandits and RLHF—while also attaining an *unregularized* regret of $\mathcal{O}(\sqrt{\log N \cdot T \text{Reg}_{\text{Sq}}(T)})$, where $N$ is the number of actions, $T$ is the total number of rounds, and $\text{Reg}_{\text{Sq}}(T)$ is the online regression oracle bound. To the best of our knowledge, this is the first result to achieve regret with only logarithmic dependence on $N$ in oracle-based contextual bandits. As a special case, in linear contextual bandits, our result yields an unregularized regret of $\tilde{\mathcal{O}}(\sqrt{dT \log N})$, where $d$ is the feature dimension. To our best knowledge, this is the first $\tilde{\mathcal{O}}(\sqrt{dT \log N})$-type regret bound achieved without resorting to supLin-type algorithms, making it substantially more practical.

## 1 Introduction

The Kullback–Leibler (KL)-regularized contextual bandit problem (Langford & Zhang, 2007; Neu et al., 2017; Xiong et al., 2023; Xie et al., 2024) has recently attracted considerable attention due to its remarkable empirical success in fine-tuning large language models (LLMs), an application commonly referred to as reinforcement learning from human feedback (RLHF) (Christiano et al., 2017; Bai et al., 2022; Ouyang et al., 2022). This framework uses KL-regularization as a key mechanism to balance reward optimization with distributional stability.

Despite these practical successes, the theoretical understanding of KL-regularization remains limited, particularly in the context of online learning. *Online exploration* is crucial for efficiently gathering informative feedback and addressing user preferences in RLHF. In this vein, many prior works have leveraged additional mechanisms to promote exploration, such as Upper Confidence Bound (UCB) (Xiong et al., 2023; 2024; Zhao et al., 2025a), forced sampling (Zhao et al., 2024), and value-incentivized policy optimization (Xie et al., 2024; Cen et al., 2024). Building on these strategies, Xiong et al. (2023); Ye et al. (2024); Xie et al. (2024); Xiong et al. (2024); Cen et al. (2024) established $\mathcal{O}(\sqrt{T})$ bounds on *KL-regularized regret* (or $\mathcal{O}(1/\epsilon^2)$ sample complexity). More recently, Zhao et al. (2024; 2025a) achieved the first logarithmic KL-regularized regret (or $\mathcal{O}(1/\epsilon)$ sample complexity).

However, optimizing the KL-regularized objective (Equation 1) already yields a randomized policy of the Gibbs distribution form (Equation 2). This implies that KL-regularization induces inherent exploration. Therefore, a natural question arises:

> *Can logarithmic KL-regularized regret be achieved without extra exploration techniques in contextual bandits and RLHF when KL-regularization is used?*

Beyond this, we raise a more fundamental question: is achieving sublinear *KL-regularized regret*, by itself, truly sufficient? To the best of our knowledge, the tightest bound to date is $\mathcal{O}(\eta \log T)$, established by Zhao et al. (2025a), where $\eta$ is the KL-regularization parameter. A direct implication of this result is that by choosing $\eta$ to be sufficiently small, one can always guarantee an arbitrarily small KL-regularized regret. Indeed, a small $\eta$ indicates that the KL-regularized optimal policy $\pi_\eta^\star$ remains very close to the reference policy $\pi_{\text{ref}}$, which makes this result appear reasonable. However, when $\pi_\eta^\star \approx \pi_{\text{ref}}$, the learner gains little to no improvement, which is undesirable since the goal is to discover a strictly better policy than the reference policy. To address this, we also consider the notion of *unregularized regret* (Equation 3), as in standard bandit settings. This regret can be large when the policy remains close to $\pi_{\text{ref}}$ (i.e., for small $\eta$) but far from the *unregularized optimal policy $\pi^\star$*. Minimizing the unregularized regret allows us to directly pursue the unregularized optimal policy $\pi^\star$, rather than being limited to the KL-regularized solution $\pi_\eta^\star$. This naturally raises the hypothesis that $\eta$ should be chosen carefully to minimize the unregularized regret, which leads to our second question:

> *By choosing $\eta$ appropriately, can we achieve sublinear unregularized regret, still without additional exploration techniques?*

In this paper, we answer these questions affirmatively. We begin by analyzing the KL-regularized (adversarial) contextual bandit setting and then extend our analysis to RLHF. To consider general algorithms, we assume access to an online regression oracle (Foster & Rakhlin, 2020), while the offline regression oracle is discussed in Appendix F. Our main contributions are summarized as:

- **KL-regularized regret.** In KL-regularized contextual bandits, we establish a KL-regularized regret bound of $\mathcal{O}(\eta \text{Reg}_{\text{Sq}}(T) + \eta \log(1/\delta))$, where $\eta$ is the regularization parameter, $\text{Reg}_{\text{Sq}}(T)$ is the online regression oracle bound, and $\delta$ is the failure probability (Theorem 1). This result is achieved solely through KL-regularization, without relying on any additional exploration techniques. To our best knowledge, this is the first result to show the provable efficiency of the KL-regularization-only approach. Since $\text{Reg}_{\text{Sq}}(T) = \mathcal{O}(\log T)$ can be attained by suitable regression oracles for a wide range of reward functions—including linear, generalized linear, and bounded eluder-dimension function classes—we achieve logarithmic KL-regularized regret.

- **Unregularized regret.** By setting $\eta = \Theta\big(\sqrt{DT/(\text{Reg}_{\text{Sq}}(T) + \log \delta^{-1})}\big)$, we obtain an unregularized regret of $\mathcal{O}(\sqrt{DT(\text{Reg}_{\text{Sq}}(T) + \log \delta^{-1})})$, where $D = \frac{1}{T}\sum_{t=1}^T \text{KL}\big(\pi^\star(\cdot\|x_t)\|\pi_{\text{ref}}(\cdot\|x_t)\big)$ (Theorem 1). To the best of our knowledge, this is the first unregularized regret bound for KL-regularized contextual bandits attained solely through KL-regularization-induced exploration.

- **First $\sqrt{\log N}$-order regret in oracle-efficient contextual bandits.** With a uniform reference policy and $\eta = \Theta(\sqrt{T \log N}/\text{Reg}_{\text{Sq}}(T))$, we obtain an (unregularized) regret bound $\mathcal{O}(\sqrt{\log N \cdot T \text{Reg}_{\text{Sq}}(T)})$, where $N$ is the number of actions. This improves upon the previous regret bound $\mathcal{O}(\sqrt{NT \text{Reg}_{\text{Sq}}(T)})$ (Foster & Rakhlin, 2020) by reducing the dependence on $N$ from $\sqrt{N}$ to $\sqrt{\log N}$. To the best of our knowledge, this is the first result to achieve regret with only logarithmic dependence on $N$ within the oracle-efficient contextual bandit framework.

- $\tilde{\mathcal{O}}(\sqrt{dT \log N})$ **regret in linear contextual bandits.** With a uniform reference policy and $\eta = \Theta(\sqrt{T \log N/(d \log T)})$, we obtain an (unregularized) regret bound of $\tilde{\mathcal{O}}(\sqrt{dT \log N})$ for linear contextual bandits (Theorem 2), where $d$ is the feature dimension. To our best knowledge, this is the first $\tilde{\mathcal{O}}(\sqrt{dT \log N})$-type regret achieved without using on supLin-type algorithms (Auer, 2002; Chu et al., 2011; Li et al., 2019), which are known to be impractical. Hence, this is the first practical algorithm to achieve minimax optimal regret for finite-armed linear contextual bandits.

- **Extension to RLHF.** We further establish similar regret bounds in the RLHF setting, with only an additional factor due to the non-linearity of the Bradley–Terry model (Theorems 3 and E.1).

## 2 RELATED WORKS

**Online RLHF.** Early works in online RLHF trace back to the dueling bandits literature (Yue et al., 2012; Zoghi et al., 2015; Saha & Gopalan, 2018; Bengs et al., 2021) and were later extended to the reinforcement learning setting (Xu et al., 2020; Novoseller et al., 2020; Chen et al., 2022; Saha et al., 2023; Zhan et al., 2023b; Wu & Sun, 2023). More recently, Xiong et al. (2023); Ye et al.

(2024) introduced provably efficient algorithms under the KL-regularized objective using UCB-style exploration. These were further refined by methods that employ optimistically biased optimization targets (Xie et al., 2024; Liu et al., 2024; Cen et al., 2024). The most closely related works are Zhao et al. (2024; 2025a), which also study the KL-regularized objective and establish $\mathcal{O}(\eta \log T)$ KL-regularized regret (or $\mathcal{O}(\eta/\epsilon)$ suboptimality gap). However, all of these prior approaches depend on additional exploration mechanisms. In contrast, our work demonstrates—for the first time, to the best of our knowledge—that KL-regularization alone suffices to achieve sublinear regret in both the regularized and unregularized forms. For additional related work, see Appendix A.

## 3 PROBLEM SETUP

**Notations.** Given a set $\mathcal{X}$, we use $|\mathcal{X}|$ to denote its cardinality. For a positive integer, $n$, we denote $[n] := \{1, 2, \ldots, n\}$. Let $N$ denote the size of the action space. We write $\mathcal{O}(\cdot)$ for asymptotics up to constants and $\tilde{\mathcal{O}}(\cdot)$ when also hiding logarithmic factors (except in $N$). For a function class $\mathcal{F}$, we denote by $\mathcal{N}_{\mathcal{F}}(\epsilon)$ its $\epsilon$-covering number.

### 3.1 KL-REGULARIZED CONTEXTUAL BANDITS

In the KL-regularized contextual bandits, at each round $t \in [T]$, the learner observes a context $x_t \in \mathcal{X}$ (which may be provided *adversarially*) and then selects an action $a_t \in \mathcal{A}$, where $\mathcal{X}$ is the context space and $\mathcal{A}$ is the action space. The learner then receives a reward $r_t \in [0, 1]$, given by:

$$r_t = R^{\star}(x_t, a_t) + \epsilon_t,$$

where $R^{\star}(x_t, a_t)$ is the unknown expected reward function, and $\epsilon_t$ is independent, zero-mean, and 1-sub-Gaussian. In this paper, we consider a general reward function class $\mathcal{R} \subseteq \{R : \mathcal{X} \times \mathcal{A} \to [0, 1]\}$, which can be a class of parametric functions, nonparametric functions, neural networks, etc.

**Assumption 1** (Realizability). *The true reward function is contained in $\mathcal{R}$, i.e., $R^{\star} \in \mathcal{R}$.*

**Assumption 2** (Pointwise relative interior). *For each $(x, a) \in \mathcal{X} \times \mathcal{A}$, define $S_{x,a} := \{R'(x, a) : R' \in \mathcal{R}\} \subseteq [0, 1]$. We assume $R(x, a) \in \mathrm{ri}_{[0,1]}(S_{x,a})$, i.e., there exists $\varepsilon_{x,a} > 0$ such that $\big(R(x, a) - \varepsilon_{x,a}, R(x, a) + \varepsilon_{x,a}\big) \cap [0, 1] \subseteq S_{x,a}$.*

Assumption 1 corresponds to the standard *realizability* assumption commonly adopted in prior works (Chu et al., 2011; Agarwal et al., 2012; Foster et al., 2018a; Foster & Rakhlin, 2020; Simchi-Levi & Xu, 2022). Assumption 2 ensures differentiability of the functions defined later with respect to $R(x, a)$ over $\mathcal{R}$. This assumption holds for most bandit settings (e.g., multi-armed, linear, GLM, and neural bandits), with the exception of finite function classes (Agarwal et al., 2012)[1]. Note that this assumption has been overlooked and not explicitly stated in prior works whose analyses similarly rely on differentiating certain reward-dependent functions to obtain logarithmic regret (Zhao et al., 2024; 2025a;b); it should have been made explicit in those papers as well.

**KL-Regularized Objective.** We consider a *KL-regularized* reward objective, defined for a regularization parameter $\eta > 0$, as:

$$J_t^{\eta}(\pi, R) := \mathbb{E}_{a \sim \pi(\cdot|x_t)}\left[R(x_t, a)\right] - \frac{1}{\eta} \mathrm{KL}\big(\pi(\cdot|x_t)\|\pi_{\mathrm{ref}}(\cdot|x_t)\big), \quad \forall t \geq 1, \tag{1}$$

where $\pi_{\mathrm{ref}}$ is the reference policy known to the learner. When $\pi_{\mathrm{ref}}$ is uniform, Equation 1 reduces to the entropy-regularized objective that encourages diverse actions and enhances robustness (Williams, 1992; Levine & Koltun, 2013; Levine et al., 2016; Haarnoja et al., 2018), which is also closely-related to the generative flow networks (GFlowNets) (Bengio et al., 2021; 2023; Tiapkin et al., 2024). When $\pi_{\mathrm{ref}}$ is instead chosen as a base model, KL-regularization has been widely adopted for RL fine-tuning of large language models (Ouyang et al., 2022; Rafailov et al., 2023). It has also been studied in online learning (Cai et al., 2020; He et al., 2022) and convex optimization (Neu et al., 2017).

Following prior work (Peters & Schaal, 2007; Rafailov et al., 2023; Zhang, 2023), it is straightforward to show that the optimal solution to the objective in Equation 1 has the following form:

$$\pi_R^{\eta}(a|x) = \frac{1}{Z_R(x)} \pi_{\mathrm{ref}}(a|x) \exp\left(\eta R(x, a)\right), \tag{2}$$

---

[1]For finite function classes, one may instead consider their convex hull $\mathrm{conv}(\mathcal{R})$ to satisfy Assumption 2.

where $Z_R(x) := \mathbb{E}_{a \sim \pi_{\text{ref}}(\cdot|x)} \exp(\eta R(x, a))$ is the normalization constant. A full derivation can be found in Appendix A.1 of Rafailov et al. (2023).

## 3.2 Reinforcement Learning from Human Feedback (RLHF)

In the RLHF problem (Ouyang et al., 2022)—more specifically, the contextual *dueling* bandit problem with a KL-regularized objective—the learner at each round $t \in [T]$ observes a context $x_t \in \mathcal{X}$ (possibly provided *adversarially*) and selects two actions $a_t^1, a_t^2 \in \mathcal{A}$, where $\mathcal{X}$ is the context space and $\mathcal{A}$ the action space. The learner then receives relative preference feedback between the two actions, rather than a scalar reward. In this paper, we consider the Bradley-Terry Model (Bradley & Terry, 1952), where the probability of $a^1$ is preferred over $a^2$ (denoted by $a^1 \succ a^2$) is given by

$$\mathbb{P}(a^1 \succ a^2 | x, a^1, a^2) = \sigma\left(R^\star(x, a^1) - R^\star(x, a^2)\right),$$

where $\sigma(x) = \frac{1}{1+e^{-x}}$ is the sigmoid function, and $R^\star : \mathcal{X} \times \mathcal{A} \to [0, 1]$ the *unknown true* reward function. We denote $\mathcal{R} \subseteq \{R : \mathcal{X} \times \mathcal{A} \to [0, 1]\}$ as the class of reward functions. To capture the non-linearity of the sigmoid function, we define $\kappa := \sup_{R \in \mathcal{R}, x \in \mathcal{X}, a \in \mathcal{A}} 1/\dot{\sigma}(R(x, a))$. As in the bandit setting, we update the policy by optimizing the KL-regularized reward objective (Equation 1).

## 3.3 KL-Regularized and Unregularized Regret

We study two types of regret to more comprehensively evaluate the performance of our algorithm.

**KL-regularized regret.** Let $\pi_\eta^\star(\cdot|x_t) = \operatorname{argmax}_\pi J_t^\eta(\pi, R^\star)$ denote the *KL-regularized optimal policy*. Our objective is to minimize the cumulative regret, defined as:

$$\mathbf{Regret}_{\text{KL}}(T, \eta) := \sum_{t=1}^T \left(J_t^\eta(\pi_\eta^\star, R^\star) - J_t^\eta(\pi_t, R^\star)\right).$$

This KL-regularized regret has been extensively studied in the prior literature (Xiong et al., 2023; Ye et al., 2024; Song et al., 2024; Zhao et al., 2024; 2025a).

**Unregularized regret.** Beyond the KL-regularized regret, we also measure performance relative to the *unregularized optimal policy* $\pi^\star(\cdot|x_t) = \operatorname{arg max}_\pi \mathbb{E}_{a \sim \pi(\cdot|x_t)}[R^\star(x_t, a)]$, and define the corresponding regret as follows:

$$\mathbf{Regret}(T) := \sum_{t=1}^T \left(\mathbb{E}_{a \sim \pi^\star(\cdot|x_t)}[R^\star(x_t, a)] - \mathbb{E}_{a \sim \pi_t(\cdot|x_t)}[R^\star(x_t, a)]\right). \tag{3}$$

The notion of this regret is standard in conventional bandit problems. This metric enables a more direct evaluation of how closely the learned policies approach the unregularized optimal policy.

## 4 KL-Regularized Contextual Bandits

In this section, we consider KL-regularized contextual bandit problems. We introduce the notion of an online regression oracle (Subsection 4.1), present our algorithm `KL-EXP` together with its regret bounds (Subsection 4.2), and provide a proof sketch (Subsection 4.3).

### 4.1 Squared-loss online regression oracle.

We assume access to a squared-loss online regression oracle (Foster & Rakhlin, 2020), denoted by `OracleSq`. At each round $t$, `OracleSq` outputs a reward estimator

$$\widehat{R}_t \leftarrow \texttt{OracleSq}_t\left((x_1, a_1, r_1), \ldots, (x_{t-1}, a_{t-1}, r_{t-1})\right), \quad \text{where } \widehat{R}_t \in \mathcal{R}. \tag{4}$$

Unlike Foster & Rakhlin (2020), we require $\widehat{R}_t \in \mathcal{R}$, a condition readily met when $\mathcal{R}$ is sufficiently rich. In conjunction with Assumption 2, this guarantees differentiability at $\widehat{R}_t(x, a)$. The prediction error of `OracleSq` is assumed to be bounded with respect to the true reward function $R^\star$.

---

**Algorithm 1** KL-EXP (**KL**-regularized **EXP**onential-weights algorithm)

---

1: **Inputs:** regularization parameter $\eta$, reference policy $\pi_{\text{ref}}$, online regression oracle `OracleSq`.
2: **Initialize:** choose any $\widehat{R}_1 \in \mathcal{R}$.
3: **for** round $t = 1$ to $T$ **do**
4:      Observe context $x_t \in \mathcal{X}$.
5:      Compute policy $\pi_t(\cdot|x_t) \propto \pi_{\text{ref}}(\cdot|x_t) \exp\big(\eta \widehat{R}_t(x_t, \cdot)\big)$ via Equation 2.
6:      Sample action $a_t \sim \pi_t(\cdot|x_t)$ and receive reward $r_t$.
7:      Update $\widehat{R}_{t+1}$ using `OracleSq` via Equation 4.
8: **end for**

---

**Assumption 3** (Guarantee of `OracleSq`). *We assume that, for every sequence $x_{1:T}, a_{1:T}, r_{1:T}$, there exists regret bound $\text{Reg}_{\text{Sq}}(T)$ such that the regression oracle `OracleSq` satisfies*

$$\sum_{t=1}^{T} \big(\widehat{R}_t(x_t, a_t) - r_t\big)^2 - \sum_{t=1}^{T} \big(R^\star(x_t, a_t) - r_t\big)^2 \leqslant \text{Reg}_{\text{Sq}}(T).$$

An important advantage of Assumption 3 is that it places no restriction on how the estimator $\widehat{R}_t$ is obtained; in particular, it does not require solving ERM exactly. Instead, $\widehat{R}_t$ can be computed via iterative methods such as (stochastic) gradient descent and implemented in an online or streaming manner, which is crucial for large-scale modern machine learning. Under realizability (Assumption 1), Assumption 3 is weaker than Assumption 2a in Foster & Rakhlin (2020), since we compete only against the fixed $R^\star$, whereas they compete against the best predictor over the sequence.

The online squared-loss regression problem is well studied, with efficient algorithms and regret guarantees for many function classes.

**Example 1** (Linear classes). *When $R^\star \in \mathcal{R}$ and the reward function class $\mathcal{R}$ is linear, i.e., $\mathcal{R} = \{R : R = \phi(x,a)^\top \theta, \theta \in \mathbb{R}^d, \|\theta\|_2 \leqslant 1\}$, where $\phi(x,a) \in \mathbb{R}^d$ is a known feature map satisfying $\|\phi(x,a)\|_2 \leqslant 1$, choosing `OracleSq` as the Vovk–Azoury–Warmuth forecaster (Vovk, 1997; Azoury & Warmuth, 2001) yields $\text{Reg}_{\text{Sq}}(T) = \mathcal{O}(d\log(T/d))$.*

**Example 2** (Generalized linear models (GLMs)). *For a fixed non-decreasing 1-Lipschitz link function $\mu : \mathbb{R} \to [0,1]$, define the reward function class $\mathcal{R} = \{R : R = \mu(\phi(x,a)^\top \theta), \theta \in \mathbb{R}^d, \|\theta\|_2 \leqslant 1\}$, where $\phi(x,a) \in \mathbb{R}^d$ is a known feature map with $\|\phi(x,a)\|_2 \leqslant 1$. If $R^\star \in \mathcal{R}$, then the `GLMtron` algorithm (Kakade et al., 2011) guarantees $\text{Reg}_{\text{Sq}}(T) = \mathcal{O}(\kappa_\mu^2 d\log(T/d))$, where $1/\dot{\mu} \leqslant \kappa_\mu$.*

**Example 3** (Bounded eluder dimension, Russo & Van Roy, 2013). *When $R^\star \in \mathcal{R}$ and the reward function class $\mathcal{R}$ has bounded eluder dimension, the empirical risk minimization (ERM) algorithm achieves, with probability at least $1 - \delta$, $\text{Reg}_{\text{Sq}}(T) = \mathcal{O}\big(d_{\text{E}} \log(\mathcal{N}_\mathcal{R}(\epsilon)T)\big)$ (Lemma C.2).*

For additional examples, the reader is referred to Foster & Rakhlin (2020) for high-dimensional linear models, Banach spaces, and RKHS, and to Deb et al. (2024) for neural networks.

## 4.2 ALGORITHM AND MAIN RESULTS

We present our KL-regularized EXPonential-weights algorithm, KL-EXP, in Algorithm 1. At each round $t \in [T]$, the algorithm observes the context $x_t \in \mathcal{X}$ and computes the policy $\pi_t$ by solving the KL-regularized objective in Equation 1, which admits the closed-form solution given in Equation 2. The algorithm then samples an action $a_t \sim \pi_t(\cdot|x_t)$ and receives a reward $r_t$. Finally, it updates the reward estimator $\widehat{R}_{t+1}$ for the next round using the squared-loss online regression oracle (Equation 4).

**Remark 1** (Ease of implementation and computational efficiency). *KL-EXP is simple and practical: it admits a closed-form solution (Equation 2) and—unlike prior approaches with general function approximation (Russo & Van Roy, 2013; Jiang et al., 2017; Jin et al., 2021; Zhao et al., 2025a)—does not require explicit computation of exploration terms (e.g., UCB), which is often intractable for large models such as transformers. It is also computationally efficient. In linear contextual bandits (ignoring oracle-related computations), the per-round cost is only $\mathcal{O}(N)$, where $N = |\mathcal{A}|$, whereas LinUCB and LinTS require $\mathcal{O}(d^2 N)$ per round.*

The main guarantees for the algorithm are stated below, with the proof deferred to Appendix B.

**Theorem 1** (Regret of `KL-EXP`). *Let $\delta > 0$ and $D := \frac{1}{T}\sum_{t=1}^{T} \mathrm{KL}\big(\pi^\star(\cdot\|x_t)\|\pi_{\mathrm{ref}}(\cdot\|x_t)\big)$. Under Assumption 1- 3, with probability at least $1 - \delta$, `KL-EXP` (Algorithm 1) guarantees*

$$\mathbf{Regret}_{\mathrm{KL}}(T, \eta) = \mathcal{O}\Big(\eta\mathrm{Reg}_{\mathrm{Sq}}(T) + \eta\log(1/\delta)\Big) \quad and$$

$$\mathbf{Regret}(T) = \mathcal{O}\left(\eta\mathrm{Reg}_{\mathrm{Sq}}(T) + \eta\log(1/\delta) + \frac{DT}{\eta}\right).$$

**Result 1: Logarithmic KL-regularized regret.** Theorem 1 shows that the KL-regularized regret of `KL-EXP` scales with $\mathrm{Reg}_{\mathrm{Sq}}(T)$, resulting in logarithmic regret in $T$ across a broad range of function classes. For example, when $\delta = \Theta(T^{-1})$, we obtain $\mathcal{O}(\eta d\log T)$ for linear classes (Example 1), $\mathcal{O}(\eta\kappa_\mu^2 d\log T)$ for generalized linear models (Example 2), and $\mathcal{O}(\eta d_{\mathrm{E}}\log(\mathcal{N}_{\mathcal{R}}(\epsilon)T))$ for function classes with bounded eluder dimension (Russo & Van Roy, 2013) (Example 3). Hence, Theorem 1 shows that logarithmic KL-regularized regret in $T$ can be achieved without the *auxiliary exploration methods* (e.g., UCB-based strategies). In contrast, prior works such as Xiong et al. (2023; 2024); Xie et al. (2024) obtained $\mathcal{O}(\sqrt{T})$ KL-regularized regret (or $\mathcal{O}(1/\epsilon^2)$ sample complexity), and more recently, Zhao et al. (2024; 2025a) established $\mathcal{O}(\eta\log T)$ KL-regularized regret (or $\mathcal{O}(\eta/\epsilon)$ sample complexity), all of which depend on the additional exploration strategies. To the best of our knowledge, this is the first result that achieves logarithmic KL-regularized regret without any additional exploration, highlighting the key insight that the KL-regularized objective alone provides sufficient exploration in contextual dueling bandits and RLHF.

**Remark 2** (Comparison with Zhao et al. (2025a)). *For classes with bounded eluder dimension, we recover the regret bound of Zhao et al. (2025a), $\mathcal{O}\big(\eta d_{\mathrm{E}}\log(\mathcal{N}_{\mathcal{R}}(\epsilon)T)\big)$. Unlike Zhao et al. (2025a), however, our algorithm does not require prior knowledge of the eluder dimension (Russo & Van Roy, 2013), which is typically unknown in practice. The full proof is provided in Appendix C.*

**Result 2: Unregularized regret and its tightness.** With the choice of the regularization parameter $\eta = \Theta\big(\sqrt{DT/(\mathrm{Reg}_{\mathrm{Sq}}(T) + \log\delta^{-1})}\big)$, we obtain $\mathbf{Regret}(T) = \mathcal{O}(\sqrt{DT(\mathrm{Reg}_{\mathrm{Sq}}(T) + \log\delta^{-1})})$. The result provides an interesting insight: with *appropriately chosen* $\eta$, it is possible to achieve a $\sqrt{T}$-type regret bound even in conventional (unregularized) contextual bandit problems. To the best of our knowledge, this is the first unregularized regret bound in KL-regularized contextual bandits achieved purely via KL-regularization–induced exploration.

To demonstrate the tightness of our bound, we consider the uniform reference policy $\pi_{\mathrm{ref}} = \mathrm{Unif}(\mathcal{A})$, under which $\mathrm{KL}(\pi\|\pi_{\mathrm{ref}}) \leq \log N$ holds for any policy $\pi$. Under this setting, our result gives $\mathbf{Regret}(T) = \mathcal{O}(\sqrt{\log N \cdot T\mathrm{Reg}_{\mathrm{Sq}}(T)})^2$, which improves upon the previous bound $\mathcal{O}(\sqrt{NT\mathrm{Reg}_{\mathrm{Sq}}(T)})$, achieved by `SquareCB` (Foster & Rakhlin, 2020), reducing the dependence from $\sqrt{N}$ to $\sqrt{\log N}$—except in finite function classes[3], where our analysis does not directly apply. To the best of our knowledge, this is the first work to break the $\sqrt{N}$ barrier and achieve regret with only logarithmic dependence on $N$ within the oracle-efficient contextual bandit framework.

Furthermore, for linear (adversarial) contextual bandits, we obtain the first $\tilde{\mathcal{O}}(\sqrt{dT\log N})$-type regret bound, to the best of our knowledge.

**Theorem 2** (Unregularized regret under linear classes). *We denote $N = |\mathcal{A}|$. Under the setting of Theorem 1, if we set $\pi_{\mathrm{ref}} = \mathrm{Unif}(\mathcal{A})$ and $\eta = \Theta(\sqrt{T\log N/(d\log T)})$, then with probability at least $1 - \frac{1}{T}$, we have $\mathbf{Regret}(T) = \mathcal{O}\big(\sqrt{dT\log N\log T}\big)$.*

The proof of Theorem 2 follows directly from two facts: $\mathrm{Reg}_{\mathrm{Sq}}(T) = \mathcal{O}(d\log(T/d))$ (Example 1) and $\mathrm{KL}(\pi^\star\|\pi_{\mathrm{ref}}) \leq \log N$ when $\pi_{\mathrm{ref}} = \mathrm{Unif}(\mathcal{A})$.

**Remark 3** (Minimax-optimality under linear classes). *We highlight that, in linear contextual bandits, our regret bound $\mathcal{O}\big(\sqrt{dT\log N\log T}\big)$ is minimax-optimal, matching the order previously attained by supLin-type algorithms (Auer, 2002; Chu et al., 2011; Li et al., 2019). To the best of our knowledge, this is the first $\tilde{\mathcal{O}}(\sqrt{dT\log N})$-type regret bound for linear (adversarial) contextual bandits*

---

[2]We set $\delta = 1/T$ and omit the $\log\delta^{-1}$ term, since $\log\delta^{-1} = \log T \leq \mathrm{Reg}_{\mathrm{Sq}}(T)$ for most cases.

[3]Recall that Assumption 2 does not hold for finite function classes.

*that avoids the impractical "layered data partitioning" technique and explicit UCB computations. Moreover, it matches the lower bound $\Omega\left(\sqrt{dT \log N \log(T/d)}\right)$ (Li et al., 2019) up to logarithmic $d$ factors, underscoring both the statistical and computational efficiency of our approach.*

Further examples for specific function classes are provided in Appendix B.4.

### 4.3 PROOF SKETCH OF THEOREM 1

**1) Second-order regret decomposition.** The regret decomposition is similar to the recent work of Zhao et al. (2025a), which establishes logarithmic KL-regularized regret. Define the function $f(x, R) := -\frac{1}{\eta} \log Z_R(x) + \mathbb{E}_{\pi_R^\eta} [R(x, a) - R^\star(x, a)]$. Since $R^\star(x, a) = \frac{1}{\eta} \log \exp(\eta R^\star(x, a))$, the unregularized regret at round $t$ can be written as follows:

$$J_t^\eta(\pi_\eta^\star, R^\star) - J_t^\eta(\pi_t, R^\star) = \frac{1}{\eta} \log Z_{R^\star}(x_t) - \frac{1}{\eta} \log Z_{\widehat{R}_t}(x_t) + \mathbb{E}_{a \sim \pi_t(\cdot|x_t)} \left[ \widehat{R}_t(x_t, a) - R^\star(x_t, a) \right]$$

$$= f(x_t, \widehat{R}_t) - f(x_t, R^\star).$$

In Zhao et al. (2025a), the decomposition takes the alternative form $J_t^\eta(\pi_\eta^\star, R^\star) - J_t^\eta(\pi_t, R^\star) = f(x_t, \tilde{R}_t) - f(x_t, R^\star)$, where $\tilde{R}_t(x, a) := \widehat{R}_t(x, a) + b_t(x, a)$ is the UCB. They then apply the mean value theorem to this expression and leverage optimism to bound $f(x_t, \tilde{R}_t) - f(x_t, R^\star)$.

In contrast, our analysis shows that it suffices to work directly with the oracle estimator $\widehat{R}_t$. Instead of invoking the mean value theorem, we use the exact *second-order Taylor expansion* of $f$.

$$f(x_t, \widehat{R}_t) - f(x_t, R^\star) = \sum_{a \in \mathcal{A}} \underbrace{\frac{\partial f(x_t, R^\star)}{\partial R(x_t, a)}}_{=0} \Delta R_t(x, a)$$

$$+ \int_0^1 (1 - \alpha) \left[ \sum_{a \in \mathcal{A}} \sum_{a' \in \mathcal{A}} \Delta R_t(x, a) \frac{\partial^2 f(x_t, R^\star + \alpha \Delta R_t)}{\partial R(x_t, a') \partial R(x_t, a)} \Delta R_t(x_t, a') \right] d\alpha$$

$$\leqslant \eta \mathbb{E}_{a \sim \pi_t(\cdot|x_t)} \left[ \left( \widehat{R}_t(x_t, a) - R^\star(x_t, a) \right)^2 \right], \tag{5}$$

where $\Delta R_t = \widehat{R}_t - R^\star$. Note that in the equation, $\frac{\partial f(x_t, R^\star)}{\partial R(x_t, a)} = 0$, which is one of our key theoretical findings. This result shows that it is unnecessary to rely on optimistic estimators such as UCB. The remaining steps then follow directly from straightforward calculus (see Lemma B.2 for details).

**2) Conversion to regression oracle bound.** By summing over $t \in [T]$ in Equation 5 and applying Freedman's inequality together with Lemma 4 of Foster & Rakhlin 2020, we obtain

$$\mathbf{Regret}_{\mathrm{KL}}(T, \eta) \leqslant \eta \sum_{t=1}^T \mathbb{E}_{a_t \sim \pi_t(\cdot|x_t)} \left[ \left( \widehat{R}_t(x_t, a_t) - R^\star(x_t, a_t) \right)^2 \right] \leqslant 2\eta \mathrm{Reg}_{\mathrm{Sq}}(T) + 16\eta \log \frac{1}{\delta}.$$

This completes the proof of the KL-regularized regret bound.

**3) Unregularized regret bound.** From the definitions of $J_t^\eta$ and $\pi_\eta^\star$, together with the non-negativity of the KL divergence, we can bound the unregularized regret as follows (Lemma B.3):

$$\mathbf{Regret}(T) \leqslant \mathbf{Regret}_{\mathrm{KL}}(T, \eta) + \frac{1}{\eta} \sum_{t=1}^T \mathrm{KL}\left(\pi^\star(\cdot\|x_t)\|\pi_{\mathrm{ref}}(\cdot\|x_t)\right).$$

By applying the KL-regularized regret bound established above, we complete the proof of Theorem 1.

**Remark 4** (Intuition behind why KL-regularization is sufficient)**.** *KL-regularization keeps the policy close to a reference policy, and by choosing the regularization parameter $\eta$ appropriately, we can induce the right amount of exploration. When the optimal policy $\pi_\eta^\star$ is far from the reference policy $\pi_{\mathrm{ref}}$, we use a larger $\eta$ to encourage more aggressive exploration; when they are close, we use a smaller $\eta$ to induce more conservative exploration. For additional intuition, consider the special case where the reference policy is uniform random. In this setting, KL-regularization resembles*

*the entropic-regularized Follow-the-Regularized-Leader (FTRL) framework (Abernethy et al., 2009; Orabona, 2019) (even though the objectives[4] and analyses differ fundamentally). Both approaches introduce a regularizer when optimizing the policy, leading to a Gibbs-style solution. This connection illustrates how KL-regularization can induce an exploratory effect similar to that of FTRL, implicitly balancing exploration and exploitation through its regularized policy optimization.*

## 5 REINFORCEMENT LEARNING FROM HUMAN FEEDBACK

### 5.1 LOG-LOSS ONLINE REGRESSION ORACLE.

Similar to the KL-regularized contextual bandit setting, we assume access to a log-loss online regression oracle (Foster & Krishnamurthy, 2021), denoted by `OracleLog`. First, we define the binary logarithmic/cross-entropy loss function ("log-loss") at round $t$ as

$$\ell_t(R) := -\Big[ y_t \log \sigma\big(R(x_t, a_t^1) - R(x_t, a_t^2)\big) + (1 - y_t) \log \sigma\big(R(x_t, a_t^2) - R(x_t, a_t^1)\big) \Big], \quad (6)$$

where $y_t$ denote the binary preference label, where $y_t = 1$ if $a_t^1$ is preferred over $a_t^2$ (i.e., $a_t^1 \succ a_t^2$) and $y_t = 0$ otherwise At each round $t$, `OracleLog` returns

$$\widehat{R}_t \leftarrow \texttt{OracleLog}_t\big((x_1, a_1^1, a_1^2, y_1), \dots, (x_{t-1}, a_{t-1}^1, a_{t-1}^2, y_{t-1})\big), \quad \text{where } \widehat{R}_t \in \mathcal{R}. \quad (7)$$

Analogous to Assumption 3, we assume that the prediction error of `OracleLog` is bounded as follows:

**Assumption 4** (Guarantee of log-loss regression oracle)**.** *We assume that, for every (possibly adaptively chosen) sequence $x_{1:T}, a_{1:T}^1, a_{1:T}^2, y_{1:T}$, there exists regret bound $\mathrm{Reg}_{\mathrm{Log}}(T)$ such that the regression oracle `OracleLog` satisfies*

$$\sum_{t=1}^T \ell_t(\widehat{R}_t) - \sum_{t=1}^T \ell_t(R^\star) \leqslant \mathrm{Reg}_{\mathrm{Log}}(T).$$

**Example 4** (Linear classes under log-loss)**.** *When $R^\star \in \mathcal{R}$ and the reward function class $\mathcal{R}$ is linear, we can use the algorithm from Foster et al. (2018b) to obtain $\mathrm{Reg}_{\mathrm{Log}}(T) = \mathcal{O}(d \log(T/d))$.*

Similar guarantees are available for kernels, generalized linear models, and many other nonparametric classes, as in the case of the squared-loss online regression oracle (Foster & Krishnamurthy, 2021).

### 5.2 ALGORITHM AND MAIN RESULTS

We now introduce an algorithm for RLHF problems, `OEPO`, described in Algorithm D.1. The overall flow is similar to `KL-EXP`; however, at each round $t \in [T]$, the current policy samples two actions, $a_t^1, a_t^2 \sim \pi_t(\cdot|x_t)$, and receives preference feedback between them. Another key difference is that the reward estimator $\widehat{R}_{t+1}$ is updated using the log-loss online regression oracle `OracleLog` (Equation 7). When `OracleLog` is implemented with a gradient-based method (e.g., SGD or Adam), `OEPO` recovers the practical online RLHF algorithm.

The regret guarantees for `OEPO` are presented below, with the proofs deferred to Appendix D.

**Theorem 3** (Regret of `OEPO`)**.** *Let $\delta > 0$, $D := \frac{1}{T}\sum_{t=1}^T \mathrm{KL}\big(\pi^\star(\cdot\|x_t)\|\pi_{\mathrm{ref}}(\cdot\|x_t)\big)$ and $\kappa := \sup_{R,x,a} 1/\dot{\sigma}(R(x,a))$. Under Assumption 1, 2, and 4, with probability at least $1 - \delta$, `OEPO` ensures*

$$\mathbf{Regret}_{\mathrm{KL}}(T, \eta) = \mathcal{O}\Big(\eta\kappa^2 \mathrm{Reg}_{\mathrm{Log}}(T) + \eta\kappa^2 \log(1/\delta)\Big) \quad \text{and}$$

$$\mathbf{Regret}(T) = \mathcal{O}\Big(\eta\kappa^2 \mathrm{Reg}_{\mathrm{Log}}(T) + \eta\kappa^2 \log(1/\delta) + \frac{DT}{\eta}\Big).$$

**Discussion of Theorem 3.** We obtain regret bounds comparable to Theorem 1, up to a $\kappa$ factor (and differences in oracle prediction error). Such $\kappa$-dependence is standard and largely unavoidable

---

[4]FTRL optimizes an objective based on cumulative losses, while KL-regularization optimizes one based on current reward estimates.

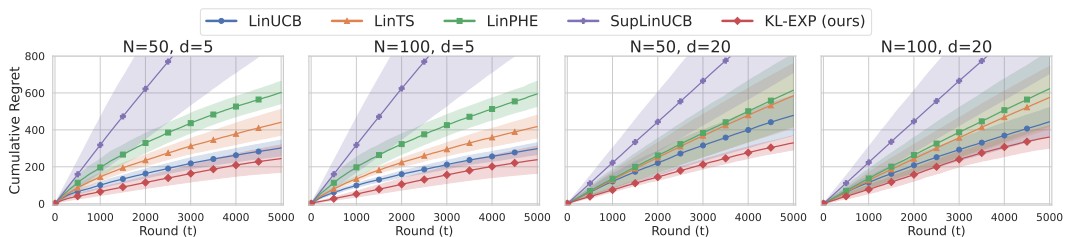

Figure 1: Cumulative regret in linear bandits with $d \in \{5, 20\}$ and $N = |\mathcal{A}| \in \{50, 100\}$.

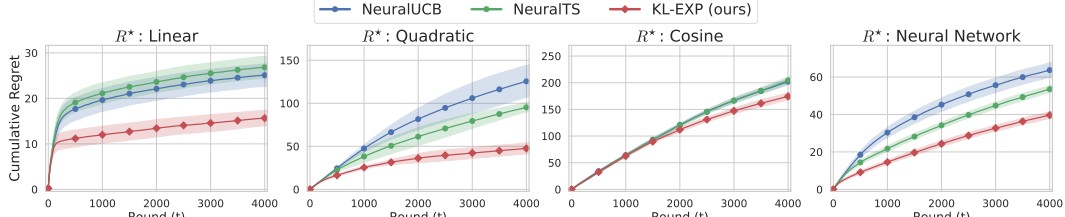

Figure 2: Cumulative regret in neural bandits under different true reward functions.

in RLHF and dueling bandits (Saha, 2021; Saha et al., 2023; Zhu et al., 2023; Xiong et al., 2023; Zhan et al., 2023b; Das et al., 2024; Xie et al., 2024; Zhao et al., 2024). With the choices $\eta = \Theta\big(\sqrt{DT/(\kappa^2 \mathrm{Reg}_{\mathrm{Log}}(T))}\big)$ and $\pi_{\mathrm{ref}} = \mathrm{Unif}(\mathcal{A})$, OEPO achieves unregularized regret $\mathbf{Regret}(T) = \mathcal{O}(\kappa\sqrt{DT\mathrm{Reg}_{\mathrm{Log}}(T)})$. As in Theorem 1, this yields $\tilde{\mathcal{O}}(\sqrt{T})$ regret guarantees for a broad range of function classes (see Foster & Krishnamurthy (2021) for bound on $\mathrm{Reg}_{\mathrm{Log}}(T)$).

**Remark 5** (Extension to DPO, Rafailov et al., 2023). *The DPO-variant algorithm (Algorithm D.2) achieves the same-order regrets, up to differences in the oracle's prediction error (see Appendix E).*

## 6 EXPERIMENTS

### 6.1 LINEAR CONTEXTUAL BANDITS

In the linear bandit experiments, we consider linear reward function class, i.e., $\mathcal{R} = \{R : R = \phi(x,a)^\top\theta, \theta \in \mathbb{R}^d, \|\theta\|_2 \leq 1\}$. For each instance we sample the true parameter $\theta^\star \sim \mathcal{N}(0, I_d)$ and normalize it so that $\|\theta^\star\|_2 \leq 1$. At each round $t$, a context $x_t \in \mathcal{X}$ is drawn uniformly at random, with feature vector $\phi(x_t, a) \in \mathbb{R}^d$ lying in the unit ball. We set $d \in \{5, 20\}$ and $N = |\mathcal{A}| \in \{50, 100\}$. We report cumulative regret averaged over 20 runs, with standard errors.

We compare the performance of our algorithm KL-EXP against four baselines: (i) LinUCB (Li et al., 2010), (ii) LinTS (Agrawal & Goyal, 2013), (iii) LinPHE (Kveton et al., 2020), and (iv) SupLinUCB (Chu et al., 2011). We use the exact theoretical confidence parameters for the baselines and the theoretically optimal regularization parameter $\eta$ from Theorem 1 for our algorithm. Figure 1 shows that our algorithm consistently and significantly outperforms the baselines across varying $d$ and $N$, while also achieving faster per-round computation than the others (see Table H.1).

### 6.2 NEURAL CONTEXTUAL BANDITS

In the neural bandit experiments, we use the neural network reward class $\mathcal{R}$, instantiated as a two-layer network with input dimension 80 and hidden width 100, equipped with ReLU activations. We evaluate four types of true reward functions: (i) linear: $R^\star(x,a) = \phi(x,a)^\top\theta^\star$, (ii) quadratic: $R^\star(x,a) = (\phi(x,a)^\top\theta^\star)^2$, (iii) cosine: $R^\star(x,a) = \cos(\pi\phi(x,a)^\top\theta^\star)$, and (iv) neural network: $R^\star \in \mathcal{R}$. Training is performed with squared loss via SGD (batch size 100, learning rate 0.005). We set $N = 20$, and report cumulative regret averaged over 10 runs with standard errors.

We compare our algorithm KL-EXP against two baselines: (i) NeuralUCB (Zhou et al., 2020) and (ii) NeuralTS (Zhang et al., 2020). For the baselines, we tune the confidence bounds via grid search

| | Llama-3-8B-Flow -SFT | Llama-3-8B-Flow -Final | XPO | OnlineDPO ($\eta$) | | | | |
| --- | --- | --- | --- | --- | --- | --- | --- | --- |
| | | | | 5.0 | 8.5 | 10.0 | 12.5 | 20.0 |
| Accuracy (%) | 59.11 | 60.47 | 61.61 ± 0.04 | 61.90 ± 0.07 | 62.04 ± 0.14 | 62.00 ± 0.11 | **62.14** ± 0.12 | 62.02 ± 0.32 |

Table 1: OnlineDPO and XPO are trained with three random seeds; we report the mean accuracy over 17 benchmarks and one standard error (small font), capturing training variance. Llama-3-8B-Flow-SFT and -Final are fixed pretrained models and thus have no training randomness.

over $\{1.0, 5.0, 10.0\}$. For `KL-EXP`, we tune $\eta$ using grid search over $\{50, 100, 500\}$, and adopt the uniform random reference policy. Figure 2 shows that our algorithm outperforms the baselines across diverse reward structures while running about $10\times$ faster (see Table H.3).

### 6.3 LLM FINE-TUNING WITH RLHF

In this subsection, we validate our key theoretical insight in the LLM fine-tuning task: *properly tuning the regularization parameter $\eta$ alone is sufficient to induce exploration.* Our DPO-variant algorithm, `ODPO`, coincides with `OnlineDPO` (Guo et al., 2024) when the regression oracle `OracleDPO` (defined in Equation E.2) is instantiated using the original DPO optimizer settings (optimizer, batch size, learning rate, and training steps). Since we adopt these original settings, we report the algorithm as `OnlineDPO` (in Table 1) rather than `ODPO`, to avoid confusion.

For experimental details, we follow the iterative `DPO` pipeline (Xu et al., 2023; Tran et al., 2023; Dong et al., 2024; Xie et al., 2024) from Dong et al. (2024), running $T = 3$ total iterations with large batches of pairs sampled from $\pi_t$. We use the same base model (Llama-3-8B-Flow-SFT[5]), prompt sets for each iteration[6], and true preference model for generating feedback[7] as in Dong et al. (2024); Xie et al. (2024), ensuring our results are directly comparable to theirs. Across all three iterations, we fix the reference policy $\pi_{\text{ref}}$ to the base model Llama-3-8B-Flow-SFT.

We consider three baselines: (i) Llama-3-8B-Flow-SFT, the reference model; (ii) Llama-3-8B-Flow-Final, the final model from Dong et al. (2024), released on Hugging Face[8]; and (iii) XPO (Xie et al., 2024). To induce exploration, Llama-3-8B-Flow-Final constructs preference pairs by maximizing heuristic uncertainty, while XPO augments the DPO objective with an additional exploration term that encourages the policy to behave optimistically. We evaluate all algorithms on 17 academic and chat benchmarks (Zhong et al., 2023; Nie et al., 2019; Hendrycks et al., 2020; Cobbe et al., 2021; Rein et al., 2024; Chen et al., 2021; Zellers et al., 2019; Sakaguchi et al., 2021; Clark et al., 2018; Lin et al., 2021; Mihaylov et al., 2018; Zellers et al., 2018; Sap et al., 2019; Pilehvar & Camacho-Collados, 2018; Levesque et al., 2012; Socher et al., 2013) and report their average accuracies. Table 1 shows that with a properly chosen $\eta = 12.5$, `OnlineDPO` (or `ODPO`) outperforms other baseline algorithms that rely on auxiliary exploration methods. This supports our main theoretical claim that additional exploration techniques are unnecessary in online RLHF—properly tuning $\eta$ suffices. See Appendix H.3 for additional experimental details, per-benchmark results, training-time accuracy, and further analysis.

## 7 CONCLUSION

We show, for the first time to our knowledge, that KL-regularization alone is sufficient for achieving sublinear regrets. In particular, the KL-regularized regret scales with the regression oracle bound, which can be logarithmic in $T$ for many function classes. Moreover, by carefully choosing the regularization parameter $\eta$, we achieve $\tilde{\mathcal{O}}(\sqrt{T})$ unregularized regret, demonstrating that the policy can be improved beyond the KL-regularized optimum. This highlights the pivotal role of $\eta$ in attaining sublinear unregularized regret. We leave further refinements of $\eta$, such as time-varying schedules, as an important direction for future work.

---

[5]https://huggingface.co/RLHFlow/LLaMA3-SFT

[6]https://huggingface.co/datasets/RLHFlow/iterative-prompt-v1-iter2-20K

[7]https://huggingface.co/RLHFlow/pair-preference-model-LLaMA3-8B

[8]https://huggingface.co/RLHFlow/LLaMA3-iterative-DPO-final

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

## THE USE OF LARGE LANGUAGE MODELS

Large language models (LLMs) were used solely as an assistive tool for non-substantive tasks in preparing this paper. Their use was limited to improving clarity, grammar, and style, as well as helping generate code snippets for figures and visualizations, which were subsequently verified and customized by the authors. No part of the research ideation, algorithm design, theoretical analysis, or experimental results involved the use of LLMs. The authors take full responsibility for the entire content of the paper, and LLMs are not considered authors or contributors.

# Appendix

## Table of Contents

## A  FURTHER RELATED WORK

In this section, we provide additional related work that complements Section 2.

**Dueling bandits.** The dueling bandit problem, first introduced by Yue et al. (2012), generalizes the classical multi-armed bandit by replacing direct reward observations with pairwise comparisons: in each round $t$, the learner chooses two arms and only observes which one is preferred. A challenge in this setting is that there may not exist a single arm that dominates all others under arbitrary preference structures. To deal with this, the literature has proposed several notions of "winners," such as the Condorcet winner (Zoghi et al., 2014; Komiyama et al., 2015), Copeland winner (Zoghi et al., 2015; Wu & Liu, 2016; Komiyama et al., 2016), Borda winner (Jamieson et al., 2015; Falahatgar et al., 2017; Heckel et al., 2018; Saha et al., 2021; Wu et al., 2023), and von Neumann winner (Ramamohan et al., 2016; Dudík et al., 2015; Balsubramani et al., 2016), each of which comes with its own performance criterion.

To incorporate contextual information, Saha (2021) introduced the contextual dueling bandit with a Bradley–Terry–Luce (BTL) model (Bradley & Terry, 1952), where pairwise preferences are determined by latent arm rewards. Building on this line, Bengs et al. (2022) analyzed a contextual linear stochastic transitivity model, and Di et al. (2023) proposed a layered algorithm with variance-sensitive regret guarantees.

Another line of research avoids parametric reward models and instead assumes that preferences are generated by a more general function class. For instance, Saha & Krishnamurthy (2022) developed an algorithm with optimal regret guarantees for $K$-armed contextual dueling bandits, and Sekhari et al. (2023) further extended the framework with algorithms that provide theoretical guarantees not only on regret but also on query complexity.

However, existing dueling bandit frameworks do not consider the KL-regularized objective, which is the main focus of our work.

**RLHF theory.** Motivated by the remarkable success of RLHF in fine-tuning LLMs, its theoretical foundations have recently become an active research topic. Much of the existing work focuses on the offline RLHF setting (Zhu et al., 2023; Zhan et al., 2023a), which is complementary to ours. Another line of research studies hybrid RLHF, where offline data are incorporated into an online RL procedure (Xiong et al., 2023; Gao et al., 2024; Chang et al., 2024).

In the context of online RLHF, much of the prior work (Xu et al., 2020; Novoseller et al., 2020; Saha et al., 2023; Xiong et al., 2023; Wu & Sun, 2023) has focused on the special case of tabular MDPs or linear MDPs (or linear reward models when the horizon length is 1), establishing sample complexity or regret bounds in this setting. The exploration bonuses used in these algorithms are specifically designed for linear structures and thus do not extend naturally to the more general function approximation regime we study (e.g., for LLMs).

To go beyond linear models, Chen et al. (2022); Wang et al. (2023); Ye et al. (2024) investigate general function approximation under the assumption of prior knowledge of the eluder dimension (Russo & Van Roy, 2013), which is notoriously difficult to quantify in practice, especially for LLMs. More recently, Zhao et al. (2025a) leveraged the properties of KL-regularization to establish the first $\mathcal{O}(\eta \log T)$ KL-regularized regret bound, again assuming prior knowledge of the eluder dimension. These approaches also require solving a complex optimization problem to compute the exploration terms, raising concerns about their practicality for large-scale language models. In parallel, Zhao et al. (2024) achieved a $\mathcal{O}(\eta/\epsilon)$ KL-regularized suboptimality gap by relying on a forced exploration phase, whose length depends on the coverage coefficient—another quantity that is difficult to determine in practice. As yet another direction, Zhao et al. (2025b) analyze $f$-divergence-regularized offline policy learning.

To improve practicality under general function approximation, Xie et al. (2024); Liu et al. (2024); Cen et al. (2024) proposed value-incentivized exploration methods that optimize the policy against optimistically biased targets. However, the optimization problems in these approaches do not admit closed-form solutions, and they introduce an additional exploration parameter $\alpha$ that must be tuned, which can make implementation sensitive to hyperparameter choices.

To the best of our knowledge, all existing online RLHF works rely on auxiliary exploration methods beyond KL-regularization. In contrast, our algorithm KL-EXP relies solely on KL-regularization. Moreover, it requires no prior knowledge of any complexity measure, admits a closed-form solution Equation 2, and is thus easy to implement.

## B    PROOF OF THEOREM 1

In this section, we present the proof of Theorem 1.

### B.1    MAIN PROOF OF THEOREM 1

Define $M_t := (\widehat{R}_t(x_t, a_t) - r_t)^2 - (R^\star(x_t, a_t) - r_t)^2$ and $Z_t := \mathbb{E}[M_t \mid \mathcal{F}_{t-1}] - M_t$, where $\mathcal{F}_{t-1} = \sigma(x_1, a_1, r_1, \ldots, x_{t-1}, a_{t-1}, r_{t-1}, x_t)$ is the filtration up to round $t - 1$. The following lemma establishes that these random variables are both bounded and self-bounding.

**Lemma B.1** (Lemma 4 of Foster & Rakhlin 2020). *Let $\mathcal{F}_{t-1}$ be the filtration up to round $t - 1$, i.e.,* $\mathcal{F}_{t-1} = \sigma(x_1, a_1, r_1, \ldots, x_{t-1}, a_{t-1}, r_{t-1}, x_t)$. *Define* $M_t := (\widehat{R}_t(x_t, a_t) - r_t)^2 - (R^\star(x_t, a_t) - r_t)^2$ *and* $Z_t := \mathbb{E}[M_t \mid \mathcal{F}_{t-1}] - M_t$. *Then, the following properties hold:*

- $|Z_t| \leqslant 1$.

- $\mathbb{E}[M_t \mid \mathcal{F}_{t-1}] = \mathbb{E}_{a \sim \pi_t(\cdot|x_t)} \left[ (\widehat{R}_t(x_t, a_t) - R^\star(x_t, a_t))^2 \right].$

- $\mathbb{E}[Z_t^2 \mid \mathcal{F}_{t-1}] \leqslant 4\mathbb{E}[M_t \mid \mathcal{F}_{t-1}].$

We now present a key lemma that is central to the proof of Theorem 1 and crucial for establishing regret guarantees *without any additional exploration*.

**Lemma B.2** (Second-order regret decomposition). *Under Assumption 1 and 2, for any $t \in [T]$, we have*

$$J_t^\eta(\pi_\eta^\star, R^\star) - J_t^\eta(\pi_t, R^\star) \leqslant \eta \mathbb{E}_{a \sim \pi_t(\cdot|x_t)} \left[ \left( \widehat{R}_t(x_t, a) - R^\star(x_t, a) \right)^2 \right].$$

The proof is deferred to Appendix B.2.1.

**Remark B.1** (Comparison with Zhao et al. (2024)). *Unlike Lemma 3.9 of Zhao et al. (2024), which bounds the regret $J_t^\eta(\pi_\eta^\star, R^\star) - J_t^\eta(\pi_t, R^\star)$ in terms of the unknown policy $\pi_{f_\gamma}^\eta$ (where $f_\gamma = \gamma \widehat{R}_t + (1 - \gamma)R^\star$ for some unknown $\gamma \in (0, 1)$), Lemma B.2 shows that our regret bound depends only on the known current policy $\pi_t$. Note that in Zhao et al. (2024), handling the unknown policy $\pi_{f_\gamma}^\eta$ requires a forced sampling phase, and the minimum number of forced sampling rounds depends on difficult-to-estimate quantities such as the data coverage coefficient (Definition 4.5 therein) and the $\epsilon$-covering number of the reward function class. In contrast, our algorithm does not rely on such quantities.*

**Remark B.2** (Comparison with Zhao et al. (2025a)). *Unlike Lemma A.1 of Zhao et al. (2025a), Lemma B.2 does not rely on the optimism event. Consequently, our algorithm does not require computing the Upper Confidence Bound (UCB) term, which is generally intractable for general function classes.*

**Lemma B.3** (Unregularized regret decomposition). *For any $t \in [T]$, we have*

$$\mathbb{E}_{a \sim \pi^\star(\cdot|x_t)}[R^\star(x_t, a)] - \mathbb{E}_{a \sim \pi_t(\cdot|x_t)}[R^\star(x_t, a)]$$

$$\leqslant J_t^\eta(\pi_\eta^\star, R^\star) - J_t^\eta(\pi_t, R^\star) + \frac{1}{\eta} \mathrm{KL}\left(\pi^\star(\cdot\|x_t)\|\pi_{\mathrm{ref}}(\cdot\|x_t)\right).$$

The proof is deferred to Appendix B.2.2.

We are now ready to provide the proof of Theorem 1.

*Proof of Theorem 1.* By Lemma B.2, we can bound the regret as follows:

$$\mathbf{Regret}_{\mathrm{KL}}(T, \eta) = \sum_{t=1}^T J_t^\eta(\pi_\eta^\star, R^\star) - J_t^\eta(\pi_t, R^\star)$$

$$\leqslant \eta \sum_{t=1}^T \mathbb{E}_{a_t \sim \pi_t(\cdot|x_t)} \left[ \left( \widehat{R}_t(x_t, a_t) - R^\star(x_t, a_t) \right)^2 \right]. \tag{B.1}$$

Let $\mathcal{F}_{t-1} = \sigma(x_1, a_1, r_1, \ldots, x_{t-1}, a_{t-1}, r_{t-1}, x_t)$ be the filtration up to round $t - 1$. Define $M_t := (\widehat{R}_t(x_t, a_t) - r_t)^2 - (R^\star(x_t, a_t) - r_t)^2$ and $Z_t := \mathbb{E}[M_t \mid \mathcal{F}_{t-1}] - M_t$. Then, by applying Freedman's inequality (Lemma G.1) with $\beta = 1/8$, with probability at least $1 - \delta$, we have

$$\sum_{t=1}^T \mathbb{E}[M_t \mid \mathcal{F}_{t-1}] \leqslant \sum_{t=1}^T M_t + \frac{1}{8} \sum_{t=1}^T \mathbb{E}[Z_t^2 \mid \mathcal{F}_{t-1}] + 8 \log \frac{1}{\delta}$$

$$= \sum_{t=1}^T M_t + \frac{1}{2} \sum_{t=1}^T \mathbb{E}[M_t \mid \mathcal{F}_{t-1}] + 8 \log \frac{1}{\delta} \qquad \text{(Lemma B.1)}$$

$$\leqslant \mathrm{Reg}_{\mathrm{Sq}}(T) + \frac{1}{2} \sum_{t=1}^T \mathbb{E}[M_t \mid \mathcal{F}_{t-1}] + 8 \log \frac{1}{\delta},$$

where the last inequality holds because

$$\sum_{t=1}^{T} M_t = \sum_{t=1}^{T}(\widehat{R}_t(x_t, a_t) - r_t)^2 - \sum_{t=1}^{T}(R^\star(x_t, a_t) - r_t)^2 \leqslant \text{Reg}_{\text{Sq}}(T). \qquad \text{(Assumption 3)}$$

This directly implies

$$\sum_{t=1}^{T} \mathbb{E}[M_t \mid \mathcal{F}_{t-1}] \leqslant 2\text{Reg}_{\text{Sq}}(T) + 16 \log \frac{1}{\delta}. \qquad \text{(B.2)}$$

Plugging Equation B.2 into Equation B.1, we obtain

$$\textbf{Regret}_{\text{KL}}(T, \eta) \leqslant \eta \sum_{t=1}^{T} \mathbb{E}_{a_t \sim \pi_t(\cdot|x_t)} \left[ \left( \widehat{R}_t(x_t, a_t) - R^\star(x_t, a_t) \right)^2 \right]$$

$$= \eta \sum_{t=1}^{T} \mathbb{E}[M_t \mid \mathcal{F}_{t-1}] \qquad \text{(Lemma B.1)}$$

$$\leqslant 2\eta\text{Reg}_{\text{Sq}}(T) + 16\eta \log \frac{1}{\delta}. \qquad \text{(Equation B.2)}$$

This concludes the proof of the regret bound for the KL-regularized objective.

We now provide the proof of the unregularized regret bound. By summing over $t \in [T]$ on both sides of the result in Lemma B.3, we directly obtain

$$\textbf{Regret}(T) \leqslant \sum_{t=1}^{T} \left( J_t^\eta(\pi_\eta^\star, R^\star) - J_t^\eta(\pi_t, R^\star) \right) + \frac{1}{\eta} \sum_{t=1}^{T} \text{KL}\left( \pi^\star(\cdot\|x_t)\|\pi_{\text{ref}}(\cdot\|x_t) \right)$$

$$= \textbf{Regret}_{\text{KL}}(T, \eta) + \frac{1}{\eta} \sum_{t=1}^{T} \text{KL}\left( \pi^\star(\cdot\|x_t)\|\pi_{\text{ref}}(\cdot\|x_t) \right) \quad \text{(Definition of } \textbf{Regret}_{\text{KL}}(T, \eta))$$

$$= \textbf{Regret}_{\text{KL}}(T, \eta) + \frac{DT}{\eta} \qquad (D := \tfrac{1}{T}\sum_{t=1}^{T} \text{KL}\left( \pi^\star(\cdot\|x_t)\|\pi_{\text{ref}}(\cdot\|x_t) \right))$$

$$= \mathcal{O}\left( \eta\text{Reg}_{\text{Sq}}(T) + \eta \log(1/\delta) + \frac{DT}{\eta} \right).$$

Hence, the proof of Theorem 1 is complete. $\qquad \square$

## B.2 PROOFS OF LEMMAS FOR THEOREM 1

### B.2.1 PROOF OF LEMMA B.2

*Proof of Lemma B.2.* For simplicity, we use the shorthand $\mathbb{E}_\pi[\cdot] = \mathbb{E}_{a \sim \pi(\cdot|x)}[\cdot]$. Noting that $R^\star(x, a) = \frac{1}{\eta} \log \exp (\eta R^\star(x, a))$, we have

$$\mathbb{E}_{\pi_\eta^\star}\left[ R^\star(x, a) - \frac{1}{\eta} \log \frac{\pi_\eta^\star(a|x)}{\pi_{\text{ref}}(a|x)} \right] - \mathbb{E}_{\pi_t}\left[ R^\star(x, a) - \frac{1}{\eta} \log \frac{\pi_t(a|x)}{\pi_{\text{ref}}(a|x)} \right]$$

$$= \frac{1}{\eta}\mathbb{E}_{\pi_\eta^\star}\left[ \log \frac{\pi_{\text{ref}}(a|x) \cdot \exp (\eta R^\star(x, a))}{\pi_\eta^\star(a|x)} \right] - \frac{1}{\eta}\mathbb{E}_{\pi_t}\left[ \log \frac{\pi_{\text{ref}}(a|x) \cdot \exp (\eta R^\star(x, a))}{\pi_t(a|x)} \right]$$

$$= \frac{1}{\eta}\mathbb{E}_{\pi_\eta^\star}\left[ \log \frac{\pi_{\text{ref}}(a|x) \cdot \exp (\eta R^\star(x, a))}{\pi_\eta^\star(a|x)} \right] - \frac{1}{\eta}\mathbb{E}_{\pi_t}\left[ \log \frac{\pi_{\text{ref}}(a|x) \cdot \exp \left( \eta \widehat{R}(x, a) \right)}{\pi_t(a|x)} \right]$$

$$+ \mathbb{E}_{\pi_t}\left[ \widehat{R}_t(x, a) - R^\star(x, a) \right]$$

$$= \frac{1}{\eta} \log Z_{R^\star}(x) - \frac{1}{\eta} \log Z_{\widehat{R}_t}(x) + \mathbb{E}_{\pi_t}\left[ \widehat{R}_t(x, a) - R^\star(x, a) \right], \qquad \text{(B.3)}$$

where the last equality holds because

$$\frac{\pi_{\text{ref}}(a|x) \cdot \exp\left(\eta R^\star(x,a)\right)}{\pi_\eta^\star(a|x)} = \frac{\pi_{\text{ref}}(a|x) \cdot \exp\left(\eta R^\star(x,a)\right)}{\pi_{\text{ref}}(a|x) \cdot \exp\left(\eta R^\star(x,a)\right)/Z_{R^\star}(x)} = Z_{R^\star}(x),$$

and

$$\frac{\pi_{\text{ref}}(a|x) \cdot \exp\left(\eta R^\star(x,a)\right)}{\pi_t(a|x)} = \frac{\pi_{\text{ref}}(a|x) \cdot \exp\left(\eta \widehat{R}_t(x,a)\right)}{\pi_{\text{ref}}(a|x) \cdot \exp\left(\eta \widehat{R}_t(x,a)\right)/Z_{\widehat{R}_t}(x)} = Z_{\widehat{R}_t}(x),$$

Define the function $f : \mathcal{X} \times \mathcal{R} \to \mathbb{R}$ as follows:

$$f(x, R) := -\frac{1}{\eta} \log Z_R(x) + \sum_{a \in \mathcal{A}} \underbrace{\frac{\pi_{\text{ref}}(a|x) \cdot \exp\left(\eta R(x,a)\right)}{Z_R(x)}}_{=\pi_R^\eta(a|x)} \cdot (R(x,a) - R^\star(x,a))$$

$$= -\frac{1}{\eta} \log Z_R(x) + \mathbb{E}_{\pi_R^\eta}\left[R(x,a) - R^\star(x,a)\right]. \tag{B.4}$$

Then, since $\pi_t = \pi_{\widehat{R}_t}^\eta$, the right-hand side of Equation B.3 can be written as:

$$\frac{1}{\eta} \log Z_{R^\star}(x) - \frac{1}{\eta} \log Z_{\widehat{R}_t}(x) + \mathbb{E}_{\pi_t}\left[\widehat{R}_t(x,a) - R^\star(x,a)\right] = f(x, \widehat{R}_t) - f(x, R^\star).$$

First, we present the lemma that gives the derivatives of $\pi_R^\eta$ and $Z_R$, with the proof given in Appendix B.3.1.

**Lemma B.4.** *Under Assumption 2, for any $(x, a) \in \mathcal{X} \times \mathcal{A}$, we have*

$$\frac{\partial Z_R(x)}{\partial R(x,a)} = \eta \pi_{\text{ref}}(a|x) \exp(\eta R(x,a)),$$

$$\frac{\partial \pi_R^\eta(a'|x)}{\partial R(x,a)} = \begin{cases} \eta \pi_R^\eta(a|x) - \eta \pi_R^\eta(a|x)^2, & \text{if } a = a', \\ -\eta \pi_R^\eta(a'|x)\pi_R^\eta(a|x), & \text{if } a \neq a'. \end{cases}$$

$$\frac{\partial \mu_R(x)}{\partial R(x,a)} = \eta \pi_R^\eta(a|x)\left(R(x,a) - R^\star(x,a) - \mu_R(x)\right) + \pi_R^\eta(a|x),$$

*where $\mu_R(x) := \mathbb{E}_{a \sim \pi_R^\eta(\cdot|x)}\left[R(x,a) - R^\star(x,a)\right]$.*

Then, we compute the derivative of $f(x, R)$ as follows:

$$\frac{\partial f(x, R)}{\partial R(x,a)} = -\frac{1}{\eta} \frac{\partial}{\partial R(x,a)} \log Z_R(x) + \frac{\partial}{\partial R(x,a)} \mathbb{E}_{\pi_R^\eta}\left[R(x,a) - R^\star(x,a)\right]$$

$$= -\frac{1}{\eta} \frac{1}{Z_R(x)} \frac{\partial Z_R(x)}{\partial R(x,a)} + \frac{\partial}{\partial R(x,a)}\left[\pi_R^\eta(a|x) \cdot \left(R(x,a) - R^\star(x,a)\right)\right]$$

$$\quad + \frac{\partial}{\partial R(x,a)}\left[\sum_{a' \neq a} \pi_R^\eta(a'|x) \cdot \left(R(x,a') - R^\star(x,a')\right)\right]$$

$$= -\pi_R^\eta(a|x) + \pi_R^\eta(a|x) + \frac{\partial \pi_R^\eta(a|x)}{\partial R(x,a)} \cdot \left(R(x,a) - R^\star(x,a)\right)$$

$$\quad + \sum_{a' \neq a} \frac{\partial \pi_R^\eta(a'|x)}{\partial R(x,a)} \cdot \left(R(x,a') - R^\star(x,a')\right) \tag{Lemma B.4}$$

$$= \eta \pi_R^\eta(a|x) \cdot \left(R(x,a) - R^\star(x,a) - \mathbb{E}_{a'' \sim \pi_R^\eta(\cdot|x)}\left[R(x,a'') - R^\star(x,a'')\right]\right) \tag{Lemma B.4}$$

$$= \eta \pi_R^\eta(a|x) \cdot \left(R(x,a) - R^\star(x,a) - \mu_R(x)\right),$$

where $\mu_R(x) := \mathbb{E}_{a'' \sim \pi_R^\eta(\cdot|x)}\left[R(x,a'') - R^\star(x,a'')\right]$. Note that when $R = R^\star$, we have $\mu_{R^\star}(x) = 0$, which implies

$$\frac{\partial f(x, R^\star)}{\partial R(x,a)} = 0.$$

Moreover, the second-order gradient of $f$ can be expressed as:

$$\frac{\partial^2 f(x, R)}{\partial R(x, a')\partial R(x, a)}$$

$$= \frac{\partial}{\partial R(x, a')}\Big(\eta \pi_R^\eta(a|x) \cdot \big(R(x,a) - R^\star(x,a) - \mu_R(x)\big)\Big)$$

$$= \eta \frac{\partial \pi_R^\eta(a|x)}{\partial R(x,a')} \cdot \big(R(x,a) - R^\star(x,a) - \mu_R(x)\big) + \eta \pi_R^\eta(a|x) \cdot \left(\mathbf{1}_{a=a'} - \frac{\partial \mu_R(x)}{\partial R(x,a')}\right)$$

$$= \eta^2 \pi_R^\eta(a|x)\left(\mathbf{1}_{a=a'} - \pi_R^\eta(a'|x)\right)\big(R(x,a) - R^\star(x,a) - \mu_R(x)\big)$$

$$\quad + \eta \pi_R^\eta(a|x)\left(\mathbf{1}_{a=a'} - \eta \pi_R^\eta(x,a')\big(R(x,a') - R^\star(x,a') - \mu_R(x)\big) + \pi_R^\eta(x,a')\right)$$
$$\text{(Lemma B.4)}$$

$$= \eta \pi_R^\eta(a|x)\left(\mathbf{1}_{a=a'} - \pi_R^\eta(a'|x)\right)$$

$$\quad + \eta^2 \pi_R^\eta(a|x)\Big[\left(\mathbf{1}_{a=a'} - \pi_R^\eta(a'|x)\right)\left(R(x,a) - R^\star(x,a) - \mu_R(x)\right)$$

$$\quad\quad\quad - \pi_R^\eta(a'|x)\left(R(x,a') - R^\star(x,a') - \mu_R(x)\right)\Big].$$

For simplicity let $\Delta R_t = \widehat{R}_t - R^\star$ and $v_t^\alpha(x,a) = \alpha \Delta R_t(x,a) - \mu_{R^\star + \alpha \Delta R_t}(x) = \alpha \Delta R_t(x,a) - \alpha \mathbb{E}_{\pi_{R^\star + \alpha \Delta R_t}^\eta}[\Delta R_t(x,a'')]$. Then, using the exact second-order Taylor expansion, we have

$$f(x, \widehat{R}_t) - f(x, R^\star) = f(x, R^\star + \alpha \Delta R_t) - f(x, R^\star)$$

$$= \int_0^1 (1-\alpha)\left[\sum_{a\in\mathcal{A}}\sum_{a'\in\mathcal{A}} \Delta R_t(x,a) \frac{\partial^2 f(x, R^\star + \alpha \Delta R_t)}{\partial R(x,a')\partial R(x,a)} \Delta R_t(x,a')\right]d\alpha \qquad (\tfrac{\partial f(x,R^\star)}{\partial R(x,a)} = 0)$$

$$= \int_0^1 (1-\alpha)\Bigg[\eta \sum_{a\in\mathcal{A}} \pi_{R^\star + \alpha \Delta R_t}^\eta(a|x)\left(\Delta R_t(x,a)\right)^2 - \eta\left(\sum_{a\in\mathcal{A}} \pi_{R^\star + \alpha \Delta R_t}^\eta(a|x)\Delta R_t(x,a)\right)^2$$

$$\quad + \eta^2 \sum_{a\in\mathcal{A}} \pi_{R^\star + \alpha \Delta R_t}^\eta(a|x)v_t^\alpha(x,a)\left(\Delta R_t(x,a)\right)^2$$

$$\quad - 2\eta^2\left(\sum_{a\in\mathcal{A}} \pi_{R^\star + \alpha \Delta R_t}^\eta(a|x)v_t^\alpha(x,a)\Delta R_t(x,a)\right)\left(\sum_{a'\in\mathcal{A}} \pi_{R^\star + \alpha \Delta R_t}^\eta(a'|x)\Delta R_t(x,a')\right)\Bigg]d\alpha.$$
$$\text{(B.5)}$$

Plugging $v_t^\alpha(x,a) = \alpha \Delta R_t(x,a) - \alpha \mathbb{E}_{\pi_{R^\star + \alpha \Delta R_t}^\eta}[\Delta R_t(x,a'')]$ into the right-hand side, we can further simplify the second and third terms as follows:

$$\eta^2 \sum_{a\in\mathcal{A}} \pi_{R^\star + \alpha \Delta R_t}^\eta(a|x)v_t^\alpha(x,a)\left(\Delta R_t(x,a)\right)^2$$

$$- 2\eta^2\left(\sum_{a\in\mathcal{A}} \pi_{R^\star + \alpha \Delta R_t}^\eta(a|x)v_t^\alpha(x,a)\Delta R_t(x,a)\right)\left(\sum_{a'\in\mathcal{A}} \pi_{R^\star + \alpha \Delta R_t}^\eta(a'|x)\Delta R_t(x,a')\right)$$

$$= \eta^2 \alpha\Bigg[\sum_{a\in\mathcal{A}} \pi_{R^\star + \alpha \Delta R_t}^\eta(a|x)\left(\Delta R_t(x,a)\right)^3$$

$$\quad - 3\mathbb{E}_{\pi_{R^\star + \alpha \Delta R_t}^\eta}[\Delta R_t(x,a'')]\sum_{a\in\mathcal{A}} \pi_{R^\star + \alpha \Delta R_t}^\eta(a|x)\left(\Delta R_t(x,a)\right)^2 + 2\left(\mathbb{E}_{\pi_{R^\star + \alpha \Delta R_t}^\eta}[\Delta R_t(x,a'')]\right)^3\Bigg]$$
$$(\mathbb{E}[(X - \mathbb{E}[X])X] = \mathbb{E}[X^2] - (\mathbb{E}[X])^2)$$

$$= \eta^2 \alpha \sum_{a\in\mathcal{A}} \pi_{R^\star + \alpha \Delta R_t}^\eta(a|x)\left(\Delta R_t(x,a) - \mathbb{E}_{\pi_{R^\star + \alpha \Delta R_t}^\eta}[\Delta R_t(x,a'')]\right)^3.$$
$$(\mathbb{E}[(X - \mathbb{E}[X])^3] = \mathbb{E}[X^3] - 3\mathbb{E}[X]\mathbb{E}[X^2] + 2(\mathbb{E}[X])^3)$$

Using this, we can rewrite the right-hand side of Equation B.5 as follows:

$$f(x, \widehat{R}_t) - f(x, R^\star) = \int_0^1 (1-\alpha)\left[\eta \operatorname{Var}_t^\alpha(x) + \eta^2 \alpha M_t^\alpha(x)\right]d\alpha, \qquad (B.6)$$

where we define

$$
\mathrm{Var}_t^\alpha(x) := \sum_{a \in \mathcal{A}} \pi_{R^\star + \alpha \Delta R_t}^\eta(a|x) \left(\Delta R_t(x,a)\right)^2 - \left(\sum_{a \in \mathcal{A}} \pi_{R^\star + \alpha \Delta R_t}^\eta(a|x) \Delta R_t(x,a)\right)^2
$$

$$
M_t^\alpha(x) := \sum_{a \in \mathcal{A}} \pi_{R^\star + \alpha \Delta R_t}^\eta(a|x) \left(\Delta R_t(x,a) - \mathbb{E}_{\pi_{R^\star + \alpha \Delta R_t}^\eta}[\Delta R_t(x,a'')]\right)^3 .
$$

The following lemma is a useful tool for calculating the right-hand side of Equation B.6. Its proof is presented in Appendix B.3.2.

**Lemma B.5.** *Let* $\pi_\alpha(a|x) := \frac{\pi_{\mathrm{ref}}(a|x) \exp(\eta R_\alpha(x,a))}{Z_\alpha(x)}$, *where* $R_\alpha = R^\star + \alpha \Delta R$ *with* $R^\star, \Delta R \in \mathbb{R}$, *and* $Z_\alpha(x) = \sum_{a \in \mathcal{A}} \pi_{\mathrm{ref}}(a|x) \exp\left(\eta R_\alpha(x,a)\right)$. *Then, under Assumption 1 and 2, for any* $(x,a) \in \mathcal{X} \times \mathcal{A}$, *we have*

$$
\frac{\mathrm{d}}{\mathrm{d}\alpha} \pi_\alpha(a|x) = \eta \pi_\alpha(a|x) \left(\Delta R(x,a) - \mathbb{E}_{\pi_\alpha}[\Delta R(x,a)]\right),
$$

$$
\frac{\mathrm{d}}{\mathrm{d}\alpha} \mathbb{E}_{\pi_\alpha}[\Delta R(x,a)] = \eta \mathbb{E}_{\pi_\alpha}\left[(\Delta R(x,a) - \mathbb{E}_{\pi_\alpha}[\Delta R(x,a)])^2\right],
$$

$$
\frac{\mathrm{d}}{\mathrm{d}\alpha} \mathbb{E}_{\pi_\alpha}[\Delta R(x,a)^2] = \eta \left(\mathbb{E}_{\pi_\alpha}[\Delta R(x,a)^3] - \mathbb{E}_{\pi_\alpha}[\Delta R(x,a)^2] \mathbb{E}_{\pi_\alpha}[\Delta R(x,a)]\right).
$$

Then, by Lemma B.5, we show that

$$
\frac{\mathrm{d}}{\mathrm{d}\alpha} \mathrm{Var}_t^\alpha(x)
$$

$$
= \frac{\mathrm{d}}{\mathrm{d}\alpha} \left( \mathbb{E}_{\pi_{R^\star + \alpha \Delta R_t}^\eta}\left[(\Delta R_t(x,a))^2\right] - \left(\mathbb{E}_{\pi_{R^\star + \alpha \Delta R_t}^\eta}\left[\Delta R_t(x,a)\right]\right)^2 \right)
$$

$$
= \frac{\mathrm{d}}{\mathrm{d}\alpha} \mathbb{E}_{\pi_{R^\star + \alpha \Delta R_t}^\eta}\left[(\Delta R_t(x,a))^2\right] - 2\mathbb{E}_{\pi_{R^\star + \alpha \Delta R_t}^\eta}\left[\Delta R_t(x,a)\right] \cdot \frac{\mathrm{d}}{\mathrm{d}\alpha} \mathbb{E}_{\pi_{R^\star + \alpha \Delta R_t}^\eta}\left[\Delta R_t(x,a)\right]
$$

$$
= \eta \Bigg( \mathbb{E}_{\pi_{R^\star + \alpha \Delta R_t}^\eta}\left[(\Delta R_t(x,a))^3\right] - \mathbb{E}_{\pi_{R^\star + \alpha \Delta R_t}^\eta}\left[\Delta R_t(x,a)\right] \mathbb{E}_{\pi_{R^\star + \alpha \Delta R_t}^\eta}\left[(\Delta R_t(x,a))^2\right]
$$

$$
- 2\mathbb{E}_{\pi_{R^\star + \alpha \Delta R_t}^\eta}\left[\Delta R_t(x,a)\right] \cdot \mathrm{Var}_t^\alpha(x) \Bigg) \tag{Lemma B.5}
$$

$$
= \eta \Bigg( \mathbb{E}_{\pi_{R^\star + \alpha \Delta R_t}^\eta}\left[(\Delta R_t(x,a))^3\right] - 3\mathbb{E}_{\pi_{R^\star + \alpha \Delta R_t}^\eta}\left[\Delta R_t(x,a)\right] \mathbb{E}_{\pi_{R^\star + \alpha \Delta R_t}^\eta}\left[(\Delta R_t(x,a))^2\right]
$$

$$
+ 2 \left(\mathbb{E}_{\pi_{R^\star + \alpha \Delta R_t}^\eta}\left[\Delta R_t(x,a)\right]\right)^3 \Bigg) \tag{Definition of $\mathrm{Var}_t^\alpha(x)$}
$$

$$
= \eta M_t^\alpha(x). \tag{Definition of $M_t^\alpha(x)$}
$$

Therefore, Equation B.6 can be further simplified as:

$$
\begin{aligned}
f(x, \widehat{R}_t) - f(x, R^\star) &= \int_0^1 (1 - \alpha) \left[ \eta \operatorname{Var}_t^\alpha(x) + \eta^2 \alpha M_t^\alpha(x) \right] \mathrm{d}\alpha \\
&= \eta \left[ \int_0^1 (1 - \alpha) \operatorname{Var}_t^\alpha(x) \mathrm{d}\alpha + \int_0^1 \alpha \frac{\mathrm{d}}{\mathrm{d}\alpha} \operatorname{Var}_t^\alpha(x) \mathrm{d}\alpha \right] \\
&= \eta \left[ \int_0^1 (1 - \alpha) \operatorname{Var}_t^\alpha(x) \mathrm{d}\alpha + [\alpha \operatorname{Var}_t^\alpha(x)]_0^1 - \int_0^1 \operatorname{Var}_t^\alpha(x) \mathrm{d}\alpha \right] \\
&\qquad\qquad\qquad\qquad\qquad\qquad\qquad\qquad\qquad\qquad \text{(integration by parts)} \\
&= \eta \left[ \operatorname{Var}_t^{\alpha=1}(x) - \int_0^1 \alpha \operatorname{Var}_t^\alpha(x) \mathrm{d}\alpha \right] \\
&= \eta \mathbb{E}_{\pi_{\widehat{R}_t}^\eta} \left[ \left( \Delta R_t(x, a) - \mathbb{E}_{\pi_{\widehat{R}_t}^\eta} [\Delta R_t(x, a)] \right)^2 \right] - \eta \int_0^1 \alpha \operatorname{Var}_t^\alpha(x) \mathrm{d}\alpha \\
&\leqslant \eta \mathbb{E}_{\pi_{\widehat{R}_t}^\eta} \left[ \left( \Delta R_t(x, a) - \mathbb{E}_{\pi_{\widehat{R}_t}^\eta} [\Delta R_t(x, a)] \right)^2 \right] \qquad (\operatorname{Var}_t^\alpha(x) \geqslant 0) \\
&\leqslant \eta \mathbb{E}_{\pi_{\widehat{R}_t}^\eta} \left[ (\Delta R_t(x, a))^2 \right] \qquad\qquad (\mathbb{E}[(X - \mathbb{E}[X])^2] \leqslant \mathbb{E}[X^2])
\end{aligned}
$$

Recall that $\pi_t = \pi_{\widehat{R}_t}^\eta$ and $\Delta R_t = \widehat{R}_t - R^\star$. Hence, we obtain

$$
J_t^\eta(\pi_\eta^\star, R^\star) - J_t^\eta(\pi_t, R^\star) \leqslant \eta \mathbb{E}_{a \sim \pi_t(\cdot|x_t)} \left[ \left( \widehat{R}_t(x_t, a) - R^\star(x_t, a) \right)^2 \right].
$$

This concludes the proof of Lemma B.2. $\qquad\qquad\qquad\qquad\qquad\qquad\qquad\square$

### B.2.2 PROOF OF LEMMA B.3

*Proof of Lemma B.3.* For simple presentation, we write $\mathbb{E}_\pi[\cdot] = \mathbb{E}_{a \sim \pi(\cdot|x)}[\cdot]$. Then, for any $t \in [T]$, we have

$$
\begin{aligned}
\mathbb{E}_{a \sim \pi^\star(\cdot|x_t)}[R^\star(x_t, a)] &= J_t^\eta(\pi^\star, R^\star) + \frac{1}{\eta} \operatorname{KL}\left( \pi^\star(\cdot\|x_t) \| \pi_{\mathrm{ref}}(\cdot\|x_t) \right) \qquad \text{(Definition of } J_t^\eta) \\
&\leqslant J_t^\eta(\pi_\eta^\star, R^\star) + \frac{1}{\eta} \operatorname{KL}\left( \pi^\star(\cdot\|x_t) \| \pi_{\mathrm{ref}}(\cdot\|x_t) \right). \qquad \text{(Definition of } \pi_\eta^\star)
\end{aligned}
$$

Moreover, since the KL divergence is always non-negative, we get

$$
\begin{aligned}
\mathbb{E}_{a \sim \pi_t(\cdot|x_t)}[R^\star(x_t, a)] &\geqslant \mathbb{E}_{a \sim \pi_t(\cdot|x_t)}[R^\star(x_t, a)] - \frac{1}{\eta} \operatorname{KL}\left( \pi_t(\cdot\|x_t) \| \pi_{\mathrm{ref}}(\cdot\|x_t) \right) \\
&= J_t^\eta(\pi_t, R^\star).
\end{aligned}
$$

Combining the above two results, we obtain

$$
\begin{aligned}
&\mathbb{E}_{a \sim \pi^\star(\cdot|x_t)}[R^\star(x_t, a)] - \mathbb{E}_{a \sim \pi_t(\cdot|x_t)}[R^\star(x_t, a)] \\
&\qquad \leqslant J_t^\eta(\pi_\eta^\star, R^\star) - J_t^\eta(\pi_t, R^\star) + \frac{1}{\eta} \operatorname{KL}\left( \pi^\star(\cdot\|x_t) \| \pi_{\mathrm{ref}}(\cdot\|x_t) \right),
\end{aligned}
$$

which concludes the proof of Lemma B.3. $\qquad\qquad\qquad\qquad\qquad\qquad\qquad\square$

### B.3 SUPPORTING RESULTS FOR LEMMA B.2

#### B.3.1 PROOF OF LEMMA B.4

*Proof of Lemma B.4.* First, we compute the derivative of $Z_R(x)$. For any $(x, a) \in \mathcal{X} \times \mathcal{A}$, we get

$$
\frac{\partial Z_R(x)}{\partial R(x, a)} = \frac{\partial}{\partial R(x, a)} \left( \mathbb{E}_{\pi_{\mathrm{ref}}}[\exp(\eta R(x, a))] \right) = \eta \pi_{\mathrm{ref}}(a|x) \exp(\eta R(x, a)),
$$

Next, we compute the derivative of the policy $\pi_R^\eta(a|x)$. For any $(x, a) \in \mathcal{X} \times \mathcal{A}$, we have

$$
\begin{aligned}
\frac{\partial \pi_R^\eta(a|x)}{\partial R(x,a)} &= \frac{\partial}{\partial R(x,a)} \left( \frac{1}{Z_R(x)} \pi_{\text{ref}}(a|x) \exp(\eta R(x,a)) \right) \\
&= \frac{\eta \pi_{\text{ref}}(a|x) \exp(\eta R(x,a))}{Z_R(x)} - \frac{\pi_{\text{ref}}(a|x) \exp(\eta R(x,a))}{Z_R(x)^2} \cdot \frac{\partial Z_R(x)}{\partial R(x,a)} \\
&= \frac{\eta \pi_{\text{ref}}(a|x) \exp(\eta R(x,a))}{Z_R(x)} - \frac{\pi_{\text{ref}}(a|x) \exp(\eta R(x,a))}{Z_R(x)^2} \cdot \eta \pi_{\text{ref}}(a|x) \exp(\eta R(x,a)) \\
&= \eta \pi_R^\eta(a|x) - \eta \pi_R^\eta(a|x)^2.
\end{aligned}
$$

Moreover, for any $(x, a, a') \in \mathcal{X} \times \mathcal{A} \times \mathcal{A}$ with $a' \neq a$, we obtain

$$
\begin{aligned}
\frac{\partial \pi_R^\eta(a'|x)}{\partial R(x,a)} &= \pi_{\text{ref}}(a'|x) \exp\big(\eta R(x,a')\big) \cdot \frac{\partial}{\partial R(x,a)} \left( \frac{1}{Z_R(x)} \right) \\
&= -\frac{\pi_{\text{ref}}(a'|x) \exp(\eta R(x,a'))}{Z_R(x)^2} \cdot \eta \pi_{\text{ref}}(a|x) \exp(\eta R(x,a)) \\
&= -\eta \pi_R^\eta(a'|x) \pi_R^\eta(a|x).
\end{aligned}
$$

Finally, we compute the derivative of $\mu_R(x) = \mathbb{E}_{a \sim \pi_R^\eta(\cdot|x)}[R(x,a) - R^\star(x,a)]$. For any $(x, a) \in \mathcal{X} \times \mathcal{A}$, we have

$$
\begin{aligned}
\frac{\partial \mu_R(x)}{\partial R(x,a)} &= \sum_{a' \in \mathcal{A}} \frac{\partial \pi_R^\eta(a'|x)}{\partial R(x,a)} \big(R(x,a') - R^\star(x,a')\big) + \pi_R^\eta(a|x) \\
&= \eta \pi_R^\eta(a|x) \sum_{a' \in \mathcal{A}} \big(\mathbf{1}_{a=a'} - \pi_R^\eta(a'|x)\big) \cdot \big(R(x,a') - R^\star(x,a')\big) + \pi_R^\eta(a|x) \\
&= \eta \pi_R^\eta(a|x) \big(R(x,a) - R^\star(x,a) - \mu_R(x)\big) + \pi_R^\eta(a|x).
\end{aligned}
$$

Thus, we conclude the proof of Lemma B.4. $\qquad\square$

### B.3.2 PROOF OF LEMMA B.5

*Proof of Lemma B.5.* For the first property, a simple calculation gives

$$
\begin{aligned}
\frac{\mathrm{d}}{\mathrm{d}\alpha} \pi_\alpha(a|x) &= \frac{\pi_{\text{ref}}(a|x) \exp\big(\eta R_\alpha(x,a)\big) \cdot \eta \Delta R(x,a) Z_\alpha(x) - \pi_{\text{ref}}(a|x) \exp\big(\eta R_\alpha(x,a)\big) \cdot \frac{\mathrm{d}Z_\alpha(x)}{\mathrm{d}\alpha}}{Z_\alpha(x)^2} \\
&= \frac{\pi_{\text{ref}}(a|x) \exp\big(\eta R_\alpha(x,a)\big)}{Z_\alpha(x)} \left[ \eta \Delta R(x,a) - \frac{1}{Z_\alpha(x)} \frac{\mathrm{d}Z_\alpha(x)}{\mathrm{d}\alpha} \right] \\
&= \pi_\alpha(a|x) \left[ \eta \Delta R(x,a) - \frac{1}{Z_\alpha(x)} \frac{\mathrm{d}Z_\alpha(x)}{\mathrm{d}\alpha} \right]. \quad\quad\quad\quad (B.7)
\end{aligned}
$$

Moreover, we get

$$
\begin{aligned}
\frac{\mathrm{d}Z_\alpha(x)}{\mathrm{d}\alpha} &= \sum_{a \in \mathcal{A}} \pi_{\text{ref}}(a|x) \exp\big(\eta R_\alpha(x,a)\big) \cdot \eta \Delta R(x,a) = \eta Z_\alpha(x) \sum_{a \in \mathcal{A}} \pi_\alpha(a|x) \Delta R(x,a) \\
&= \eta Z_\alpha(x) \mathbb{E}_{\pi_\alpha}[\Delta R(x,a)]. \quad\quad\quad\quad (B.8)
\end{aligned}
$$

Plugging Equation B.8 into Equation B.7, we obtain the first property.

Now, we prove the second property.

$$
\begin{aligned}
\frac{\mathrm{d}}{\mathrm{d}\alpha} \mathbb{E}_{\pi_\alpha}[\Delta R(x,a)] &= \sum_{a \in \mathcal{A}} \frac{\mathrm{d}\pi_\alpha(a|x)}{\mathrm{d}\alpha} \Delta R(x,a) \\
&= \eta \sum_{a \in \mathcal{A}} \pi_\alpha(a|x) \big(\Delta R(x,a) - \mathbb{E}_{\pi_\alpha}[\Delta R(x,a)]\big) \Delta R(x,a) \quad \text{(first property)} \\
&= \eta \left( \mathbb{E}_{\pi_\alpha}[\Delta R(x,a)^2] - (\mathbb{E}_{\pi_\alpha}[\Delta R(x,a)])^2 \right) \\
&= \eta \mathbb{E}_{\pi_\alpha} \left[ (\Delta R(x,a) - \mathbb{E}_{\pi_\alpha}[\Delta R(x,a)])^2 \right].
\end{aligned}
$$

Similarly, substituting $\Delta R(x,a)$ with $\Delta R(x,a)^2$ in the above analysis, we obtain

$$\frac{\mathrm{d}}{\mathrm{d}\alpha}\mathbb{E}_{\pi_\alpha}[\Delta R(x,a)^2] = \eta \sum_{a \in \mathcal{A}} \pi_\alpha(a|x)\big(\Delta R(x,a) - \mathbb{E}_{\pi_\alpha}[\Delta R(x,a)]\big)\Delta R(x,a)^2 \quad \text{(first property)}$$

$$= \eta\left(\mathbb{E}_{\pi_\alpha}[\Delta R(x,a)^3] - \mathbb{E}_{\pi_\alpha}[\Delta R(x,a)^2]\mathbb{E}_{\pi_\alpha}[\Delta R(x,a)]\right),$$

which proves the last property. $\qquad\square$

### B.4 DISCUSSION ON SPECIFIC FUNCTION CLASSES

In this subsection, we supplement the result of Theorem 1 by providing a more detailed discussion of the tightness of our (unregularized) regret bound for several special function classes. We set the reference policy to be uniform, i.e., $\pi_{\text{ref}} = \text{Unif}(\mathcal{A})$. Then, for any policy $\pi$, it holds that $\text{KL}(\pi\|\pi_{\text{ref}}) = \sum_a \big(\pi(a)\log\pi(a) - \pi(a)\log\frac{1}{|\mathcal{A}|}\big) \leqslant \log|\mathcal{A}| = \log N$. Hence, KL-EXP yields the following regret bounds for special function classes:

**1. Linear classes:** When $R^\star \in \mathcal{R}$ and the reward function class $\mathcal{R}$ is linear, i.e., $\mathcal{R} = \{R : R = \phi(x,a)^\top\theta, \theta \in \mathbb{R}^d, \|\theta\|_2 \leqslant 1\}$, where $\phi(x,a) \in \mathbb{R}^d$ is a known feature map satisfying $\|\phi(x,a)\|_2 \leqslant 1$, the Vovk–Azoury–Warmuth forecaster (Vovk, 1997; Azoury & Warmuth, 2001) guarantees $\text{Reg}_{\text{Sq}}(T) = \mathcal{O}(d\log(T/d))$ (Example 1), which implies **Regret**$(T) = \mathcal{O}(\sqrt{dT\log N \log T})$. As stated in Remark 3, this bound is minimax-optimal, matching the lower bound $\Omega(\sqrt{dT\log N \log(T/d)})$ (Li et al., 2019) up to logarithmic $d$ factors. It is remarkable that we obtain this $\tilde{\mathcal{O}}(\sqrt{dT\log N})$-type regret bound without relying on the difficult-to-implement "layered data partitioning" technique required in prior works (Auer, 2002; Chu et al., 2011; Li et al., 2019). Our algorithm is simple to implement: it only requires solving the KL-regularized objective in Equation 1 (with the closed-form solution in Equation 2) using the reward estimator $\widehat{R}_t$ returned by the online regression oracle. We believe this opens a promising direction for developing algorithms that are both practical and statistically optimal in linear contextual bandits.

**2. Multi-armed bandits (MABs):** The function class in an MAB problem can be viewed as an $N$-dimensional hypercube. Consequently, the MAB setting follows directly from the linear case by taking $d = N$. In this case, we achieve $\text{Reg}_{\text{Sq}}(T) = \mathcal{O}(N\log(T/N))$ and **Regret**$(T) = \mathcal{O}(\sqrt{NT\log N \log(T/N)})$, which matches the lower bound $\Omega(\sqrt{NT})$ of Auer et al. (2002) up to logarithmic factors.

**3. Generalized linear models (GLMs):** For GLM reward function class, i.e., $\mathcal{R} = \{R : R = \mu(\phi(x,a)^\top\theta), \theta \in \mathbb{R}^d, \|\theta\|_2 \leqslant 1\}$, where $\mu : \mathbb{R} \to [0,1]$ is a fixed non-decreasing 1-Lipschitz link function and $\phi(x,a) \in \mathbb{R}^d$ is a known feature map with $\|\phi(x,a)\|_2 \leqslant 1$, if $R^\star \in \mathcal{R}$, the GLMtron algorithm (Kakade et al., 2011) guarantees $\text{Reg}_{\text{Sq}}(T) = \mathcal{O}(\kappa_\mu^2 d\log(T/d))$, where $1/\dot{\mu} \leqslant \kappa_\mu$. This, in turn, implies **Regret**$(T) = \mathcal{O}(\kappa_\mu\sqrt{dT\log N \log T})$, which is tighter than the bound $\mathcal{O}(\kappa_\mu(\log T)^{1.5}\sqrt{dT\log N})$ (Li et al., 2017) by a factor of $\log T$. On the other hand, Lee et al. (2024); Sawarni et al. (2024) establish a $\kappa_\mu$-improved regret bound of $\tilde{\mathcal{O}}\left(d\sqrt{T/\kappa_\mu^\star}\right)$, where $\kappa_\mu^\star := \frac{1}{\dot{\mu}((x^\star)^\top\theta^\star)}$, though with a looser dependence on $\sqrt{d}$ than ours. It remains an open question whether a $\tilde{\mathcal{O}}(\sqrt{dT\log N})$-type regret bound can be attained while simultaneously improving the dependence on $\kappa_\mu$.

**4. Bounded eluder dimension:** Under the realizability assumption (Assumption 1), i.e., $R^\star \in \mathcal{R}$, and the reward function class $\mathcal{R}$ has bounded eluder dimension (Definition C.1), the empirical risk minimization (ERM) algorithm achieves, with probability at least $1 - \delta$, $\text{Reg}_{\text{Sq}}(T) = \mathcal{O}(d_{\text{E}}\log(\mathcal{N}_{\mathcal{R}}(\epsilon)T))$ (Lemma C.2). Consequently, we obtain the unregularized regret bound **Regret**$(T) = \mathcal{O}(\sqrt{d_{\text{E}}T\log N \log(\mathcal{N}_{\mathcal{R}}(\epsilon)T)})$. In comparison, the existing bound of Russo & Van Roy (2013) is $\mathcal{O}(\sqrt{d_{\text{E}}T\log(\mathcal{N}_{\mathcal{R}}(\epsilon)T)})$, which shows that our result is tight up to a $\sqrt{\log N}$ factor.

**Remark B.3** (Not directly applicable to finite function classes)**.** *Our analysis is not directly applicable to the finite function class setting (Agarwal et al., 2012), as a finite class violates Assumption 2. In particular, the derivative-based arguments employed in Lemmas B.2, B.4, and B.5 do not hold in this case. For a finite function class $\mathcal{R}$, we instead consider its convex hull $\text{conv}(\mathcal{R})$ (so that Assumption 2 holds)*

*and analyze it using eluder-dimension arguments. This gives* $\mathrm{Reg}_{\mathrm{Sq}}(T) = \mathcal{O}\big(d_{\mathrm{E}} \log(\mathcal{N}_{\mathrm{conv}(\mathcal{R})}(\epsilon)T)\big)$ *and* $\mathbf{Regret}(T) = \mathcal{O}\big(\sqrt{d_{\mathrm{E}} T \log N \log(\mathcal{N}_{\mathrm{conv}(\mathcal{R})}(\epsilon)T)}\big)$, *where* $d_{\mathrm{E}}$ *denotes the eluder dimension with respect to* $\mathrm{conv}(\mathcal{R})$, *and* $\mathcal{N}_{\mathrm{conv}(\mathcal{R})}(\epsilon)$ *is its* $\epsilon$-*covering number (see Section C for complete proofs). Compared to the minimax-optimal (unregularized) regret bound* $\mathcal{O}(\sqrt{NT \log|\mathcal{R}|})$ *established by Foster & Rakhlin (2020), our bound can be looser since* $d_{\mathrm{E}} \log(\mathcal{N}_{\mathrm{conv}(\mathcal{R})}(\epsilon)T)$ *is typically larger than* $N \log|\mathcal{R}|$, *especially when* $|\mathcal{R}|$ *is small. Therefore, for problems with a finite function class, we recommend using the* `SquareCB` *algorithm proposed by Foster & Rakhlin (2020).*

# C   CASE: $\mathcal{R}$ WITH BOUNDED ELUDER DIMENSION (REMARK 2)

In this subsection, we analyze the setting where the reward function class $\mathcal{R}$ has bounded eluder dimension (Russo & Van Roy, 2013), in order to enable a direct comparison with prior work (Zhao et al., 2025a).

We define the uncertainty and eluder dimension, following Zhao et al. (2025a).

**Definition C.1.** *For any sequence* $\mathcal{D}_t = \{(x_s, a_s)\}_{s=1}^{t-1}$, *we define the uncertainty of* $(x, a)$ *with respect to* $\mathcal{R}$ *as:*

$$U_{\mathcal{R},\lambda}(x, a; \mathcal{D}_t) := \sup_{R_1, R_2 \in \mathcal{R}} \frac{|R_1(x, a) - R_2(x, a)|}{\sqrt{\lambda + \sum_{s=1}^{t-1} \big(R_1(x_s, a_s) - R_2(x_s, a_s)\big)^2}}.$$

*And the eluder dimension is defined as:*

$$d_{\mathrm{E}} := \sup_{x_{1:T}, a_{1:T}} \sum_{t=1}^{T} \min\left\{1, U_{\mathcal{R},\lambda}(x_t, a_t; \mathcal{D}_t)^2\right\}. \tag{C.1}$$

We also define the confidence set $\mathcal{R}_t$ as follows:

$$\mathcal{R}_t := \left\{ R \in \mathcal{R} : \sum_{s=1}^{t-1} \big(R(x_s, a_s) - \widehat{R}_t(x_s, a_s)\big)^2 + \lambda \leqslant \beta_T^2 = 16 \log(\mathcal{N}_{\mathcal{R}}(\epsilon)T/\delta) \right\},$$

where $\lambda > 0$. We can then bound the estimation error using the following lemma.

**Lemma C.1** (Lemma 4.5 of Zhao et al. 2025a). *Let* $\widehat{R}_t$ *be the empirical risk minimizer (ERM), i.e.,* $\widehat{R}_t \leftarrow \mathrm{argmin}_{R \in \mathcal{R}} \sum_{s=1}^{t-1}(R(x_s, a_s) - y_s)^2$. *Then, under Assumption 1 and the condition that the noises* $\epsilon_t$ *are conditional 1-subGaussian, we have with probability at least* $1 - \delta$, *for all* $t \in [T]$, *we have*

$$\widehat{R}_t(x, a) - R^{\star}(x, a) \leqslant \min\{1, \beta_T \cdot U_{\mathcal{R}_t, \lambda}(x, a; \mathcal{D}_t)\}, \quad \forall(x, a) \in \mathcal{X} \times \mathcal{A}.$$

The following lemma is useful for the subsequent analysis.

**Lemma C.2.** *Under Assumption 1, if* `OracleSq` *is chosen as the standard ERM algorithm, then with probability at least* $1 - \delta$ *we obtain*

$$\sum_{t=1}^{T} \big(\widehat{R}_t(x_t, a_t) - r_t\big)^2 - \sum_{t=1}^{T} \big(R^{\star}(x_t, a_t) - r_t\big)^2 = \mathcal{O}\big(d_{\mathrm{E}} \log(\mathcal{N}_{\mathcal{R}}(\epsilon)T)\big).$$

*Proof of Lemma C.2.* Let $M_t := (\widehat{R}_t(x_t, a_t) - r_t)^2 - (R^{\star}(x_t, a_t) - r_t)^2$ and $Z_t := M_t - \mathbb{E}[M_t \mid \mathcal{F}_{t-1}]$. We define the filtration $\mathcal{F}_{t-1} = \sigma(x_1, a_1, r_1, \ldots, x_{t-1}, a_{t-1}, r_{t-1}, x_t)$. Then, by Lemma B.1

and Freedman's inequality (Lemma G.1) with $\beta = 1/8$, with probability at least $1 - \delta$, we have

$$\sum_{t=1}^{T} M_t \leqslant \sum_{t=1}^{T} \mathbb{E}[M_t | \mathcal{F}_{t=1}] + \frac{1}{8} \sum_{t=1}^{T} \mathbb{E}[Z_t^2 | \mathcal{F}_{t-1}] + 8 \log \frac{1}{\delta} \qquad \text{(Lemma G.1, w.p. } 1 - \delta\text{)}$$

$$\leqslant \frac{3}{2} \sum_{t=1}^{T} \mathbb{E}[M_t | \mathcal{F}_{t-1}] + 8 \log \frac{1}{\delta} \qquad \text{(Lemma B.1)}$$

$$= \frac{3}{2} \sum_{t=1}^{T} \mathbb{E}_{a \sim \pi_t} \left[ (\widehat{R}_t(x_t, a_t) - R^\star(x_t, a_t))^2 \mid \mathcal{F}_{t-1} \right] + 8 \log \frac{1}{\delta}$$

$$\leqslant 3 \sum_{t=1}^{T} \left( \widehat{R}_t(x_t, a_t) - R^\star(x_t, a_t) \right)^2 + 16 \log \frac{2}{\delta}. \qquad \text{(Lemma G.2, w.p. } 1 - \delta\text{)}$$

Hence, we derive

$$\sum_{t=1}^{T} \left( \widehat{R}_t(x_t, a_t) - r_t \right)^2 - \sum_{t=1}^{T} \left( R^\star(x_t, a_t) - r_t \right)^2$$

$$\leqslant 3 \sum_{t=1}^{T} \left( \widehat{R}_t(x_t, a_t) - R^\star(x_t, a_t) \right)^2 + 16 \log \frac{2}{\delta}$$

$$\leqslant 3\beta_T^2 \sum_{t=1}^{T} \min \left\{ 1, U_{\mathcal{R}_t, \lambda}(x_t, a_t; \mathcal{D}_t)^2 \right\} + 16 \log \frac{2}{\delta} \qquad \text{(Lemma C.1, w.p. } 1 - \delta\text{)}$$

$$\leqslant 48 d_{\mathrm{E}} \log(\mathcal{N}_{\mathcal{R}}(\epsilon) T / \delta) + 16 \log \frac{2}{\delta}.$$

By setting $\delta \leftarrow \frac{\delta}{3}$, the proof is complete. $\qquad\square$

We now present the claim in Remark 2 more formally.

**Proposition C.1** (Regret under bounded eluder dimension)**.** *Suppose the eluder dimension defined in Equation C.1 is finite. Let the online regression oracle* `OracleSq` *be the ERM predictor. Under Assumptions 1 and 3, for any $\delta > 0$,* `KL-EXP` *(Algorithm 1) guarantees that with probability at least $1 - \delta$,*

$$\textbf{Regret}_{\mathrm{KL}}(T, \eta) = \mathcal{O}\big(\eta d_{\mathrm{E}} \log\left(\mathcal{N}_{\mathcal{R}}(\epsilon) T\right)\big), \quad \text{and} \quad \textbf{Regret}(T) = \mathcal{O}\left(\eta d_{\mathrm{E}} \log\left(\mathcal{N}_{\mathcal{R}}(\epsilon) T\right) + \frac{DT}{\eta}\right),$$

*where $D := \frac{1}{T} \sum_{t=1}^{T} \mathrm{KL}\big(\pi^\star(\cdot \| x_t) \| \pi_{\mathrm{ref}}(\cdot \| x_t)\big)$.*

*Proof of Proposition C.1.* Then, following a similar analysis to the proof of Theorem 1, we can bound the regret as follows:

$$\textbf{Regret}_{\mathrm{KL}}(T, \eta) = \sum_{t=1}^{T} J_t^\eta(\pi_\eta^\star, R^\star) - J_t^\eta(\pi_t, R^\star)$$

$$\leqslant \eta \sum_{t=1}^{T} \mathbb{E}_{a_t \sim \pi_t(\cdot | x_t)} \left[ \left( \widehat{R}_t(x_t, a_t) - R^\star(x_t, a_t) \right)^2 \right] \qquad \text{(Lemma B.2)}$$

$$\leqslant 2\eta \left[ \sum_{t=1}^{T} \left( \widehat{R}_t(x_t, a_t) - r_t \right)^2 - \sum_{t=1}^{T} \left( R^\star(x_t, a_t) - r_t \right)^2 \right] + 16 \log \frac{1}{\delta}$$
$$\text{(Lemma G.1 and B.1 w.p. } 1 - \delta\text{)}$$

$$= \mathcal{O}\big(\eta d_{\mathrm{E}} \log\left(\mathcal{N}_{\mathcal{R}}(\epsilon) T\right)\big). \qquad \text{(Lemma C.2 w.p. } 1 - \delta\text{)}$$

Setting $\delta \leftarrow \frac{\delta}{2}$ yields the bound for $\textbf{Regret}_{\mathrm{KL}}(T, \eta)$.

The bound for $\textbf{Regret}(T)$ then follows directly from Lemma B.3. Thus, the proof of Proposition C.1 is complete. $\qquad\square$

---

**Algorithm D.1** OEPO (**O**racle-**E**fficient **P**olicy **O**ptimization)

---

1: **Inputs:** regularization parameter $\eta$, reference policy $\pi_{\text{ref}}$, online regression oracle OracleLog.
2: **Initialize:** choose any $\widehat{R}_1 \in \mathcal{R}$.
3: **for** round $t = 1$ to $T$ **do**
4:     Observe context $x_t \in \mathcal{X}$.
5:     Compute policy $\pi_t(\cdot|x_t) \propto \pi_{\text{ref}}(\cdot|x_t) \exp\big(\eta \widehat{R}_t(x_t, \cdot)\big)$ via Equation 2.
6:     Sample action $a_t^1, a_2^t \sim \pi_t(\cdot|x_t)$ and receive preference feedback $y_t$.
7:     Update $\widehat{R}_{t+1}$ for the next round using OracleLog via Equation 7.
8: **end for**

---

## D    PROOF OF THEOREM 3

In this section, we present the proof of Theorem 3.

### D.1    MAIN PROOF OF THEOREM 3

We begin by introducing the key lemmas used to prove Theorem 3.

**Lemma D.1.** *With probability at least $1 - \delta$, we have*

$$\sum_{t=1}^{T} \left( [R^\star(x_t, a_t^1) - \widehat{R}_t(x_t, a_t^1)] - [R^\star(x_t, a_t^2) - \widehat{R}_t(x_t, a_t^2)] \right)^2$$

$$\leqslant \kappa^2 \left( \sum_{t=1}^{T} \ell_t(\widehat{R}_t) - \sum_{t=1}^{T} \ell_t(R^\star) \right) + 2\kappa^2 \log \frac{1}{\delta}.$$

The proof is deferred to Appendix D.2.1.

**Lemma D.2** (Second-order regret decomposition with baseline)**.** *Under Assumption 1 and 2, for any $t \in [T]$ and any $g : \mathcal{X} \to \mathbb{R}$, we have*

$$J_t^\eta(\pi_\eta^\star, R^\star) - J_t^\eta(\pi_t, R^\star) \leqslant \eta \mathbb{E}_{a \sim \pi_t(\cdot|x_t)} \left[ \left( \widehat{R}_t(x_t, a) - R^\star(x_t, a) + g(x_t) \right)^2 \right].$$

The proof is deferred to Appendix D.2.2.

We now provide the proof of Theorem 3.

*Proof of Theorem 3.* By applying Lemma D.2 with setting

$$g_t(x) = -\mathbb{E}_{a^2 \sim \pi_t(\cdot|x)} \left[ \widehat{R}_t(x, a^2) - R^\star(x, a^2) \right],$$

we have

$$\textbf{Regret}_{\text{KL}}(T, \eta) = \sum_{t=1}^{T} J_t^\eta(\pi_\eta^\star, R^\star) - J_t^\eta(\pi_t, R^\star)$$

$$\leqslant \eta \sum_{t=1}^{T} \mathbb{E}_{a^1, a^2 \sim \pi_t(\cdot|x_t)} \left[ \left( \widehat{R}_t(x_t, a^1) - R^\star(x_t, a^1) - \left( \widehat{R}_t(x_t, a^2) - R^\star(x_t, a^2) \right) \right)^2 \right]$$

$$\text{(Lemma D.2)}$$

$$\leqslant 2\eta \sum_{t=1}^{T} \left( \widehat{R}_t(x_t, a_t^1) - R^\star(x_t, a_t^1) - \left( \widehat{R}_t(x_t, a_t^2) - R^\star(x_t, a_t^2) \right) \right)^2 + 32\eta \log \frac{2}{\delta}$$

$$\text{(Lemma G.2, w.p. } 1 - \delta)$$

$$\leqslant 2\eta\kappa^2 \left( \sum_{t=1}^{T} \ell_t(\widehat{R}_t) - \sum_{t=1}^{T} \ell_t(R^\star) \right) + 4\eta\kappa^2 \log \frac{1}{\delta} + 32\eta \log \frac{2}{\delta} \quad \text{(Lemma D.1, w.p. } 1 - \delta)$$

$$\leqslant 2\eta\kappa^2 \text{Reg}_{\text{Log}}(T) + 4\eta\kappa^2 \log \frac{1}{\delta} + 32\eta \log \frac{2}{\delta}. \quad \text{(Assumption 4)}$$

By setting $\delta \leftarrow \frac{\delta}{2}$, we establish the bound for $\mathbf{Regret}_{\mathrm{KL}}(T, \eta)$.

Furthermore, the bound on $\mathbf{Regret}(T)$ follows immediately from Lemma B.3, using the same analysis as in the proof of Theorem 1. Hence, this completes the proof of Theorem 3. $\qquad\square$

## D.2 PROOFS OF LEMMAS FOR THEOREM 3

### D.2.1 PROOF OF LEMMA D.1

*Proof of Lemma D.1.* The proof of Lemma D.1 follows the analysis of Lemma D.1 in Zhao et al. (2024). However, unlike Zhao et al. (2024), where the estimator $\widehat{R}$ is fixed for all $t$, our setting accommodates a time-varying sequence $\{\widehat{R}_t\}_{t=1}^T$.

For completeness, we present the full proof below.

For simplicity, we write $p_t^\star = \sigma\big(R^\star(x_t, a_t^1) - R^\star(x_t, a_t^2)\big)$ and $p_t = \sigma\big(\widehat{R}_t(x_t, a_t^1) - \widehat{R}_t(x_t, a_t^2)\big)$. We define

$$X_t := \frac{1}{2}\left(\ell_t(R^\star) - \ell_t(\widehat{R}_t)\right) = -\frac{1}{2}\left(y_t \log \frac{p_t^\star}{p_t} + (1 - y_t)\log \frac{1 - p_t^\star}{1 - p_t}\right).$$

Then, by Lemma G.3, with probability at least $1 - \delta$, we have

$$\frac{1}{2}\left(\sum_{t=1}^T \ell_t(R^\star) - \sum_{t=1}^T \ell_t(\widehat{R}_t)\right) = \sum_{t=1}^T X_t \leqslant \sum_{t=1}^T \log\big(\mathbb{E}_{t-1}[e^{X_t}]\big) + \log\frac{1}{\delta} \qquad \text{(Lemma G.3)}$$

$$= \sum_{t=1}^T \log\left(p_t^\star\left(\frac{p_t^\star}{p_t}\right)^{-1/2} + (1 - p_t^\star)\left(\frac{1 - p_t^\star}{1 - p_t}\right)^{-1/2}\right) + \log\frac{1}{\delta}$$

$$= \sum_{t=1}^T \log\left(\sqrt{p_t^\star p_t} + \sqrt{(1 - p_t^\star)(1 - p_t)}\right) + \log\frac{1}{\delta}$$

$$\leqslant \sum_{t=1}^T \left(\sqrt{p_t^\star p_t} + \sqrt{(1 - p_t^\star)(1 - p_t)} - 1\right) + \log\frac{1}{\delta}$$

$$(\log x \leqslant x - 1, \text{ for } x > 0)$$

$$= -\frac{1}{2}\sum_{t=1}^T \left[\left(\sqrt{p_t^\star} - \sqrt{p_t}\right)^2 + \left(\sqrt{1 - p_t^\star} - \sqrt{1 - p_t}\right)^2\right] + \log\frac{1}{\delta}$$

$$(1 = \tfrac{1}{2}(p_t^\star + (1 - p_t^\star) + p_t + (1 - p_t)))$$

$$\leqslant -\frac{1}{2}\sum_{t=1}^T (p_t^\star - p_t)^2 + \log\frac{1}{\delta}. \qquad (\text{D.1})$$

where the last inequality follows from the fact that, for any $p, q \in [0, 1]$, $(\sqrt{p} - \sqrt{q})^2 + (\sqrt{1 - p} - \sqrt{1 - q})^2 \geqslant (p - q)^2$.

Now, consider the term $p_t^\star - p_t$. For simplicity, let $\Delta_t^\star = R^\star(x_t, a_t^1) - R^\star(x_t, a_t^2)$ and $\Delta_t = \widehat{R}_t(x_t, a_t^1) - \widehat{R}_t(x_t, a_t^2)$. Then, by the mean value theorem, we obtain

$$p_t^\star - p_t = \sigma(\Delta_t^\star) - \sigma(\Delta_t)$$

$$= (\Delta_t^\star - \Delta_t)\int_0^1 \dot{\sigma}\left(\Delta_t + \tau(\Delta_t^\star - \Delta_t)\right)\mathrm{d}\tau \qquad \text{(mean value theorem)}$$

$$\geqslant \frac{1}{\kappa}\left(\Delta_t^\star - \Delta_t\right). \qquad (\dot{\sigma}(z) \geqslant \tfrac{1}{\kappa}, \text{ Definition of } \kappa)$$

Hence, substituting the above result into Equation D.1 and rearranging terms, we obtain

$$\sum_{t=1}^T \left([R^\star(x_t, a_t^1) - R^\star(x_t, a_t^2)] - [\widehat{R}_t(x_t, a_t^1) - \widehat{R}_t(x_t, a_t^2)]\right)^2$$

$$\leqslant \kappa^2\left(\sum_{t=1}^T \ell_t(\widehat{R}_t) - \sum_{t=1}^T \ell_t(R^\star)\right) + 2\kappa^2 \log\frac{1}{\delta},$$

---

**Algorithm D.2** ODPO (**O**racle-efficient **D**irect **P**olicy **O**ptimization)

---

1: **Inputs:** regularization parameter $\eta$, reference policy $\pi_{\text{ref}}$, online regression oracle `OracleLog`.
2: **Initialize:** choose any $\pi_1 \in \Pi$.
3: **for** round $t = 1$ to $T$ **do**
4:     Observe context $x_t \in \mathcal{X}$.
5:     Sample action $a_t^1, a_2^t \sim \pi_t(\cdot|x_t)$ and receive preference feedback $y_t$.
6:     Update $\pi_{t+1}$ for the next round using `OracleDPO` via Equation E.2.
7: **end for**

---

which concludes the proof. $\qquad\qquad\qquad\qquad\qquad\qquad\qquad\qquad\qquad\qquad\qquad\square$

### D.2.2 Proof of Lemma D.2

*Proof of Lemma D.2.* Recall the definition of $f : \mathcal{X} \times \mathcal{R} \to \mathbb{R}$ in equation B.4:

$$f(x, R) := -\frac{1}{\eta} \log Z_R(x) + \mathbb{E}_{\pi_R^\eta}\left[R(x, a) - R^\star(x, a)\right].$$

Note $f$ is invariant to adding any action-independent baseline $g : \mathcal{X} \to \mathbb{R}$.

$$f(x, R + g) = -\frac{1}{\eta} \log Z_{R+g}(x) + \mathbb{E}_{\pi_{R+g}^\eta}\left[R(x, a) + g(x) - R^\star(x, a)\right]$$

$$= -\frac{1}{\eta}\left(\log Z_R(x) + \eta g(x)\right) + \mathbb{E}_{\pi_R^\eta}\left[R(x, a) + g(x) - R^\star(x, a)\right] \quad (\pi_{R+g}^\eta = \pi_R^\eta)$$

$$= -\frac{1}{\eta} \log Z_R(x) + \mathbb{E}_{\pi_R^\eta}\left[R(x, a) - R^\star(x, a)\right] = f(x, R),$$

where the second equality holds because

$$Z_{R+g}(x) = \sum_{a \in \mathcal{A}} \pi_{\text{ref}}(a|x) e^{\eta(R(x,a) + g(x))} = e^{\eta g(x)} \sum_{a \in \mathcal{A}} \pi_{\text{ref}}(a|x) e^{\eta R(x,a)} = e^{\eta g(x)} Z_R(x),$$

and

$$\pi_{R+g}^\eta(a|x) = \frac{\pi_{\text{ref}}(a|x) \cdot e^{\eta(R(x,a)+g(x))}}{Z_{R+g}(x)} = \frac{\pi_{\text{ref}}(a|x) \cdot e^{\eta R(x,a)} \cdot e^{\eta g(x)}}{e^{\eta g(x)} Z_R(x)} = \pi_R^\eta(a|x).$$

Therefore, by substituting $\widehat{R}_t(x, a) \leftarrow \widehat{R}_t(x, a) + g(x)$ and the following the proof from Equation B.4 in Lemma B.2, we derive

$$J_t^\eta(\pi_\eta^\star, R^\star) - J_t^\eta(\pi_t, R^\star) \leqslant \eta \mathbb{E}_{a \sim \pi_t(\cdot|x_t)}\left[\left(\widehat{R}_t(x_t, a) - R^\star(x_t, a) + g(x_t)\right)^2\right].$$

which concludes the proof. $\qquad\qquad\qquad\qquad\qquad\qquad\qquad\qquad\qquad\qquad\qquad\square$

## E Extension to Direct Preference Optimization (DPO)

In this section, we extend our method to the DPO objective (Rafailov et al., 2023). The problem setup is identical to the RLHF setting (Subsection 3.2), except that DPO bypasses reward learning and directly optimizes the policy within the policy class $\Pi$. Rearranging Equation 2, we can express the reward function as follows:

$$R(x, a) = \frac{1}{\eta} \log \frac{\pi(a|x)}{\pi_{\text{ref}}(a|x)} + \frac{1}{\eta} \log Z_R(x). \tag{E.1}$$

Accordingly, the Bradley–Terry model for preference feedback takes the form

$$\mathbb{P}(a^1 > a^2|x, a^1, a^2) = \sigma\left(\frac{1}{\eta} \log \frac{\pi(a^1|x)}{\pi_{\text{ref}}(a^1|x)} - \frac{1}{\eta} \log \frac{\pi(a^2|x)}{\pi_{\text{ref}}(a^2|x)}\right),$$

where $\sigma(x) = \frac{1}{1+e^{-x}}$ is the sigmoid function. Finally, the DPO loss at round $t$ is defined as

$$\ell_t^{\mathrm{DPO}}(\pi) := -\log \sigma\left(\frac{1}{\eta}\log\frac{\pi(a_t^1|x_t)}{\pi_{\mathrm{ref}}(a_t^1|x_t)} - \frac{1}{\eta}\log\frac{\pi(a_t^2|x_t)}{\pi_{\mathrm{ref}}(a_t^2|x_t)}\right).$$

Note that $\ell_t^{\mathrm{DPO}}(\pi)$ is exactly the same as $\ell_t(R)$ defined in Equation 6.

Similar to Subsection 3.2, we assume access to an online DPO regression oracle, denoted by `OracleDPO`. At each round $t$, rather than estimating a reward function, this oracle directly returns a policy:

$$\pi_t \leftarrow \mathtt{OracleDPO}_t\left((x_1, a_1^1, a_1^2, y_1), \ldots, (x_{t-1}, a_{t-1}^1, a_{t-1}^2, y_{t-1})\right), \quad \text{where } \pi_t \in \Pi. \qquad \text{(E.2)}$$

We assume that the prediction error of `OracleDPO` is bounded with respect to the policy class $\Pi$.

**Assumption E.1** (Guarantee of online DPO regression oracle). *We assume that, for every (possibly adaptively chosen) sequence $x_{1:T}, a_{1:T}^1, a_{1:T}^2, y_{1:T}$, there exists regret bound $\mathrm{Reg}_{\mathrm{DPO}}(T)$ such that the regression oracle `OracleDPO` satisfies*

$$\sum_{t=1}^{T}\ell_t^{\mathrm{DPO}}(\pi_t) - \sum_{t=1}^{T}\ell_t^{\mathrm{DPO}}(\pi_\eta^\star) \leqslant \mathrm{Reg}_{\mathrm{DPO}}(T).$$

Using this oracle, we establish the following regret bound, analogous to Theorem 3.

**Theorem E.1** (Regret of `ODPO`). *Let $\delta > 0$ and $\kappa := \sup_{R,x,a}\frac{1}{\sigma(R(x,a))}$. Under Assumption 1, 2, and E.1, `ODPO` guarantees that with probability at least $1 - \delta$,*

$$\mathbf{Regret}_{\mathrm{KL}}(T, \eta) = \mathcal{O}\big(\eta\kappa^2\mathrm{Reg}_{\mathrm{DPO}}(T) + \eta\kappa^2\log(1/\delta)\big), \quad \text{and}$$
$$\mathbf{Regret}(T) = \mathcal{O}\left(\eta\kappa^2\mathrm{Reg}_{\mathrm{DPO}}(T) + \eta\kappa^2\log(1/\delta) + \frac{DT}{\eta}\right),$$

*where $D := \frac{1}{T}\sum_{t=1}^{T}\mathrm{KL}\big(\pi^\star(\cdot\|x_t)\|\pi_{\mathrm{ref}}(\cdot\|x_t)\big)$.*

*Proof of Theorem E.1.* By Lemma D.1, together with the fact that $\ell_t^{\mathrm{DPO}}(\pi) = \ell_t(R)$ and the reward reformulation in Equation E.1, we obtain

**Corollary E.1.** *With probability at least $1 - \delta$, we have*

$$\sum_{t=1}^{T}\left(\frac{1}{\eta}\log\pi_\eta^\star(a_t^1|x_t) - \frac{1}{\eta}\log\pi_t(a_t^1|x_t) - \left(\frac{1}{\eta}\log\pi_\eta^\star(a_t^2|x_t) - \frac{1}{\eta}\log\pi_t(a_t^2|x_t)\right)\right)^2$$
$$\leqslant \kappa^2\left(\sum_{t=1}^{T}\ell_t^{\mathrm{DPO}}(\pi_t) - \sum_{t=1}^{T}\ell_t^{\mathrm{DPO}}(\pi_\eta^\star)\right) + 2\kappa^2\log\frac{1}{\delta}.$$

Then, by Lemma D.2, we get

$$\mathbf{Regret}_{\mathrm{KL}}(T, \eta) = \sum_{t=1}^{T} J_t^{\eta}(\pi_\eta^\star, R^\star) - J_t^{\eta}(\pi_t, R^\star)$$

$$\leqslant \eta \sum_{t=1}^{T} \mathbb{E}_{a^1, a^2 \sim \pi_t(\cdot|x_t)} \left[ \left( \left( \widehat{R}_t(x_t, a^1) - R^\star(x_t, a^1) \right) - \left( \widehat{R}_t(x_t, a^2) - R^\star(x_t, a^2) \right) \right)^2 \right]$$

$$\text{(Lemma D.2 with } g_t(x_t) = -\mathbb{E}_{a^2 \sim \pi_t(\cdot|x_t)} \left[ \widehat{R}_t(x_t, a^2) - R^\star(x_t, a^2) \right] )$$

$$\leqslant 2\eta \sum_{t=1}^{T} \left( \widehat{R}_t(x_t, a_t^1) - R^\star(x_t, a_t^1) - \left( \widehat{R}_t(x_t, a_t^2) - R^\star(x_t, a_t^2) \right) \right)^2 + 32\eta \log \frac{2}{\delta}$$

$$\text{(Lemma G.2, w.p. } 1 - \delta)$$

$$= 2\eta \sum_{t=1}^{T} \left( \frac{1}{\eta} \log \pi_t(a_t^1|x_t) - \frac{1}{\eta} \log \pi_\eta^\star(a_t^1|x_t) - \left( \frac{1}{\eta} \log \pi_t(a_t^2|x_t) - \frac{1}{\eta} \log \pi_\eta^\star(a_t^2|x_t) \right) \right)^2$$

$$+ 32\eta \log \frac{2}{\delta} \qquad\qquad\qquad\qquad\qquad\qquad\qquad\qquad\qquad\qquad \text{(Equation E.1)}$$

$$\leqslant 2\eta\kappa^2 \left( \sum_{t=1}^{T} \ell_t^{\mathrm{DPO}}(\pi_t) - \sum_{t=1}^{T} \ell_t^{\mathrm{DPO}}(\pi_\eta^\star) \right) + 4\eta\kappa^2 \log \frac{1}{\delta} + 32\eta \log \frac{2}{\delta} \quad \text{(Corollary E.1, w.p. } 1 - \delta)$$

$$\leqslant 2\eta\kappa^2 \mathrm{Reg}_{\mathrm{DPO}}(T) + 4\eta\kappa^2 \log \frac{1}{\delta} + 32\eta \log \frac{2}{\delta}. \qquad\qquad\qquad\qquad \text{(Assumption E.1)}$$

By setting $\delta \leftarrow \frac{\delta}{2}$, we obtain the bound for $\mathbf{Regret}_{\mathrm{KL}}(T, \eta)$.

In addition, the bound for $\mathbf{Regret}(T)$ follows directly from Lemma B.3, by applying the same reasoning as in the proof of Theorem 1. This concludes the proof of Theorem E.1. $\qquad\square$

### E.1 COMPARISON TO LOWER BOUND IN PROPOSITION 2.1 OF XIE ET AL. (2024)

A careful reader might wonder whether the logarithmic KL-regularized regret established in Theorem E.1 contradicts the lower bound in Proposition 2.1 of Xie et al. (2024). This is not the case: their analysis considers only the restricted policy class $\Pi = \{\pi_{\mathrm{ref}}, \pi_\eta^\star\}$, rather than the full family of Gibbs policies (Equation 2), so their lower bound does not apply to our setting. For clarity, we first restate Proposition 2.1 from Xie et al. (2024).

**Proposition E.1** (Necessity of deliberate exploration, Proposition 2.1 of Xie et al. 2024). *Fix* $\eta > \frac{8}{\log 2}$, *and consider the two-armed bandit setting of* $\mathcal{X} = \varnothing$, *and* $|\mathcal{A}| = N = 2$. *Let* $\Pi = \{\pi_{\mathrm{ref}}, \pi_\eta^\star\}$. *There exists a reference policy* $\pi_{\mathrm{ref}}$ *such that for all* $T \leqslant \frac{1}{2} \exp\left(\frac{\eta}{8}\right)$, *with constant probability, all of policies* $\pi_1, \ldots, \pi_{T+1}$ *produced by* `OnlineDPO` *satisfiy*

$$\max_{\pi \in \Pi} J_t^{\eta}(\pi, R) - J_t^{\eta}(\pi_t, R) \geqslant \frac{1}{8}, \quad \forall t \in [T + 1].$$

As is clear, this proposition only applies to the restricted class $\Pi = \{\pi_{\mathrm{ref}}, \pi_\eta^\star\}$, where the learner can update its policy only by switching between these two candidates. In contrast, our analysis permits the learner to choose from the full family of Gibbs policies—beyond just $\{\pi_{\mathrm{ref}}, \pi_\eta^\star\}$—with the choice adaptively guided by data collected through online interactions. Therefore, their lower bound is not directly comparable to our upper bound.

## F KL-REGULARIZED CONTEXTUAL BANDITS WITH OFFLINE REGRESSION ORACLE

In this section, we assume access to an *offline regression oracle* instead of the online regression oracle defined in Equation 4. Note that an online regression oracle must provide robust guarantees against arbitrary data sequences generated by an adaptive adversary, which becomes challenging to implement when the function class $\mathcal{R}$ is complex. While the minimax regret rates for online

regression with general function classes are well understood (Rakhlin & Sridharan, 2014), to the best of our knowledge, computationally efficient algorithms are only known for specific function classes.

Unlike the online regression oracle setting, where contexts may be chosen adversarially, we now adopt a stochastic context assumption.

**Assumption F.1** (Stochastic context). *At each round $t$, the context $x_t \in \mathcal{X}$ is drawn i.i.d. from an unknown but fixed distribution $\rho$.*

In this section, we redefine the *KL-regularized* and *unregularized* regrets in the stochastic contextual setting as follows (we use the same regret notations for simplicity):

$$\mathbf{Regret}_{\mathrm{KL}}(T, \eta) := \sum_{t=1}^{T} \mathbb{E}_{x_t \sim \rho}\big[J_t^{\eta}(\pi_{\eta}^{\star}, R^{\star}) - J_t^{\eta}(\pi_t, R^{\star})\big] \quad \text{and}$$

$$\mathbf{Regret}(T) := \sum_{t=1}^{T} \mathbb{E}_{x_t \sim \rho}\big[\mathbb{E}_{a \sim \pi^{\star}(\cdot|x_t)}[R^{\star}(x_t, a)] - \mathbb{E}_{a \sim \pi_t(\cdot|x_t)}[R^{\star}(x_t, a)]\big].$$

### F.1 OFFLINE REGRESSION ORACLE

We now introduce the notion of an *offline regression oracle*. Given a reward function class $\mathcal{R}$, an offline regression oracle associated with $\mathcal{R}$, denoted by `OracleOff`, is a procedure that produces a predictor $\widehat{R} : \mathcal{X} \times \mathcal{A} \to \mathbb{R}$ based on input data. In statistical learning theory, the performance of $\widehat{R}$ is typically evaluated in terms of its *out-of-sample error*, that is, its expected error on random, unseen test data. Similar to online regression setting, we assume the statistical learning guarantees of `OracleOff`.

**Assumption F.2** (Guarantee of offline regression oracle). *Let $\pi : \mathcal{X} \to \Delta(\mathcal{A})$ be an arbitrary policy. Given $n$ training samples $(x_{1:n}, a_{1:n}, r_{1:n})$ where $x_i \sim \rho$ and $a_i \sim \pi(\cdot|x_i)$ i.i.d., the offline regression oracle `OracleOff` returns a reward estimator $\widehat{R} : \mathcal{X} \times \mathcal{A} \to \mathbb{R}$. For any $\delta > 0$, with probability at least $1 - \delta$, we have*

$$\mathbb{E}_{x \sim \rho, a \sim \pi(\cdot|x)}\left[\left(\widehat{R}(x, a) - R^{\star}(x, a)\right)^2\right] \leqslant \mathcal{E}_{\delta}(n).$$

Under the realizability assumption (Assumption 1), this squared distance corresponds to the estimation error or excess risk of $\widehat{R}$.

### F.2 ALGORITHM AND RESULTS

We provide an algorithm `KL-EXP-Off` in Algorithm F.1. Unlike Algorithm 1, which updates the predictor at every round, `KL-EXP-Off` adopts an epoch-based learning protocol, updating the reward estimator only once per epoch via the offline regression oracle. In addition, rather than feeding all past data into the oracle, we restrict its input to the data collected in the immediately preceding epoch $(m - 1)$. As a consequence of this strategy, the algorithm proceeds in gradually increasing epochs, i.e., $\tau_m = 2^m$.

Let $m(T)$ denote the total number of epochs. We then establish the following regret bound under the offline regression oracle.

**Theorem F.1** (Regret of `KL-EXP-Off`). *Consider an epoch schedule $\tau_m = 2^m$ for $m \leqslant m(T)$. Then, Under Assumption 1, 2, and F.2, with probability at least $1 - \delta$, the regret of `KL-EXP-Off` is bounded by*

$$\mathbf{Regret}_{\mathrm{KL}}(T, \eta) = \mathcal{O}\big(\eta \mathcal{E}_{\delta/\log T}(T) \cdot T\big), \quad \text{and}$$

$$\mathbf{Regret}(T) = \mathcal{O}\left(\eta \mathcal{E}_{\delta/\log T}(T) \cdot T + \frac{DT}{\eta}\right),$$

*where $D := \frac{1}{T}\sum_{t=1}^{T} \mathrm{KL}\big(\pi^{\star}(\cdot\|x_t)\|\pi_{\mathrm{ref}}(\cdot\|x_t)\big)$.*

**Remark F.1** (Computational efficiency). *The algorithm `KL-EXP-Off` requires only $\mathcal{O}(\log T)$ calls to the offline regression oracle.*

---

**Algorithm F.1** `KL-EXP-Off`

---

1: **Inputs:** regularization parameter $\eta$, reference policy $\pi_{\text{ref}}$, offline regression oracle `OracleOff`, epoch schedule $0 = \tau_0 < \tau_1 < \tau_2 < \cdots$.
2: **Initialize:** choose any $\widehat{R}_1 \in \mathcal{R}$.
3: **for** epoch $m = 1, 2, \ldots, m(T)$ **do**
4:     **for** round $t = \tau_{m-1} + 1, \cdots, \tau_m$ **do**
5:         Observe context $x_t \in \mathcal{X}$.
6:         Compute policy $\pi_t(\cdot|x_t) \propto \pi_{\text{ref}}(\cdot|x_t) \exp\big(\eta \widehat{R}_m(x_t, \cdot)\big)$ via Equation 2.
7:         Sample action $a_t \sim \pi_t(\cdot|x_t)$ and receive reward $r_t$.
8:     **end for**
9:     Feed *only* the data in epoch $m - 1$ into `OracleOff` and obtain $\widehat{R}_{m+1}$ .
10: **end for**

---

**Example F.1** (Linear classes). *When Assumption 1 holds and the reward function class $\mathcal{R}$ is linear (refer Example 1), by using the least squares regression oracle, `KL-EXP-Off` achieves $\mathbf{Regret}_{\text{KL}}(T, \eta) = \mathcal{O}(\eta d \log T)$ and $\mathbf{Regret}(T) = \mathcal{O}\big(\sqrt{dDT \log T}\big)$, with the choice $\eta = \Theta\left(\sqrt{\frac{DT}{d \log T}}\right)$. Moreover, by setting $\pi_{\text{ref}}$ to be uniform random, we have $\mathbf{Regret}(T) = \mathcal{O}\big(\sqrt{dT \log N \log T}\big)$ since $D \leqslant \log N$. This upper bound matches the lower bound $\Omega\big(\sqrt{dT \log N \log(T/d)}\big)$ established by Li et al. (2019), up to logarithmic $d$ factors.*

**Example F.2** (Neural Networks). *Let Assumption 1 hold and $\mathcal{R} = \mathcal{G}^N$, where $\mathcal{G}$ denotes the class of Multi-Layer Perceptrons (MLPs) as described in Section 2.1 of Farrell et al. (2021). For each $(x, a) \in \mathcal{X} \times \mathcal{A}$, let the reward function be $R^\star(x, a) = g_a^\star(x)$. Assume the context distribution $\rho$ is continuous over $[-1, 1]^d$, and that $g_1^\star, \ldots, g_N^\star$ lie in a Sobolev ball with smoothness $\beta \in \mathbb{N}$. Then, by Theorem 1 of Farrell et al. (2021), the deep MLP-ReLU network estimator attains $\mathcal{O}\left(n^{-\frac{\beta}{\beta+d}}\right)$ estimation error. Consequently, by using this estimator as the offline regression oracle, `KL-EXP-Off` achieves $\mathbf{Regret}_{\text{KL}}(T, \eta) = \tilde{\mathcal{O}}\left(\eta T^{\frac{d}{\beta+d}}\right)$ and $\mathbf{Regret}(T) = \tilde{\mathcal{O}}\left(T^{\frac{\beta+2d}{2\beta+2d}}\right)$ (ignoring dependence on other parameters) with the parameter choice $\eta = \tilde{\Theta}\left(T^{\frac{\beta}{2\beta+2d}}\right)$. Our derived unregularized regret, $\tilde{\mathcal{O}}\left(T^{\frac{\beta+2d}{2\beta+2d}}\right)$, has the same order as the regret established by Simchi-Levi & Xu (2022).*

### F.3   Main Proof of Theorem F.1

In this subsection, we present the proof of Theorem F.1.

*Proof of Theorem F.1.* For any $t \in [T]$, by Lemma B.2, we have

$$\mathbf{Regret}_{\text{KL}}(T, \eta) = \sum_{t=1}^{T} \mathbb{E}_{x_t \sim \rho} \left[ J_t^\eta(\pi_\eta^\star, R^\star) - J_t^\eta(\pi_t, R^\star) \right]$$

$$\leqslant \eta \sum_{t=1}^{T} \mathbb{E}_{x_t \sim \rho} \mathbb{E}_{a_t \sim \pi_t(\cdot|x_t)} \left[ \left( \widehat{R}_m(x_t, a_t) - R^\star(x_t, a_t) \right)^2 \right] \qquad \text{(Lemma B.2)}$$

Let $\mathcal{F}_t := \sigma(x_1, a_1, r_1, \ldots, x_t, r_t, a_t)$ be the filtration up to round $t$. We introduce the following lemma to further bound the regret.

**Lemma F.1** (Lemma 2 of Simchi-Levi & Xu 2022). *For all $m \geqslant 2$ and all $t \in \{\tau_{m-2}+1, \cdots, \tau_{m-1}\}$, with probability at least $1 - \delta/(2m^2)$, we have*

$$\mathbb{E}_{x_t \sim \rho, a_t \sim \pi_t(\cdot|x_t)} \left[ \left( \widehat{R}_m(x_t, a_t) - R^\star(x_t, a_t) \right)^2 \mid \mathcal{F}_{t-1} \right] \leqslant \mathcal{E}_{\delta/(2m^2)}(\tau_{m-1} - \tau_{m-2}).$$

By applying Lemma F.1, with probability $1 - \delta$, we obtain

$$
\mathbf{Regret}_{\mathrm{KL}}(T, \eta) \leqslant \eta \sum_{t=1}^{T} \mathbb{E}_{x_t \sim \rho} \mathbb{E}_{a_t \sim \pi_t(\cdot | x_t)} \left[ \left( \widehat{R}_{m(t)}(x_t, a_t) - R^\star(x_t, a_t) \right)^2 \right]
$$

$$
= \eta \sum_{t=1}^{T} \mathbb{E}_{x_t \sim \rho} \mathbb{E}_{a_t \sim \pi_t(\cdot | x_t)} \left[ \left( \widehat{R}_{m(t)}(x_t, a_t) - R^\star(x_t, a_t) \right)^2 \mid \mathcal{F}_{t-1} \right]
$$

$$
\leqslant \eta \sum_{t=\tau_1+1}^{T} \mathcal{E}_{\delta/(2m(t)^2)}(\tau_{m(t)-1} - \tau_{m(t)-2}) + \tau_1
$$

$$
= \eta \sum_{m=2}^{m(T)} \mathcal{E}_{\delta/(2m^2)}(\tau_{m-1} - \tau_{m-2}) \cdot (\tau_m - \tau_{m-1}) + \tau_1
$$

$$
= \mathcal{O}\left( \eta \mathcal{E}_{\delta/\log T}(T) \cdot T \right).
$$

This completes the proof of the upper bound on the KL-regularized regret. Moreover, the bound for the unregularized regret follows directly from the same analysis as in the proof of Theorem 1. $\qquad\square$

## G  Technical Lemmas

**Lemma G.1** (Freedman's inequality, Freedman, 1975). *Let $(Z_t)_{t \leqslant T}$ be a real-valued martingale difference sequence adapted to a filtration $\mathcal{F}_{t-1}$, and let $\mathbb{E}_t[\cdot] = \mathbb{E}[\cdot \mid \mathcal{F}_{t-1}]$. If $|Z_t| \leqslant B$ almost surely, then for any $\beta \in (0, 1/B)$, it holds that, with probability at least $1 - \delta$,*

$$
\sum_{t=1}^{T} Z_t \leqslant \beta \sum_{t=1}^{T} \mathbb{E}_{t-1}[Z_t^2] + \frac{B \log(1/\delta)}{\beta}.
$$

**Lemma G.2** (Lemma A.3 of Foster et al. 2021). *Let $(X_t)_{t \leqslant T}$ be a sequence of random variables adapted to a filtration $(\mathcal{F}_t)_{t \leqslant T}$. If $0 \leqslant X_t \leqslant B$ almost surely, then with probability at least $1 - \delta$,*

$$
\sum_{t=1}^{T} X_t \leqslant \frac{3}{2} \sum_{t=1}^{T} \mathbb{E}_{t-1}[X_t] + 4B \log \frac{2}{\delta}, \quad \text{and} \quad \sum_{t=1}^{T} \mathbb{E}_{t-1}[X_t] \leqslant 2 \sum_{t=1}^{T} X_t + 8B \log \frac{2}{\delta}.
$$

**Lemma G.3** (Lemma A.4 of Foster et al. 2021). *For any sequence of real-valued random variables $(X_t)_{t \leqslant T}$ adapted to a filtration $(\mathcal{F}_t)_{t \leqslant T}$, it holds that with probability at least $1 - \delta$, for all $T' \leqslant T$,*

$$
\sum_{t=1}^{T'} X_t \leqslant \sum_{t=1}^{T'} \log \left( \mathbb{E}_{t-1}[e^{X_t}] \right) + \log \frac{1}{\delta}.
$$

## H  Additional Experimental Results

### H.1  Additional Results on Linear Contextual Bandit Experiments

#### H.1.1  Computational Cost in Linear Contextual Bandits

| $N$ | $d$ | LinUCB | LinTS | LinPHE | SupLinUCB | KL-EXP (ours) |
|-----|-----|--------|-------|--------|-----------|---------------|
| 50  | 5   | 0.321  | 0.274 | 0.862  | 0.203     | **0.173**     |
| 100 | 5   | 0.465  | 0.336 | 0.927  | 0.225     | **0.190**     |
| 50  | 20  | 1.414  | 1.504 | 1.877  | 1.274     | **1.227**     |
| 100 | 20  | 1.616  | 1.546 | 1.942  | 1.378     | **1.253**     |

Table H.1: Average per-round computation time (μs) for linear bandits.

### H.1.2 ABLATION STUDY ON $\eta$ IN LINEAR CONTEXTUAL BANDITS

| $d$ | $N$ | KL-EXP ($\eta$) | | | | | LinUCB | LinTS | LinPHE | SupLinUCB |
|---|---|---|---|---|---|---|---|---|---|---|
| | | $0.2\eta^\star$ | $0.5\eta^\star$ | $\eta^\star$ | $2\eta^\star$ | $5\eta^\star$ | | | | |
| 5 | 50 | 596.37 | 367.31 | 244.52 | **222.57** | 267.98 | 302.06 | 440.90 | 602.85 | 1486.69 |
| | | ±112.63 | ±129.12 | ±78.35 | ±67.61 | ±88.32 | ±45.40 | ±73.82 | ±63.90 | ±636.21 |
| 5 | 100 | 508.08 | 410.16 | **238.09** | 267.38 | 320.29 | 297.72 | 417.66 | 594.41 | 1497.95 |
| | | ±131.46 | ±152.76 | ±77.78 | ±213.09 | ±106.25 | ±33.71 | ±64.97 | ±71.29 | ±641.16 |
| 20 | 50 | 541.04 | 342.24 | 329.34 | **321.84** | 340.00 | 478.25 | 584.17 | 614.89 | 1105.45 |
| | | ±227.71 | ±105.33 | ±40.41 | ±70.77 | ±76.04 | ±113.83 | ±182.24 | ±207.34 | ±416.75 |
| 20 | 100 | 684.46 | 416.29 | **361.01** | 379.26 | 400.35 | 443.73 | 575.69 | 622.88 | 1104.46 |
| | | ±212.17 | ±108.76 | ±55.66 | ±135.18 | ±106.67 | ±80.81 | ±177.43 | ±212.86 | ±420.46 |

Table H.2: Average cumulative regret at the final round $T = 5000$, with standard deviations (small font), under varying regularization parameters $\eta$ in linear contextual bandits. Here, $\eta^\star = \sqrt{T \log N / (2d \log T + 16 \log(1/\delta))}$ denotes the theoretically optimal choice proposed in Theorem 1.

## H.2 ADDITIONAL RESULTS ON NEURAL BANDIT EXPERIMENTS

### H.2.1 COMPUTATION COST IN NEURAL BANDITS

| NeuralUCB | NeuralTS | KL-EXP (ours) |
|---|---|---|
| 0.0507 | 0.0665 | **0.0048** |

Table H.3: Average per-round computation time (s) for neural bandits.

### H.2.2 ABLATION STUDY ON $\eta$ IN NEURAL BANDITS

| Reward Function | KL-EXP ($\eta$) | | | | | | NeuralUCB | NeuralTS |
|---|---|---|---|---|---|---|---|---|
| | 50 | 100 | 500 | 1000 | 3000 | 5000 | | |
| Linear | 52.48 | 27.17 | **19.49** | 20.05 | 20.59 | 21.96 | 29.56 | 31.61 |
| | ±2.01 | ±1.55 | ±1.12 | ±1.23 | ±1.52 | ±1.82 | ±2.67 | ±2.85 |
| Quadratic | 134.61 | 70.89 | 51.57 | 50.61 | **46.16** | 48.47 | 142.59 | 108.89 |
| | ±3.65 | ±2.29 | ±6.44 | ±4.88 | ±5.89 | ±4.12 | ±16.75 | ±6.36 |
| Cosine | 211.67 | 210.07 | 207.85 | **204.95** | 210.89 | 215.84 | 246.58 | 250.42 |
| | ±7.69 | ±6.12 | ±6.51 | ±9.72 | ±9.62 | ±10.02 | ±6.73 | ±6.77 |
| Neural Network | 139.10 | 83.55 | 54.76 | **53.75** | 53.92 | 58.58 | 79.43 | 68.96 |
| | ±2.35 | ±1.78 | ±1.24 | ±1.63 | ±1.53 | ±2.24 | ±4.27 | ±1.80 |

Table H.4: Average cumulative regret at the final round $T = 4000$, with standard deviations (small font), under varying regularization parameters $\eta$ in neural bandits.

## H.3 RLHF EXPERIMENTS: DETAILS AND ADDITIONAL RESULTS

In this section, we present the RLHF experimental setup in detail and provide additional results.

**Implementation details.** For fair comparison, we follow the experimental setup of Dong et al. (2024); Xie et al. (2024). In each iteration, we fix the base model (Llama-3-8B-Flow-SFT) as the reference model $\pi_{\text{ref}}$ and set the regularization parameter to $\eta = 10.0$. Training is performed with a global batch size of 16, a learning rate of $5 \times 10^{-7}$ with cosine scheduling, 2 epochs per iteration, and a warmup ratio of 0.03. For XPO, following Xie et al. (2024), we set $\tilde{\pi}_t = \pi_t$ and $\mathcal{D}_t^{\text{opt}} = \mathcal{D}_t^{\text{pref}}$, and use their exploration schedule $\alpha \in \{1 \times 10^{-5}, 5 \times 10^{-6}, 0\}$ across the three iterations(see their definitions). All experiments were conducted on $8 \times$ Nvidia H100 GPUs.

We train XPO (Xie et al., 2024) and OnlineDPO using three random seeds and report the mean and standard error of their average accuracy across 17 benchmarks to ensure statistical reliability. For the

baselines Llama-3-8B-Flow-SFT ($\pi_{\text{ref}}$) and Llama-3-8B-Flow-Final (Dong et al., 2024), we directly evaluate the pretrained models released on Hugging Face, so training randomness is not reported for these two baselines.

**Full benchmark results.** Table H.5 reports the accuracies of the algorithms on all 17 academic and chat benchmarks (Zhong et al., 2023; Nie et al., 2019; Hendrycks et al., 2020; Cobbe et al., 2021; Rein et al., 2024; Chen et al., 2021; Zellers et al., 2019; Sakaguchi et al., 2021; Clark et al., 2018; Lin et al., 2021; Mihaylov et al., 2018; Zellers et al., 2018; Sap et al., 2019; Pilehvar & Camacho-Collados, 2018; Levesque et al., 2012; Socher et al., 2013), as well as the performance of `OnlineDPO` (or `ODPO`) with varying regularization parameters $\eta \in \{5.0, 8.5, 10.0, 12.5, 20.0\}$. The **bold** values represent the best performance for each benchmark. The results show that `OnlineDPO` with a carefully chosen $\eta$ ($= 12.5$) outperforms other baselines that rely on additional exploration techniques.

**Robustness to sampling temperature.** We evaluate the performance of models produced by different alignment algorithms across a range of sampling temperatures. We also report the win rates (%) computed by GPT-4o-mini (Hurst et al., 2024) on the RLHFlow test dataset[9] , comparing each model against the reference policy (Llama-3-8B-Flow-SFT). In Figure H.1, the results indicate that OnlineDPO with $\eta = 12.5$ outperforms the other baselines across the sampling temperatures $\tau \in \{0.5, 0.7, 1.0\}$. Moreover, we observe that OnlineDPO achieves its highest win rate at $\tau = 1.0$, whereas the other baselines perform best at $\tau \in \{0.5, 0.7\}$. This behavior is, however, consistent with our theoretical framework: the policy is trained at $\tau = 1.0$, and the regret is also defined with respect to the $\tau = 1.0$ policy. In other words, the primary objective is to minimize regret for the policy corresponding to $\tau = 1.0$, making this outcome expected.

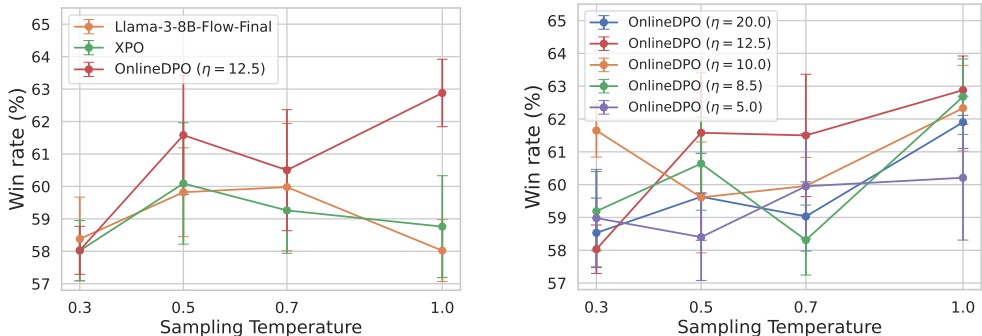

Figure H.1: The frontier of the ground-truth reward reward vs KL to the reference policy.

**Reward vs. KL to the reference policy.** We additionally report the reward, evaluated by the ground-truth reward model against the KL divergence at the end of each iteration. The figure H.2 shows that that OnlineDPO achieves the most efficient frontier—obtaining the highest reward while keeping the KL divergence small.

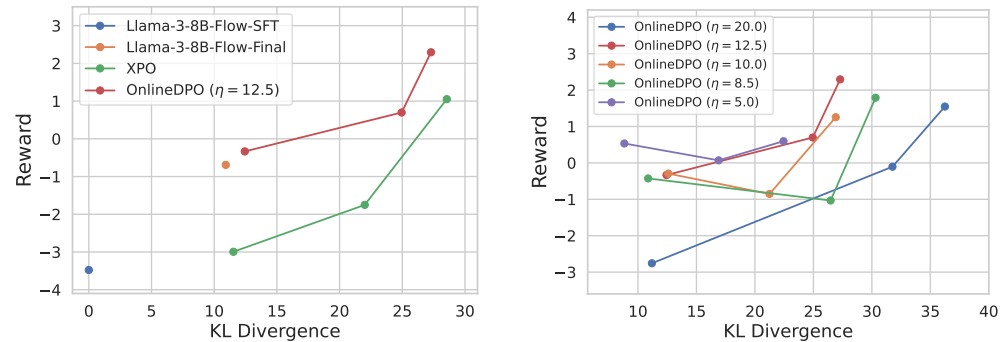

Figure H.2: The Reward-KL trade-off curves.

---

[9]`https://huggingface.co/datasets/RLHFlow/test_generation_2k`

| Model | $\eta$ | iteration | AGIEval | ANLI | MMLU | GSM8K | GPQA | HumanEval | HellaSwag | WinoGrande | ARC-C |
|---|---|---|---|---|---|---|---|---|---|---|---|
| Llama-3-8B-Flow-SFT | 10.0 | | 39.33 | 40.51 | 62.63 | 74.15 | 34.34 | 54.27 | 59.89 | 76.48 | 53.50 |
| Llama-3-8B-Flow-Final | 10.0 | | **41.75** | 46.29 | 63.36 | 74.75 | 31.31 | 54.88 | 61.22 | 76.95 | 52.73 |
| XPO | 10.0 | iter 1 | 39.33 | 43.74 | 63.13 | 80.14 | 33.33 | 57.11 | 62.16 | 75.82 | 56.60 |
| | | | ±0.007 | ±0.147 | ±0.084 | ±0.347 | ±0.505 | ±0.931 | ±0.036 | ±0.199 | ±0.178 |
| | | iter 2 | 40.01 | 47.80 | 63.34 | 80.31 | 31.14 | 58.64 | 62.48 | 76.16 | 56.31 |
| | | | ±0.089 | ±0.350 | ±0.079 | ±0.266 | ±0.292 | ±0.976 | ±0.075 | ±0.158 | ±0.443 |
| | | iter 3 | 40.35 | 46.43 | **63.46** | 81.91 | 33.16 | 58.94 | 62.94 | **77.01** | **56.83** |
| | | | ±0.259 | ±0.316 | ±0.083 | ±0.904 | ±0.292 | ±0.931 | ±0.095 | ±0.456 | ±0.256 |
| OnlineDPO | 5.0 | iter 1 | 39.47 | 45.70 | 63.19 | 81.32 | 32.83 | 56.71 | 62.31 | 76.22 | 56.11 |
| | | | ±0.154 | ±0.258 | ±0.128 | ±0.306 | ±1.010 | ±2.199 | ±0.333 | ±0.254 | ±0.130 |
| | | iter 2 | 40.17 | 46.68 | 63.24 | 83.04 | **34.51** | 57.93 | 62.82 | 76.19 | 56.09 |
| | | | ±0.354 | ±1.296 | ±0.253 | ±1.161 | ±1.458 | ±0.610 | ±0.397 | ±0.091 | ±0.793 |
| | | iter 3 | 40.52 | 47.26 | 63.23 | 82.59 | 33.00 | 58.74 | 63.08 | 76.35 | 56.40 |
| | | | ±0.325 | ±1.276 | ±0.080 | ±0.219 | ±1.051 | ±1.269 | ±0.546 | ±0.329 | ±0.597 |
| | 8.5 | iter 1 | 39.65 | 45.44 | 63.33 | 81.67 | 31.66 | 57.58 | 62.61 | 76.22 | 56.14 |
| | | | ±0.179 | ±0.796 | ±0.095 | ±0.368 | ±0.282 | ±0.458 | ±0.141 | ±0.228 | ±0.224 |
| | | iter 2 | 40.33 | 47.72 | 63.34 | 82.89 | 33.00 | 58.03 | 63.20 | 76.06 | 55.66 |
| | | | ±0.259 | ±0.924 | ±0.213 | ±1.595 | ±0.583 | ±0.187 | ±0.268 | ±0.182 | ±0.485 |
| | | iter 3 | 40.53 | 48.90 | 63.38 | 82.82 | 33.33 | 59.76 | 63.48 | 76.40 | 55.69 |
| | | | ±0.215 | ±0.341 | ±0.079 | ±0.382 | ±1.010 | ±1.829 | ±0.219 | ±0.285 | ±0.130 |
| | 10.0 | iter 1 | 39.47 | 45.00 | 63.34 | 81.87 | 31.99 | 57.78 | 62.66 | 76.06 | 56.08 |
| | | | ±0.105 | ±0.790 | ±0.082 | ±0.258 | ±0.764 | ±0.415 | ±0.095 | ±0.046 | ±0.174 |
| | | iter 2 | 40.40 | 48.03 | 63.37 | 82.74 | 32.83 | 57.72 | 63.29 | 76.16 | 55.57 |
| | | | ±0.219 | ±0.808 | ±0.207 | ±1.630 | ±0.505 | ±0.352 | ±0.180 | ±0.000 | ±0.394 |
| | | iter 3 | 40.74 | **48.91** | 63.32 | 83.07 | 32.83 | 58.13 | 63.58 | 76.22 | 55.83 |
| | | | ±0.284 | ±0.352 | ±0.127 | ±0.389 | ±0.505 | ±0.352 | ±0.244 | ±0.164 | ±0.261 |
| | 12.5 | iter 1 | 39.57 | 45.86 | 63.26 | 81.75 | 31.14 | **59.96** | 62.75 | 76.16 | 55.97 |
| | | | ±0.077 | ±0.215 | ±0.009 | ±0.438 | ±1.166 | ±0.352 | ±0.080 | ±0.137 | ±0.148 |
| | | iter 2 | 40.33 | 47.80 | 63.16 | **84.00** | 32.49 | 59.55 | 63.42 | 76.87 | 55.12 |
| | | | ±0.220 | ±0.258 | ±0.143 | ±0.330 | ±1.166 | ±0.931 | ±0.072 | ±0.158 | ±0.171 |
| | | iter 3 | 40.81 | 48.55 | 63.26 | 83.37 | 33.00 | 58.33 | 63.72 | 76.59 | 55.52 |
| | | | ±0.153 | ±0.358 | ±0.095 | ±0.358 | ±1.543 | ±0.931 | ±0.177 | ±0.389 | ±0.215 |
| | 20.0 | iter 1 | 39.70 | 45.98 | 63.27 | 82.56 | 31.99 | 57.93 | 62.94 | 76.16 | 55.86 |
| | | | ±0.104 | ±0.353 | ±0.175 | ±0.273 | ±0.583 | ±1.613 | ±0.041 | ±0.158 | ±0.099 |
| | | iter 2 | 40.38 | 47.40 | 63.18 | 83.34 | 32.32 | 58.94 | 63.57 | 76.51 | 54.52 |
| | | | ±0.299 | ±0.370 | ±0.047 | ±1.031 | ±0.875 | ±0.931 | ±0.106 | ±0.690 | ±0.644 |
| | | iter 3 | 40.90 | 47.30 | 63.37 | 83.47 | 31.99 | 58.33 | **63.80** | 76.69 | 55.29 |
| | | | ±0.168 | ±0.870 | ±0.168 | ±0.263 | ±0.582 | ±1.763 | ±0.047 | ±0.501 | ±0.823 |

| Model | $\eta$ | iteration | ARC-E | TruthfulQA | OpenBookQA | SWAG | Social IQa | WiC | WSC273 | SST-2 | **Average** |
|---|---|---|---|---|---|---|---|---|---|---|---|
| Llama-3-8B-Flow-SFT | 10.0 | | 83.33 | 45.38 | 35.40 | 58.07 | 52.35 | 56.74 | 87.55 | 90.94 | 59.11 |
| Llama-3-8B-Flow-Final | 10.0 | | 81.94 | 53.71 | 37.20 | 58.15 | 52.10 | 62.54 | 87.18 | 91.97 | 60.47 |
| XPO | 10.0 | iter 1 | 84.10 | 48.81 | 37.27 | 59.30 | **54.32** | **63.53** | 87.91 | 90.60 | 61.01 |
| | | | ±0.064 | ±0.344 | ±0.115 | ±0.040 | ±0.550 | ±0.001 | ±0.115 | | ±0.063 |
| | | iter 2 | 84.25 | 51.70 | 37.87 | 59.63 | 53.46 | 61.91 | 87.06 | 90.56 | 61.33 |
| | | | ±0.064 | ±0.331 | ±0.115 | ±0.033 | ±0.565 | ±0.560 | ±0.066 | | ±0.013 |
| | | iter 3 | 83.94 | 52.67 | **38.07** | 59.88 | 53.09 | 59.87 | 88.03 | 90.71 | 61.61 |
| | | | ±0.064 | ±0.433 | ±0.231 | ±0.053 | ±0.107 | ±1.659 | ±0.211 | ±0.115 | ±0.044 |
| OnlineDPO | 5.0 | iter 1 | 84.41 | 50.13 | 37.11 | 59.19 | 53.29 | 62.55 | 87.76 | 90.32 | 61.10 |
| | | | ±0.175 | ±1.004 | ±1.188 | ±0.400 | ±0.590 | ±0.602 | ±1.173 | ±0.138 | ±0.144 |
| | | iter 2 | 84.26 | 52.34 | 36.85 | 59.59 | 53.24 | 61.88 | 88.34 | 90.66 | 61.64 |
| | | | ±0.437 | ±0.422 | ±1.418 | ±0.853 | ±1.040 | ±0.770 | ±0.879 | ±1.502 | ±0.028 |
| | | iter 3 | 84.09 | 54.03 | 36.35 | 59.60 | 53.19 | 62.65 | **89.58** | 91.60 | 61.90 |
| | | | ±0.547 | ±0.687 | ±1.340 | ±0.748 | ±0.419 | ±0.246 | ±0.437 | ±0.513 | ±0.068 |
| | 8.5 | iter 1 | 84.38 | 51.86 | 37.26 | 59.51 | 53.17 | 62.57 | 88.22 | 90.74 | 61.29 |
| | | | ±0.218 | ±0.453 | ±0.245 | ±0.021 | ±1.632 | ±0.680 | ±0.171 | ±0.138 | ±0.051 |
| | | iter 2 | 83.98 | 54.32 | 37.27 | 59.85 | 52.64 | 62.19 | 88.32 | 91.33 | 61.77 |
| | | | ±0.310 | ±0.569 | ±0.231 | ±0.053 | ±0.680 | ±0.827 | ±0.802 | ±0.659 | ±0.061 |
| | | iter 3 | 83.67 | 55.53 | 36.94 | 59.80 | 52.51 | 61.52 | 88.97 | 91.41 | 62.04 |
| | | | ±0.443 | ±0.310 | ±1.318 | ±0.390 | ±0.307 | ±0.810 | ±1.032 | ±0.541 | ±0.141 |
| | 10.0 | iter 1 | **84.49** | 52.07 | 37.26 | 59.50 | 53.05 | 62.36 | 88.22 | 90.78 | 61.29 |
| | | | ±0.172 | ±0.180 | ±0.245 | ±0.019 | ±1.559 | ±0.784 | ±0.171 | ±0.142 | ±0.073 |
| | | iter 2 | 83.99 | 54.47 | 37.20 | 59.89 | 52.56 | 61.93 | 88.69 | 91.18 | 61.77 |
| | | | ±0.319 | ±0.658 | ±0.200 | ±0.046 | ±0.544 | ±0.859 | ±0.437 | ±0.648 | ±0.059 |
| | | iter 3 | 83.53 | 56.18 | 37.40 | 60.01 | 52.29 | 61.58 | 88.69 | 91.69 | 62.00 |
| | | | ±0.353 | ±0.287 | ±0.393 | ±0.121 | ±0.078 | ±0.799 | ±0.802 | ±0.121 | ±0.109 |
| | 12.5 | iter 1 | 84.26 | 52.33 | 37.27 | 59.64 | 53.10 | 62.59 | 88.40 | 90.90 | 61.47 |
| | | | ±0.219 | ±0.222 | ±0.231 | ±0.012 | ±0.059 | ±0.905 | ±0.423 | ±0.066 | ±0.039 |
| | | iter 2 | 83.64 | 55.12 | 36.80 | 59.99 | 52.34 | 62.12 | 89.01 | 91.78 | 61.97 |
| | | | ±0.088 | ±0.265 | ±0.200 | ±0.043 | ±0.156 | ±0.394 | ±0.366 | ±0.132 | ±0.116 |
| | | iter 3 | 83.17 | 56.53 | 37.13 | 60.26 | 52.00 | 62.54 | 89.26 | 92.35 | **62.14** |
| | | | ±0.042 | ±0.147 | ±0.231 | ±0.266 | ±0.051 | ±0.313 | ±0.423 | ±0.132 | ±0.121 |
| | 20.0 | iter 1 | 84.08 | 52.83 | 37.09 | 59.61 | 52.88 | 63.01 | 88.44 | 91.00 | 61.49 |
| | | | ±0.064 | ±0.853 | ±0.103 | ±0.166 | ±0.341 | ±1.496 | ±0.511 | ±0.093 | ±0.065 |
| | | iter 2 | 83.24 | 56.11 | 37.00 | 59.94 | 51.89 | 61.46 | 89.09 | 92.01 | 61.82 |
| | | | ±0.479 | ±0.715 | ±0.400 | ±0.382 | ±0.263 | ±0.840 | ±0.145 | ±0.174 | ±0.165 |
| | | iter 3 | 82.93 | **56.87** | 37.17 | **60.34** | 51.58 | 62.47 | 89.26 | **92.57** | 62.02 |
| | | | ±0.274 | ±0.501 | ±0.058 | ±0.150 | ±0.195 | ±1.019 | ±0.560 | ±0.332 | ±0.319 |

Table H.5: Full benchmark evaluation of `OnlineDPO` with varying $\eta \in \{5.0, 8.5, 10.0, 12.5, 20.0\}$ and of other algorithms that use additional exploration strategies. **Bold** values indicate the best performance. Smaller font indicates standard deviation over three random seeds.

