# OpenReview forum: "KL-Regularization Is Sufficient in Contextual Bandits and RLHF"
_ICLR.cc/2026/Conference — ICLR 2026 Conference Withdrawn Submission_

### Official Review · Reviewer_77Wy · 2025-10-28

**Soundness:** 3
**Presentation:** 3
**Contribution:** 3
**Rating:** 8
**Confidence:** 4

**Summary:**

The paper considers the problem of contextual bandits and RLHF, in which they consider the class of algorithms that choose an action based on a policy that maximizes KL-regularized cumulative reward. This has become highly popular in recent times particuarly for RLHF. While there are several such algorithms proposed in literature that fall under this class, all of them use some sort of forced exploration to address the exploration-exploitation trade-off. In this work, the authors state and prove that no additional exploration term is required and the policies that are obtained by maximizing KL-regularized cumulative reward achieve optimal regret. The authors prove their claims for both contextual bandits and RLHF. They also corroborate their theoretical findings via extensive empirical studies on both synthetic and real-world tasks.

**Strengths:**

I think the paper is solid and delivers a simple yet neat result, that can be useful for several applications. The key result in this work is where the authors show that the instantaneous (KL-regularized) regret is is bounded by the squared error on the reward estimation process. This provides a nice bridge between regret minimization via KL-regularized policies and reward estimation, which is typical in online learning.

**Weaknesses:**

I think the reader would strongly benefit from a discussion about **why** this approach works? There is obvious a mathematical description as to why it works, but an intuitive description would definitely help improve the paper and make it more accessible to readers who do not want to go into the technical details.

**Questions:**

In addition to the above question regarding an intuitive explanation, I have some follow-up questions:

- In the "exact second-order" Taylor expansion, how and why does the factor $(1 - \alpha)$ appear in the integral in Eqn. 5? It might just be a different way of writing things but I do not recall seeing such an expression.

- What happens when the context and actions spaces are infinite? While they are indeed finite for RLHF, it may not be so for other contextual bandit problems. How does your approach scale in such situations?

- Do you think this methodology can be combined with lazy update schemes to reduce computational costs?

---

> ### Author Response · Authors · 2025-11-22
>
> Thank you for the strong support and the insightful feedback.
>
> ---
>
> ### **Intuitive explanation of KL-regularization**
> Thank you for bringing up this point. We appreciate the opportunity to explain the underlying intuition behind KL-regularization.
>
> In this paper, we show that **KL-regularization alone is sufficient to induce adequate exploration**, eliminating the need for additional methods such as UCB or Thompson Sampling. When the reference policy is uniform random, this regularization-based exploration approach shares a similar spirit with the *Follow-the-Regularized-Leader (FTRL)* framework with an entropic regularizer: both introduce a regularizer when optimizing the policy, leading to a Gibbs-style solution. We posit that KL-regularization can induce a **similar exploratory effect** to that of FTRL, effectively balancing exploitation and exploration through its implicit regularization structure.
>
>
>
> ---
>
> ### **Clarification of the $1-\alpha$ term in Equation (5)**
>
> We explicitly show why the $1-\alpha$ term appears in the exact second-order Taylor expansion. For simplicity, let $g(\alpha):= f(x, R^\star + \alpha (\hat{R}_t - R^\star))$. Then, we have
>
> $$
>     f(x, \hat{R}_t) - f(x, R^\star) = g(1) - g(0) = \int_0^1 g'(\alpha) d\alpha.
> $$
>
> Since, we can write $g'(\alpha) = g'(0) + \int_{0}^\alpha g''(s) d s$, integrating both sides over $\alpha \in [0,1]$ gives
>
> $$
>     \int_0^1 g'(\alpha) d\alpha
>     = g'(0) +  \int_0^1  \int_{0}^\alpha g''(s) d s d\alpha
>     = g'(0) +  \int_{0}^1 \int_s^1    g''(s) d\alpha d s
>     =  g'(0) +  \int_{0}^1 (1-s)   g''(s) d s,
> $$
>
> where the second-to-last equality holds by Fubini’s theorem, since we merely swap the order of integration over the triangular region $\{(\alpha,s): 0 \le s \le \alpha \le 1\}$. Renaming $s$ back to $\alpha$, we recover the exact second-order Taylor integral form shown in Equation (5).
>
>
> ---
>
> ### **Infinite context and action spaces**
>
> **Infinite context space:** For the context space, our framework indeed accommodates an infinite domain, since the policy can still be computed for any given context $x_t$.
>
> **Infinite action space:** The situation is different when the action space is infinite. In this case, the Gibbs policy is no longer available in closed form because the normalization constant $Z_R(x)$ becomes intractable. One would instead need to approximate the policy using methods such as Monte Carlo sampling or variational estimation. As long as the approximation is sufficiently accurate, analogous theoretical guarantees can still be obtained. For example, if the action space is a $d_A$-dimensional Euclidean ball and the reference policy is uniform, then $\mathrm{KL}(\pi^\star \\| \pi_{\text{ref}}) \lesssim d_A \log T$, implying that the $\sqrt{T}$-order unregularized regret bound continues to hold (and the regularized regret bound remains unaffected).
>
> On the other hand, our **DPO-variant (ODPO) can naturally handle infinite action spaces**, since it directly updates the policy without requiring exact computation of the Gibbs policy (Eqn. 2). In this setting, the policy can be updated using standard optimization methods such as SGD or Adam (Eqn. E.2). As long as the bound for the DPO online regression oracle remains sublinear, we can still achieve the desired guarantees (logarithmic regularized regret and $\sqrt{T}$-type unregularized regret).
>
> ---
>
> ### **Lazy update**
>
> Please refer to Appendix F, which shows that our method can be extended to the offline regression oracle setting while **requiring only $\log T$ oracle calls and policy updates (Algorithm F.1)**. Therefore, our approach can naturally be adapted to lazy update schemes, significantly reducing the computational cost.

---

### Official Review · Reviewer_DoSW · 2025-11-01

**Soundness:** 3
**Presentation:** 3
**Contribution:** 2
**Rating:** 4
**Confidence:** 3

**Summary:**

The paper studies KL-regularized contextual bandits and an RLHF variant. The authors propose an algorithm that plays policy via a single step of exponential update from the reference policy $\pi_t(\cdot | x) \propto \pi_{\mathrm{ref}}(\cdot | x) \exp(\eta \hat R_t(x, \cdot))$, where $\hat R_t(x, \cdot)$ is computed from an online regression oracle. The paper claims this is the first algorithm that removes additional exploration in this setting. They show that the algorithm achieves regret of $O(\eta Reg_{Sq}(T))$ against the KL-regularized policy, and an unregularized regret $O(\eta Reg_{Sq}(T) + DT / \eta)$, where $D = (1/T) \sum_t \mathrm{KL}(\pi^* \| \pi_{\mathrm{ref}})$. The key lemma uses a second-order expansion around the ground-truth $R^*$ to avoid UCB/optimism. Finally, experimental results are presented for both linear and neural bandits, as well as an online DPO pipeline.

**Strengths:**

The paper studies KL-regularized bandits, which is an important problem in reinforcement learning from human feedback. The algorithm is the first that removes the pure exploration phase in this setting, matching the desiderata that many researchers hope for in RLHF. Experiments are provided to demonstrate the practical effectiveness of the algorithm.

**Weaknesses:**

- The final bound of the algorithm is exactly a two-term trade-off between the stability $Reg_{Sq}(T)$ and the bias $\sum_t \mathrm{KL}(\pi^* \| \pi_{\mathrm{ref}})$, which mirrors the standard mirror-descent/FTRL analyses for entropic regularization. The central "no optimism" claim hinges on the second-order expansion, but the resulting decomposition and the KL-to-unregularized conversion are conceptually standard once written. From my perspective, the regularized step is itself another standard way of introducing exploration. As a result, I do not see enough conceptual distance from known adversarial-bandit templates to justify the novelty of the "KL is sufficient" claim.

- Even though the paper claims that the algorithm does not require prior knowledge of the eluder dimension as in previous works, it still requires prior knowledge of the oracle’s regret bound $Reg_{Sq}(T)$, which is in fact a similar complexity quantity that measures the hardness of online decision making. There is no parameter-free schedule to handle this issue in the theory.

**Questions:**

- Do you think it is possible to provide a parameter-free $\eta$ schedule that does not require any prior knowledge about the online regression oracle?

---

> ### Author Response · Authors · 2025-11-22
> **Official Comment by Authors (1/3)**
>
> We appreciate the time you took to review our paper and your valuable feedback.
>
> ---
>
> ### **Comparison to FTRL/OMD**
>
> Thank you for the insightful comment! We would like to clarify that **(1) our analysis is fundamentally different from the classical FTRL/OMD framework** (although there may be some conceptual connections, as both approaches induce exploration through regularization), and **(2) our framework applies to general function classes, which classical FTRL analyses do not address.** For this reason, any conceptual similarity between our approach and FTRL should be viewed as a **strength** of our work—not a weakness.
>
> **(1) vs. FTRL/OMD**
>
> - **Differences in objective and regret bounds (vs. FTRL):** First, we show that our objective and the resulting regret bound are fundamentally different from those in the classical FTRL/OMD framework.
>
>     In entropy-regularized FTRL, the policy is chosen by minimizing the *cumulative loss* (as a function of the policy $\pi$) plus an entropy term, e.g., $\sum_{s=1}^t \ell_s(\pi) + \frac{1}{\eta}\sum_{a}\pi(a) \ln \pi(a)$. When each $\ell_t(\pi)$ is convex, Corollary 7.7 in [1] shows that the resulting (unregularized) regret satisfies
>     $$
>         \textbf{Regret}(T) = \mathcal{O}\left( \frac{\log N}{\eta} + \eta \sum_{t=1}^T \\|g_t \\|_{\infty}^2 \right),
>     $$
>     for any $g_t \in \partial \ell_t(\pi_t)$, where $N$ is the number of arms.
>
>     In contrast, KL-regularized bandits optimize *only the current reward estimate*, not a sum of past losses, i.e., $\mathbb{E}\_{\pi}[\hat{R}_t(a)] - \frac{1}{\eta}\text{KL}(\pi \\|\pi\_{\text{ref}})$. And our Theorem 1 shows that
>
>     $$
>         \textbf{Regret}(T) =  \mathcal{O}\left(
>             \frac{\sum_{t=1}^T \text{KL}(\pi^\star \\| \pi\_{\text{ref}}) }{T \eta} + \eta \text{Reg}\_{\text{sq}}(T)
>         \right),
>     $$
>     and when $\pi\_{\text{ref}}$ is uniform,
>     $$
>         \textbf{Regret}(T) =  \mathcal{O}\left(
>             \frac{\log N}{ \eta} + \eta \text{Reg}\_{\text{sq}}(T)
>         \right).
>     $$

---

> ### Author Response · Authors · 2025-11-22
> **Official Comment by Authors (2/3)**
>
> &emsp; This comparison highlights three key differences:
>
>  &emsp; (a) **Objective**: FTRL optimizes an objective based on the *sum of cumulative losses*, whereas KL-regularization optimizes an objective based on the *current reward estimates*.
>
> &emsp; (b)  **Reference policy**: FTRL implicitly regularizes toward the *uniform* reference policy, whereas KL-regularization uses an *explicit reference policy* $\pi_{\text{ref}}$.
>
> &emsp; (c) $\mathbf{\text{Reg}\_{\text{sq}}(T) \neq \sum_{t=1}^T \\|g_t \\|\_{\infty}^2}$: Even though the bounds may look similar at first sight, the underlying quantities are not. $\mathrm{Reg}_{\mathrm{sq}}(T)$ comes from a *reward-regression oracle*, not from summing policy-gradient norms.
>
> - **Differences in objective and regret bounds (vs. OMD):** Moreover, unlike the standard OMD objective—where each update is taken with respect to the current policy $\pi_t$—KL-regularization uses a *fixed* reference policy $\pi_{\text{ref}}$ throughout the optimization. Consequently, the resulting regret bounds are also distinct (as in the FTRL case). This makes KL-regularization fundamentally different from OMD as well.
>
>
> - **Differences in analysis:** In our proof of the unregularized regret upper bound, we decompose the unregularized regret into the KL-regularized regret and an additional KL term, i.e., $\frac{1}{\eta} \sum_{t=1}^T \text{KL}(\pi^\star \\| \pi_{\text{ref}})$. Our key contribution is showing that the unregularized regret can be bounded **without any additional exploration**, yielding $\mathcal{O}(\eta \text{Reg}_{\text{sq}}(T))$. To the best of our knowledge, this type of decomposition—together with obtaining a logarithmic bound on the KL-regularized regret—does not appear in existing analyses of FTRL. Hence, our analysis is fundamentally different from that of FTRL/OMD.
>
>
> [1] Orabona, Francesco. "A modern introduction to online learning." arXiv preprint arXiv:1912.13213 (2019).
>
> ---
>
> ### **We did not claim that KL-regularization does not provide exploration.**
>
> We would like to clarify a potential misunderstanding of our main message. Our central claim is that KL-regularization is sufficient and already provides enough exploration when the parameter $\eta$ is properly tuned. We never state that KL-regularization does not provide exploration.
>
>
> ---
>
> ### **Parameter-free $\eta$ is already achievable**
>
> First, we emphasize that our algorithm **does not require knowing the regression oracle bound** in order to **achieve logarithmic regularized regret**. The optimal choice $\eta^* \approx \sqrt{DT/\text{Reg}_{\text{sq}}(T)}$ is used only to obtain the o*ptimal regularized regret*, not to run the algorithm. This stands in sharp contrast to existing approaches, such as Zhao et al. (2024, 2025), which must know complexity parameters like the eluder dimension to ensure optimism—quantities that are *unknown* in practice. This difference is crucial and highlights the practicality of our algorithm.

---

> ### Author Response · Authors · 2025-11-22
> **Official Comment by Authors (3/3)**
>
> Moreover, even for the **unregularized regret**, our algorithm **already admits a parameter-free bound** of $\tilde{\mathcal{O}}((D + \text{Reg}_{\text{sq}}(T))\sqrt{T})$, simply by setting $\eta = \sqrt{T}$. This directly yields the desired $\sqrt{T}$-order regret (although not the optimal rate). The choice of $\eta$ used in Theorem 1 serves only to obtain the *optimal* regret—it is NOT required to achieve $\sqrt{T}$-order regret.
>
>
> This behavior is fundamentally different from that of the conventional general function class bandit framework. In those settings, if the relevant complexity parameter (e.g., the eluder dimension) is unknown, the optimism event does not hold, and consequently one cannot guarantee even $\sqrt{T}$-order regret. In contrast, our analysis does not rely on any optimism-based analysis, meaning that we do not need prior knowledge of quantities such as  $\text{Reg}_{\text{sq}}(T)$. We view this as a significant advantage of our approach over existing general function class bandit algorithms.

---

### Official Review · Reviewer_5qSz · 2025-11-01

**Soundness:** 3
**Presentation:** 3
**Contribution:** 4
**Rating:** 6
**Confidence:** 3

**Summary:**

This is a clever and elegant idea: adding a KL-to-reference term with finite inverse temperature (η>0) yields a simple Gibbs policy and an exact second‑order expansion that shows KL itself induces exploration, turning regularized regret into a squared estimation‑error term without UCB/TS. However, weaknesses remain: the theory mainly covers single‑step bandits (not general MDPs), it relies on realizability and a nonzero temperature during training, and the experiments look underpowered—baseline methods like LinUCB/LinTS are sensitive to exploration parameters that should be thoroughly swept, η sensitivity/annealing should be reported, and the average accuracy gaps are small, warranting multi‑seed runs and confidence intervals.

**Strengths:**

Strong, elegant theory for single-step settings: The “exact second-order expansion” shows that KL-to-reference (finite inverse temperature η) alone yields sufficient exploration and tight regularized-regret bounds, without UCB/TS. It works with general function approximation via regression oracles and gives a closed-form Gibbs update.

Practical simplicity and scalability: Only one main knob (η, the inverse temperature); no confidence sets or posterior sampling machinery. The Gibbs update is cheap and plugs into diverse regressors, which makes the method easy to scale (important for RLHF/LLMs).

Stability via anchoring to a reference policy: KL(π || π_ref) both induces soft exploration and constrains distribution shift, a desirable property in RLHF (style/safety preservation). The role of temperature is conceptually clear: finite η>0 is needed; η→∞ (greedy) breaks exploration.

**Weaknesses:**

Experimental evaluation is underpowered and lacks key sweeps: Baselines likely under-tuned: regret of LinUCB/LinTS is sensitive to the exploration parameter (e.g., α in LinUCB, prior/variance in LinTS). These should be systematically swept (and reported) because regret curves can change materially with α. Small differences in average accuracy on RLHF benchmarks: Given the narrow gaps, results should include confidence intervals, multiple seeds, and statistical tests. Sensitivity to decoding/training randomness should be reported. η sensitivity/ablations are insufficient: Accuracy and regret typically depend on η (and any annealing schedule). It’s important to plot performance vs η and vs target KL levels to π_ref. I cannot confirm from the current context whether the authors ran a comprehensive η sweep; if not, this is a clear gap to address.

Limited scope beyond bandits (no MDP guarantees): The results hinge on single-step structure. Extending the second-order technique to MDPs faces fixed-point coupling, occupancy/concentrability, and error propagation issues; without strong coverage/mixing assumptions, KL alone may not ensure state-space exploration.

Dependence on assumptions and constants: Requires realizability and strong regression oracles; bounds include constants that can be large (e.g., κ from preference curvature, D = E[KL(π* || π_ref)]). If π_ref is far from optimal, guarantees degrade. Necessitates nonzero temperature during training (finite η). Deterministic training (η→∞) can yield poor exploration and invalidate the proof technique.

**Questions:**

Please refer to the weakness part.

---

> ### Author Response · Authors · 2025-11-22
> **Official Comment by Authors (1/3)**
>
> We appreciate the positive review and constructive feedback.
>
> The reviewer’s primary concerns seem to be concentrated on the experimental results, particularly regarding robustness and confidence. We conducted additional experiments—within the limits of time and resources—to better address these concerns.
>
> ---
>
> ### **Experiment**
>
> -  **Exploration parameter in LinUCB/LinTS:** We updated the linear contextual bandit experiments to more precisely align with the theoretical formulation. Specifically, we use the exact theoretical values for the confidence parameters in the baselines (e.g., $\alpha = \sqrt{2d \log(T/\delta)} + \lambda$ for LinUCB), and we use the theoretically optimal regularization parameter $\eta = \sqrt{T \log N / (2d \log T + 16 \log(1/\delta))}$ for our algorithm. Consequently, **no tuning of $\alpha$ (or $\eta$) is required**, as we directly adopt the values prescribed in the respective theoretical results.
>
>
>     | d  | N   | LinUCB           | LinTS            | LinPHE             | SupLinUCB            | KL-EXP (ours)       |
>     |----|-----|------------------|------------------|---------------------|-----------------------|---------------------|
>     | 5  | 50  | 302.06 ± 45.40   | 440.90 ± 73.82   | 602.85 ± 63.90      | 1486.69 ± 636.21      | **244.52** ± 78.35      |
>     | 5  | 100 | 297.72 ± 33.71   | 417.66 ± 64.97   | 594.41 ± 71.29      | 1497.95 ± 641.16      | **238.09** ± 77.78      |
>     | 20 | 50  | 478.25 ± 113.83  | 584.17 ± 182.24  | 614.89 ± 207.34     | 1105.45 ± 416.75      | **329.34** ± 40.41      |
>     | 20 | 100 | 443.73 ± 80.81   | 575.69 ± 177.43  | 622.88 ± 212.86     | 1104.46 ± 420.46      | **361.01** ± 55.66      |
>
>     The results show that our algorithm consistently outperforms all baseline methods by a large margin across different values of $d$ and $N$, even under purely theoretical parameter choices.
>
> - **Training randomness (RLHF)**: As requested by the reviewer, we include additional runs so that our results now cover three random seeds (the most we could reasonably run given our available compute and time), and we report the mean and standard error of the average accuracy across 17 benchmarks to ensure statistical reliability. The full set of results is reported in the revised version (Table H.5).
>
>     | Metric | Llama-3-8B-Flow-SFT | Llama-3-8B-Flow-Final | XPO    | OnlineDPO ($\eta=5.0$) | OnlineDPO ($\eta=8.5$) | OnlineDPO ($\eta=10.0$) | OnlineDPO ($\eta=12.5$) | OnlineDPO ($\eta=20.0$) |
>     |--------|----------------------|------------------------|--------|---------|---------|----------|----------|----------|
>     | Accuracy (%)   | 59.11              | 60.47                | 61.61 ± 0.044 | 61.90 ± 0.068  | 62.04 ± 0.141 | 62.00 ± 0.109 | **62.14** ± 0.121     | 62.02 ± 0.319     |
>
>     For the baselines Llama-3-8B-Flow-SFT ($\pi_{\text{ref}}$) and Llama-3-8B-FlowFinal (Dong et al., 2024), we evaluate the pretrained models released on Hugging Face; therefore, we do not report training randomness for these two baselines. The results show that our algorithm (OnlineDPO) with $\eta=12.5$ achieves the best performance, with relatively small standard deviation.
>
>     An interesting observation is that as $\eta$  becomes larger (and thus the regularization effect becomes weaker), the training randomness also increases. This is intuitive: a larger $\eta$ allows the policy to deviate more freely from the reference policy, leading to greater variability across runs.

---

> ### Author Response · Authors · 2025-11-22
> **Official Comment by Authors (2/3)**
>
> - **Decoding randomness (RLHF)**: Since it is standard practice to use deterministic inference (i.e., zero sampling temperature) when evaluating benchmark performance, our original results do not involve any decoding randomness. Hence, to address the reviewer’s concern, we instead report GPT-4o-mini–computed win rates (%) on the RLHFlow test dataset (https://huggingface.co/datasets/RLHFlow/test_generation_2k), comparing each model against the reference policy (Llama-3-8B-Flow-SFT).
>
>     | Sampling Temperature ($\tau$) | Llama-3-8B-Flow-Final      | XPO               | OnlineDPO ($\eta=12.5$) |
>     |------|--------------------|-------------------|------------------------|
>     | 0.3  | **58.38** ± 2.58       | 58.02 ± 1.86      | 58.03 ± 1.48           |
>     | 0.5  | 59.82 ± 2.74       | 60.09 ± 3.74      | **61.58** ± 3.66           |
>     | 0.7  | 59.98 ± 3.92       | 59.26 ± 2.64      | **60.50** ± 3.73           |
>     | 1.0  | 58.02 ± 1.91       | 58.76 ± 3.14      | **62.88** ± 2.08           |
>
>     We also report the win rates of OnlineDPO across different values of $\eta$:
>
>     | Sampling Temperature ($\tau$) | OnlineDPO ($\eta=20.0$)        | OnlineDPO ($\eta=12.5$)        |  OnlineDPO ($\eta=10.0$)        | OnlineDPO ($\eta=8.5$)        | OnlineDPO ($\eta=5.0$)        |
>     |------|------------------|------------------|------------------|------------------|------------------|
>     | 0.3  | 58.53 ± 2.12     | 58.03 ± 1.48     | **61.65** ± 1.62     | 59.19 ± 2.42     | 58.98 ± 2.97     |
>     | 0.5  | 59.63 ± 2.65     | **61.58** ± 3.66     | 59.61 ± 3.38     | 60.64 ± 2.84     | 58.40 ± 2.64     |
>     | 0.7  | 59.03 ± 2.11     | **61.50** ± 3.73     | 59.96 ± 1.74     | 58.31 ± 2.13     | 59.95 ± 3.21     |
>     | 1.0  | 61.90 ± 1.59     | **62.88** ± 2.08     | 62.33 ± 2.61     | 62.68 ± 2.30     | 60.21 ± 3.80     |
>
>     The results indicate that OnlineDPO with $\eta = 12.5$ outperforms the other baselines for sampling temperatures $\tau \in \\{0.5, 0.7, 1.0\\}$. Notably, OnlineDPO achieves its highest win rate at $\tau = 1.0$, which contrasts with the other baselines whose best performance typically occurs at $\tau \in \\{0.5, 0.7\\}$. This behavior is, however, consistent with our theoretical framework: the policy is trained at $\tau = 1.0$, and the regret is also defined with respect to the $\tau = 1.0$ policy. In other words, the primary objective is to minimize regret for the policy corresponding to $\tau = 1.0$, making this outcome expected. For the visual version, please refer to Figure H.1 in our revised manuscript.
>
>
> - **Performance vs. $\eta$ (RLHF):** We have **already conducted an extensive sweep over $\eta \in \\{5.0, 8.5, 10.0, 12.5, 20.0\\}$**, and the results are provided in Table H.3 in the Appendix. Please refer to that table for detailed comparisons.
>
>
> - **Performance vs. target KL (RLHF):** Could you clarify what “target KL” refers to? If it means the PPO-style hyperparameter that specifies a desired KL divergence between the current policy and a reference policy, then it does not apply here. Since our training uses a fixed regularization parameter $\eta$, there is no mechanism that adjusts the policy to match any target KL value, and such a parameter has no effect in our setup.
>
>     That said, we suspect the reviewer may have been interested in the reward–KL trade-off over the course of training. To address this, we additionally report the reward—evaluated by the ground-truth reward model (https://huggingface.co/RLHFlow/pair-preference-model-LLaMA3-8B) against the KL divergence at the end of each iteration. These results show that OnlineDPO attains the most efficient frontier, achieving the highest reward while still keeping the KL small—that is, it achieves the largest ratio of $\frac{\text{Reward}}{\text{KL}}$.
>
>     | Algorithm               |Iteration | KL          | Reward        | $\frac{\text{Reward}}{\text{KL}}$          |
>     |-------------------------|-|------------|---------------|---------------------|
>     | Llama-3-8B-Flow-SFT ($\pi_{\text{ref}}$) |     | 0.0         | -3.476       | -    |
>     | Llama-3-8B-Flow-Final   | | 10.919     | -0.690  | -0.063             |
>     | XPO                     | 1 | 11.530| -2.994 | -0.259             |
>     | XPO                     | 2 | 22.006| -1.750  | -0.079             |
>     | XPO                     | 3 | 27.549 | 1.051        | 0.038              |
>     | OnlineDPO ($\eta=12.5$)    | 1 | 12.440 | -0.334       | -0.026             |
>     | OnlineDPO ($\eta=12.5$)    | 2 | 24.946  | 0.697   | 0.028              |
>     | OnlineDPO ($\eta=12.5$)    | 3 | 27.289   | 2.293    | **0.084**              |
>
>     For the visual version, please refer to Figure H.2 in the revised version.

---

> ### Author Response · Authors · 2025-11-22
> **Official Comment by Authors (3/3)**
>
> ### **Extension to RL setting**
>
> As the reviewer pointed out, our current analysis does not directly extend to the multi-step RL setting; our focus is on the bandit and single-step RLHF framework, and extending the theory to full multi-step RL is beyond the scope of this work. That said, we believe there is promising potential to adapt our framework to *token-level MDP* formulations such as VPO (Rafailov et al., 2024) or to *Deterministic Contextual MDPs such* as XPO (Xie et al., 2024). Combining their analyses with ours could lead to a interesting direction for future work.
>
> ---
>
> ### **Assumptions and Parameter Dependencies**
>
> - **Realizability assumption:** The realizability assumption is **common** in previous works (Chu et al., 2011; Agarwal et al., 2012; Foster et al., 2018a; Foster & Rakhlin, 2020; Simchi Levi & Xu, 2022). Therefore, please do **not consider it as our unique weakness.**
>
> - **Applicability to offline regression oracle**: Regarding the reviewer’s concern about the online regression-oracle bound, we note that Appendix F already demonstrates that our method extends to the **offline regression-oracle** setting, which enjoys a well-known and stronger theoretical bound while requiring only $\log T$ oracle calls. Offline regression oracles are also better understood and often more computationally efficient. We hope this extension helps address the reviewer’s concern about the regression-oracle assumption.
>
> - **Preference curvature $\kappa$:**  The dependency on $\kappa$ is **common** across most RLHF and dueling-bandit analyses (Saha, 2021; Saha et al., 2023; Zhu et al., 2023; Xiong et al., 2023; Zhan et al., 2023b; Das et al., 2024; Xie et al., 2024; Zhao et al., 2024), and therefore should **not be viewed as a limitation specific to our work.** For the linear rewards, it may be possible to remove this dependence by leveraging recent techniques such as [1]. However, beyond the linear setting, eliminating the $\kappa$-dependence remains an interesting and important direction for future work.
>
>
>     [1] Chen, Mingyu, et al. "Avoiding exp ($R_{\text{max}}$) scaling in rlhf through preference-based exploration." NeurIPS 2025.
>
> - **Distance between $\pi_{\text{ref}}$ and $\pi^\star$:**
> First, we would like to clarify that for the KL-regularized regret—the main objective in KL contextual bandits and RLHF—**we can achieve logarithmic (KL-regularized) regret regardless of how large the KL divergence between $\pi_{\text{ref}}$ and $\pi^\star$ is.**
> Furthermore, for the unregularized regret, it is indeed natural that the (unregularized) regret becomes large when the reference policy $\pi_{\text{ref}}$ is far from the optimal policy $\pi^\star$. However, to the best of our knowledge, NO prior work has explicitly reflected this dependence in their theoretical bounds—in fact, unregularized regret is rarely considered in previous works, despite its importance. **Our analysis makes this relationship precise and mathematically explicit, and should therefore be regarded as a strength** of our paper rather than a weakness.
>
>
> - **Deterministic policy ($\eta \rightarrow \infty$)**: One of the central messages of our paper is that **selecting an appropriate value of $\eta$ is crucial** for achieving the $\sqrt{T}$-order (unregularized) regret. Setting $\eta$ to infinity directly contradicts this message. Furthermore, it is not clear to us why the absence of a deterministic policy during training should be viewed as a weakness. Since $\eta$ also plays the role of an exploration parameter, taking the limit $\eta \to \infty$ is analogous to setting the confidence parameter $\alpha$ to *zero* in LinUCB—thereby eliminating exploration, which is well known to lead to poor performance, and for which the standard analyses also completely break down.

---

### Official Review · Reviewer_8aug · 2025-11-04

**Soundness:** 3
**Presentation:** 3
**Contribution:** 3
**Rating:** 6
**Confidence:** 3

**Summary:**

This paper's central argument is that KL regularization alone is sufficient to guarantee efficient exploration in contextual bandits and Reinforcement Learning from Human Feedback (RLHF). The authors show that the additional, complex exploration strategies commonly used in prior work—such as UCB, Thompson sampling, or optimism—are "unnecessary".

**Strengths:**

1. A New, Simpler Algorithm: It proposes KL-EXP for contextual bandits and its variant OEPO for RLHF. Unlike previous methods, these algorithms do not use any explicit exploration bonuses (like UCB) and rely solely on the inherent exploration provided by the KL-regularized objective.

2. Logarithmic KL-Regularized Regret: The paper proves that KL-EXP/OEPO achieves logarithmic KL-regularized regret (e.g., $\mathcal{O}(\eta d \log T)$ for linear classes). This is the first theoretical result to show that a KL-regularization-only approach can achieve this standard, strong regret bound without extra exploration mechanisms6666

**Weaknesses:**

1. It is interesting that they do not need exploration, but in the proof sketch, it is not clear why their analysis do not need optimism. Could the authors provide more ideas about how to derive equation (5) in the proof sketch.

2. Their proof technique resembles this work [1]. They are suggested to cite this paper.

[1] Qingyue Zhao, et al, Towards a Sharp Analysis of Offline Policy Learning for f-Divergence-Regularized Contextual Bandits

3. For the experiment, it is better to ablate on the effect of different values of eta to study its effect on the performance.

**Questions:**

See weaknesses.

---

> ### Author Response · Authors · 2025-11-22
> **Official Comment by Authors (1/2)**
>
> Thank you very much for your positive review. Below, we provide our responses to your comments and concerns.
>
> ---
>
> ### **Clarification of Equation (5)**
>
> In Equation (5), we apply the exact second-order Taylor expansion of $f$. The first-order term vanishes because $\frac{\partial f(x_t,R^\star)}{\partial R(x_t,a)} = 0$. Therefore, it remains to bound the second-order term, which is easier to handle. With simple calculus, the second-order term can be bounded by $\eta \mathbb{E}[(\hat{R}_t - R^\star)^2]$, which is further controlled by the regression oracle bound.
>
>
> In contrast, previous works such as Zhao et al. (2025) use a first-order expansion and must handle the term $\frac{\partial f(x_t,\bar{R}_t)}{\partial R(x_t,a)}$, where $\bar{R}_t$ is the convex combination of $\hat{R}_t$ and $R^\star$. Since $\bar{R}_t$ is unknown, this first-order term cannot be eliminated and becomes difficult to bound. To handle this issue, their analysis depends on a UCB-style argument.
>
>
> ---
>
> ### **Additional citation**
> We have included the paper in our revised version. Thank you for the constructive feedback!
>
> ---
>
> ### **Ablation study on $\eta$**
>
> As per the reviewer’s request, we we additionally conduct ablation studies on $\eta$ for the linear and neural bandit experiments. Note that we have already included results for different values of $\eta$ for the RLHF experiments (see Table H.3 in the Appendix).
>
> **(1) Linear bandits**
>
> For the linear contextual bandit experiments, we made a slight update to the setup to more precisely reflect the theoretical approach (and to address Reviewer 5qSz’s concern). Specifically, we use the exact theoretical parameters for the confidence bounds in the baselines (e.g., $\sqrt{2d \log (T/\delta)} + \lambda$ for LinUCB), and we use the theoretically optimal regularization parameter $\eta^\star = \sqrt{ T \log N / (2 d \log T + 16 \log (1/\delta))}$ for our algorithm. With this updated setup, we now present the ablation study on $\eta$ of our algorithm, KL-EXP. We report the cumulative regret at the final round $T=5000$.
>
>
>
> | d      | N       |$0.2\eta^\star$             | $0.5\eta^\star$            | $\eta^\star$              | $2\eta^\star$                | $5\eta^\star$               |
> | ------ | ------- | --------------- | --------------- | -------------- | --------------- | --------------- |
> | **5**  | **50**  | 596.37 ± 112.63 | 367.31 ± 129.12 | 244.52 ± 78.35 | **222.57** ± 67.61  | 267.98 ± 88.32  |
> | **5**  | **100** | 508.08 ± 131.46 | 410.16 ± 152.76 | **238.09** ± 77.78 | 267.38 ± 213.09 | 320.29 ± 106.25 |
> | **20** | **50**  | 541.04 ± 227.71 | 342.24 ± 105.33 | 329.34 ± 40.41 | **321.84** ± 70.77  | 340.00 ± 76.04  |
> | **20** | **100** | 684.46 ± 212.17 | 416.29 ± 108.76 | **361.01** ± 55.66 | 379.26 ± 135.18 | 400.35 ± 106.67 |
>
> Below, we also report the updated cumulative regrets for the baseline algorithms, using the exact theoretical confidence bounds:
>
>
> | d  | N   | LinUCB           | LinTS            | LinPHE             | SupLinUCB            |
> |----|-----|------------------|------------------|---------------------|-----------------------|
> | 5  | 50  | 302.06 ± 45.40   | 440.90 ± 73.82   | 602.85 ± 63.90      | 1486.69 ± 636.21      |
> | 5  | 100 | 297.72 ± 33.71   | 417.66 ± 64.97   | 594.41 ± 71.29      | 1497.95 ± 641.16      |
> | 20 | 50  | 478.25 ± 113.83  | 584.17 ± 182.24  | 614.89 ± 207.34     | 1105.45 ± 416.75      |
> | 20 | 100 | 443.73 ± 80.81   | 575.69 ± 177.43  | 622.88 ± 212.86     | 1104.46 ± 420.46      |
>
>
> The results show that our algorithm outperforms all baseline methods when using the optimal $\eta^\star$,
> and remains consistently better across a wide range of values $0.5\eta^\star \leq  \eta \leq 5\eta^\star$. Moreover, the results highlight that choosing an appropriate value of $\eta$ is crucial for achieving strong performance, consistent with the theoretical results.

---

> > ### Author Response · Authors · 2025-11-22
> > **Official Comment by Authors (2/2)**
> >
> > **(2) Nueral bandits**
> >
> > Similarly, for the neural bandit experiments, we present the cumulative regret measured at the final round $T=4000$. Since the exact theoretical values are unknown for neural bandits (unlike in the linear bandit setting), we tune both the confidence bounds and $\eta$ using a predefined range of grid search. To clearly show how the regret behaves as $\eta$ varies, we include a wider range of $\eta$ values here than in the main manuscript.
> >
> >
> > | Reward Function       | $\eta = 50$                  | $\eta = 100$                   | $\eta = 500$                   | $\eta = 1000$                  | $\eta = 3000$                  | $\eta = 5000$                  |
> > |-----------------|----------------------|-----------------------|-----------------------|-----------------------|-----------------------|-----------------------|
> > | Linear          | 52.48 ± 2.01         | 27.17 ± 1.55          | **19.49** ± 1.12          | 20.05 ± 1.23          | 20.59 ± 1.52          | 21.96 ± 1.82          |
> > | Quadratic       | 134.61 ± 3.65        | 70.89 ± 2.29          | 51.57 ± 6.44          | 50.61 ± 4.88          | **46.16** ± 5.89          | 48.47 ± 4.12          |
> > | Cosine          | 211.67 ± 7.69        | 210.07 ± 6.12         | 207.85 ± 6.51         | **204.95** ± 9.72         | 210.89 ± 9.62         | 215.84 ± 10.02        |
> > | Neural Network  | 139.10 ± 2.35        | 83.55 ± 1.78          | 54.76 ± 1.24          | **53.75** ± 1.63          | 53.92 ± 1.53          | 58.58 ± 2.24          |
> >
> >
> > And the cumulative regrets for the baselines (with the best confidence bounds) are summarized below:
> >
> > | Reward Function       | NeuralUCB          | NeuralTS           |
> > |-----------------|--------------------|--------------------|
> > | Linear          | 29.56 ± 2.67       | 31.61 ± 2.85       |
> > | Quadratic       | 142.59 ± 16.75     | 108.89 ± 6.36      |
> > | Cosine          | 246.58 ± 6.73      | 250.42 ± 6.77      |
> > | Neural Network  | 79.43 ± 4.27       | 68.96 ± 1.80       |
> >
> > As in the linear setting, choosing an appropriate value of $\eta$ is essential for achieving strong performance, consistent with the theoretical results.

---

> ### Comment · Area_Chair_SqrN · 2025-11-22
> **Re: Zhao et al. (2025)**
>
> I checked the proof of Zhao et al [1], and found that your second order decomposition (Lemma B.2) is the same as their proof (Lemma 2.15 and Lemma 2.16). In particular, they prove $G(\gamma)$ is monotonic and its maximum value of $G(\gamma)$ is attained at $\gamma= 0$, which is essentially is statement of your Lemma B.2.
>
> [1] Qingyue Zhao, et al, Towards a Sharp Analysis of Offline Policy Learning for f-Divergence-Regularized Contextual Bandits

---

> > ### Author Response · Authors · 2025-11-26
> >
> > We sincerely appreciate the AC for thoroughly reviewing our paper. Our intention in this lemma was to derive the bound for the KL-regularized regret in terms of $\eta \mathbb{E}\_{\pi\_t}[(\hat{R}\_t - R^\star)^2]$, rather than $\eta \mathbb{E}\_{\pi\_\lambda}[(\hat{R}\_t - R^\star)^2]$ (as in Lemmas 2.15 and 2.16 of Zhao et al. 2025), where $\pi_\lambda$ is an unknown policy introduced through the mean value theorem. They are distinct. However, as the AC correctly pointed out in the comment above, our analysis contains a technical error, because of which, we decided to withdraw our paper.

---

### Author Response · Authors · 2025-11-22

We sincerely thank the reviewers for taking the time to review our paper, recognize its contributions, and provide thoughtful, detailed feedback. We have uploaded a revised version with the changes highlighted in **blue**.

---

### Comment · Area_Chair_SqrN · 2025-11-22
**There is an issue in the proof of Lemma B.5**

I believe there is an issue in the proof of Lemma B.5.

In particular, in Line 1301 in the revised paper (or Line 1174 in the original submission), you have

\begin{align}
f(x,\hat{R}\_t) - f(x, R^*) =  \eta \bigg[\int_{0}^{1} (1-\alpha) \mathrm{Var}\_{t}^{\alpha} (x) \mathrm{d} \alpha   + \int_{0}^{1} \alpha \frac{\mathrm{d} }{\mathrm{d} \alpha} \mathrm{Var}\_{t}^{\alpha}(x) \mathrm{d} \alpha \bigg]
\end{align}

It is incorrect and should be

\begin{align}
f(x,\hat{R}\_t)-f(x, R^*)  =  \eta \bigg [\int_{0}^{1} (1-\alpha) \mathrm{Var}\_{t}^{\alpha} (x) \mathrm{d} \alpha  + \int_{0}^{1} (1-\alpha)\alpha \frac{\mathrm{d} }{\mathrm{d} \alpha} \mathrm{Var}\_{t}^{\alpha}(x) \mathrm{d} \alpha  \bigg]
\end{align}
which, by integration by part, yields
\begin{align}
 =  \eta \bigg [\int_{0}^{1} (1-\alpha) \mathrm{Var}\_{t}^{\alpha} (x) \mathrm{d} \alpha  + [(1-\alpha) \alpha  \mathrm{Var}\_{t}^{\alpha}(x)] |_0^1 -  \int\_{0}^{1} (1-2\alpha) \mathrm{Var}\_{t}^{\alpha}(x) \mathrm{d} \alpha  \bigg]
\end{align}

\begin{align}
 =  \eta \bigg [\int_{0}^{1} (1-\alpha) \mathrm{Var}\_{t}^{\alpha} (x) \mathrm{d} \alpha  -  \int\_{0}^{1} (1-2\alpha) \mathrm{Var}\_{t}^{\alpha}(x) \mathrm{d} \alpha  \bigg]
\end{align}

\begin{align}
 =  \eta \int_{0}^{1} \alpha  \mathrm{Var}\_{t}^{\alpha} (x) \mathrm{d} \alpha
\end{align}

So the subsequent proofs in your paper appear to be incorrect due to the issue above. Could the authors please clarify this issue and examine whether it can be fixed?

---

### Note · Authors · 2025-11-26

**Comment:**

We sincerely appreciate the attention you gave to our paper, the time you took to carefully read our proofs, and for pointing out the technical error. We acknowledge that this is an error that cannot be immediately fixed. Hence, we decided to withdraw the paper.
We thank the AC for the detailed feedback and all reviewers for the time and effort they have devoted to evaluating our work.

**Withdrawal Confirmation:**

I have read and agree with the venue's withdrawal policy on behalf of myself and my co-authors.